

# The HD(CP)2 Observational Prototype Experiment HOPE – An Overview

Andreas Macke[1], Patric Seifert[1], Holger Baars[1], Christoph Beekmans[2], Andreas Behrendt[3], Birger Bohn[4], Johannes Bühl[1], Susanne Crewell[7], Thomas Damian[8], Hartwig Deneke[1], Sebastian Düsing[1], Andreas Foth[9], Paolo Di Girolamo[10], Eva Hammann[3], Rieke Heinze[5,6], Anne Hirsikko[4,13], John Kalisch[1,11], Norbert Kalthoff[8], Stefan Kinne[5], Martin Kohler[8], Ulrich Löhnert[7], Bomidi Lakshmi Madhavan[1,14], Vera Maurer[8], Shravan Kumar Muppa[3], Jan Schween[7], Ilya Serikov[5], Holger Siebert[1], Clemens Simmer[2], Florian Späth[3], Sandra Steinke[7], Katja Träumner[8,12], Birgit Wehner[1], Andreas Wieser[8], Volker Wulfmeyer[3], Xinxin Xie[2]

[1]Leibniz Institute for Tropospheric Research (TROPOS), Leipzig, Germany
[2]Meteorological Institute, University of Bonn, Bonn, Germany
[3]Institute of Physics and Meteorology (IPM), University of Hohenheim, Stuttgart, Germany
[4]Institute of Energy and Climate Research (IEK-8), Forschungszentrum Jülich GmbH (FZJ), Jülich, Germany
[5]Max-Planck-Institut für Meteorologie (MPIM), Hamburg, Germany
[6]Institut für Meteorologie und Klimatologie, Leibniz Universität Hannover, Hannover, Germany
[7]Institute for Geophysics and Meteorology (IGMK), University of Cologne, Cologne, Germany
[8]Institute of Meteorology and Climate Research - Troposphere Research (IMK-TRO), Karlsruhe Institute of Technology (KIT), Karlsruhe, Germany
[9]Leipzig Institute of Meteorology, University of Leipzig, Leipzig, Germany
[10]Scuola di Ingegneria, Università degli Studi della Basilicata, Potenza, Italy
[11]Department of Energy and Semiconductor Research, Institute of Physics – Oldenburg University, Oldenburg, Germany
[12]NDT Global GmbH & Co. KG, Stutensee, Germany
[13]Finnish Meteorological Institute, Helsinki, Finland
[14]Department of Marine Sciences, Goa University, Goa, India

*Correspondence to*: Andreas Macke (andreas.macke@tropos.de)

**Abstract.** The "HD(CP)2 Observational Prototype Experiment" (HOPE) was executed as a major 2-month field experiment in Jülich, Germany, performed in April and May 2013, followed by a smaller campaign in Melpitz, Germany in September 2013. HOPE has been designed to provide a critical evaluation of the new German community atmospheric Icosahedral non-hydrostatic (ICON) model at the scale of the model simulations and further to provide information on land-surface-atmospheric boundary layer exchange, cloud and precipitation processes as well as on sub-grid variability and microphysical properties that are subject to parameterizations. HOPE focuses on the onset of clouds and precipitation in the convective atmospheric boundary layer. The paper summarizes the instrument set-ups, the intensive observation periods as well as example results from both campaigns.

HOPE-Jülich instrumentation included a radio sounding station, 4 Doppler lidars, 4 Raman lidars (3, 3, and 4 of these provide temperature, water vapor, and particle backscatter data, respectively), 1 water vapour differential absorption lidar, 3 cloud radars, 5 microwave radiometers, 3 rain radars, 6 sky imagers, 99 pyranometers, and 5 Sun photometers operated in synergy





at different supersites. The HOPE-Melpitz campaign combined ground-based remote sensing of aerosols and clouds with helicopter- and balloon-based in-situ observations in the atmospheric column and at the surface.

HOPE provided an unprecedented collection of atmospheric dynamical, thermodynamical, and micro- and macrophysical properties of aerosols, clouds and precipitation with high spatial and temporal resolution within a cube of approximately 10 x 10 x 10 km$^3$. HOPE data will significantly contribute to our understanding of boundary layer dynamics and the formation of clouds and precipitation. The datasets are made available through a dedicated data portal.

## 1 Introduction

Clouds and precipitation play a central role in the climate system and were repeatedly identified as the largest problem in a realistic modelling of atmospheric processes, forcing and feedbacks (IPCC, 2013;Jakob, 2010). Uncertainties in the characterization of clouds and precipitation have manifold consequences on virtually all non-atmospheric climate components from ocean mixed layer stability to vegetation variability, to net mass balance of ice sheets.

To achieve progress in the improvement of the representation of clouds and precipitation in atmospheric models, the German research initiative "High Definition Clouds and Precipitation for advancing Climate Prediction" HD(CP)$^2$ was launched. HD(CP)$^2$ aims at a significant reduction in the uncertainty of climate change predictions by means of better resolving cloud and precipitation processes. The newly developed convection-resolving HD(CP)$^2$ model will be used to develop new convection parameterizations for large-scale eddy simulation models. It is a coordinated initiative to provide atmospheric scenarios, including multiple thermodynamic phases, multi-mode microphysics, and a realistic orography with high spatial resolution of 100 m in the horizontal and 10 - 50 m in the vertical at a temporal resolution of 1-10 s over climatologically relevant scales, i.e. over several thousand kilometres and several years. The 100-m scale is believed to be most critical for the onset of clouds and precipitation as it sufficiently resolves the convective boundary layer and cloud formation (Stevens and Lenschow, 2001). The anticipated high resolution shall thus enable to associate differences in modelled and observed atmospheric fields to problems with the dynamical core or with parameterizations of physical processes rather than with resolution issues.

The HD(CP)² project consists of a modelling, an observational, and a synthesis part (see http://www.hdcp2.eu for further information concerning the overall project descriptions and goals). As a first step of HD(CP)$^2$, the high-resolution HD(CP)$^2$ model in LES mode must be evaluated in order to test the suitability for parameterization development application. The test bed for these observations was provided by means of HOPE.

Within the M-module (modelling) of HD(CP)$^2$, the new ICON (Icosahedral non-hydrostatic) general circulation model was developed (Zängl et al., 2015) and its performance in LES modelling was evaluated (Dipankar et al., 2015). The O-module (Observations) was defined to provide observational datasets for both initialization and evaluation of the newly developed ICON model as well as for the development of new parameterizations that are suitable for application in a high-resolution model. The scope of the S-module (synthesis) was to provide first improvements of parameterizations from the use of model





and observation results. Key to this effort was the provision of modelled scenarios at 100 m grid resolution over thousands of kilometres, which will be used to analyse, improve or develop parameterizations related to cloud and precipitation development in climate models.

The O4 project in the O module of HD(CP)$^2$ - denoted as "HD(CP)$^2$ Observational Prototype Experiment" HOPE has been

designed to provide a critical model evaluation at the scale of the model simulations and further to provide information on sub-grid variability and microphysical properties that are subject to parameterizations even at high-resolution simulations. In order to derive the atmospheric state and the 3-d fields of water vapor, temperature, wind and cloud and precipitation properties at the scale of 100-m resolution for an area of about $10 \times 10 \times 10$ km$^3$ three close-by supersites, separated by a distance of approximately 4 km, complemented by larger networks were deployed. HOPE focuses on the onset of clouds and precipitation

in the convective atmospheric boundary layer. The experiment complements the larger spatiotemporal Full-Domain (O2) and Supersites (O1) activities in the observations module in HD(CP)$^2$ providing continuous time series of 2D fields across the HD(CP)$^2$ domain and 1D profiles at four dedicated locations, respectively. The scope of Module O3 was to establish a data flow from the observation modules to the model and synthesis modules. In 2016 HD(CP)$^2$ entered its second phase, which puts a much stronger effort on the synthesis part.

HOPE builts on the experience gained in previous field campaigns like the Convective and Orographically-induced Precipitation Study (COPS) (Wulfmeyer et al., 2011), however, with a stronger focus on multi-sensor synergy covering a micro- to meso-scale domain. COPS aimed at the observation of orographically driven initiation of convection with supersites several tens of km apart in strongly structured terrain. Complementary to COPS, HOPE is covering a smaller domain with higher resolution, and is accompanied by long-term supersite observations within the framework of the heavily instrumented

Rur catchment Terrestrial Environmental Observatoria TERENO Programme (Simmer et al., 2015) around the ground-based remote sensing supersite JOYCE (Löhnert et al., 2015), and the TROPOS long-term aerosol observatory in Melpitz (Spindler et al., 2012).

This article mainly serves as a guide through the sites and instrumentation used during the HOPE campaigns and it is aiming on giving a motivation to learn about the details and specific conclusions described in the individual publications this overview

is built upon. The structure is as follows. Section 2 describes the site setups and measurements performed during HOPE including information about the meteorological conditions and data availability. Examples from each of the research topics are presented in section 3. In section 4, first comparisons between models and observations are discussed. A summary and conclusions on the further applications of the HOPE data as well as designs for future observational strategies are presented in section 5. Individual work performed during HOPE is published in the ACP/AMT HOPE special issue (Buehler and

Russchenberg, 2016) and other journals and is cited in the present overview correspondingly.

## 2 Description of field campaigns

The measurement activities during HOPE mainly consisted of a major field experiment in Jülich, Germany, denoted as HOPE-Jülich, conducted from April 3 to May 30, 2013 followed by a smaller campaign that was performed in Melpitz, denoted as



HOPE-Melpitz, Germany, which was conducted from September 9 to September 29, 2013. Figure 1 gives an overview of the broad spectrum of instruments installed during the two campaigns and their overall setup. A detailed introduction is given below.

### 2.1 Instrumentation

#### 2.1.1 HOPE-Jülich

In order to derive the atmospheric state of water vapor, temperature, wind and cloud and precipitation properties with 100-m resolution for an area of about 10x10x10 km three close-by (ca. 4 km) supersites complemented by larger networks were operating. Figure 2 gives an overview about the different sites and networks within HOPE-Jülich, which are further described in Table 1. The monitored area encompasses approximately 40 km in radius around the Research Centre of Jülich. The natural

topography around Jülich is rather flat with an average elevation of around 100 m above sea level (asl). Approximately 20 km south of Jülich the Eifel Mountains approach up to 800 m asl. Locally, within a radius of 10 km, the area around Jülich is dominated by open pit coal mining. Two open pit mines are located within 1-3 km east and west of the HOPE-Jülich area, respectively. Along a 10-km line between these two pit mines, the elevation range spans over 571 m, from as low as -270 m asl within the pit mines (pit mine of Hambach, see Figure 2) to 301 m asl at the top of the debris hill Sophienhöhe. The

instruments and observations were deployed at supersites in the rather flat terrain between the pit mines or within networks. The TERENO sites as well as the X-band radar sites JuXPol and BoXPol that are shown in Figure 2 also contributed to the HOPE observations, even though they are operated in the frame of other research projects, mainly Terrestrial Environmental Observatories (TERENO) (Zacharias et al., 2011) and the Transregional Collaborative Research Centre 32 (TR32) (Simmer et al., 2015), which are implemented for longer time periods than it was the case for HOPE.

As can be seen from Table 1, most instruments were deployed at the three supersites Jülich (JUE), Krauthausen (KRA), and Hambach (HAM) with its outpost close to a pump station "Wasserwerk" (WAS). At each supersite one or several main remote-sensing facilities were deployed. At JUE this was the instrumentation of the permanently installed Jülich ObservatorY for Cloud Evolution (JOYCE), at HAM the Karlsruhe Institute for Technology mobile facility KITcube and the lidar systems of the Institute for Physics and Meteorology (IPM) of the University of Hohenheim (UHOH) were deployed, and at KRA the

Leipzig Aerosol and Cloud Remote Observations System (LACROS) was operated. In some publications that are based on HOPE-Jülich observations, the supersite names are also referring to the main facility deployed at each site, e.g. LAC for LACROS at the supersite KRA, JOY for JOYCE at the supersite JUE, and KIT for KITcube at the supersite HAM. The instrumentation that was present at each site is listed in Table 2. In total, the HOPE-Jülich set of instruments included a radio sounding station, 5 Doppler lidars, 4 Raman lidars, 1 differential absorption lidar (DIAL), 3 cloud radars, 5 microwave

radiometers (MWR), 3 precipitation radars, 6 sky imagers, 99 pyranometers, and 5 Sun photometers. Below, the operating institutions and available measurement devices at all three supersites are briefly outlined.





All measurements during HOPE-Jülich were built around the central supersite Jülich where JOYCE (Löhnert et al., 2015) is operated continuously at the Research Center Jülich. JOYCE (http://www.joyce.cloud ) is a joint research initiative of the Institute for Geophysics and Meteorology (IGMK) of the University of Cologne and the Jülich Research Centre (FZJ). It is permanently installed at FZJ.  Amongst other instruments (see Löhnert et al. (2015)), JOYCE contributed to HOPE with

observations of a continuously scanning cloud radar, a Doppler lidar, and three microwave radiometers (one continuously scanning, one vertically pointing, and one continuously obtaining temperature profiles) for the spatiotemporal characterisation of humidity and liquid water fields. The observations at the supersite Jülich were supported by high-resolved measurements of the vertical profile of the atmospheric temperature and water vapour mixing ratio, both at daytime and at night, which have been performed with the multi-wavelength polarization Raman lidar system BASIL of the Università degli Studi della

Basilicata (UniBas), Italy (Di Girolamo et al., 2009) and the lidar system ARL-2 of the Max Planck Institute for Meteorology (MPIM) (Wandinger et al., 2016). Temperature and moisture turbulent fluctuations have been observed by BASIL and are reported by Di Girolamo et al. (2016). BASIL as well as the ARL-2 lidar also provided measurements of aerosol scattering properties at 355, 532, and 1064 nm.

With the newly designed observing system KITcube (Kalthoff et al., 2013), the Institute of Meteorology and Climate Research

(IMK) of the Karlsruhe Institute of Technology (KIT) provides meteorological and convection-related parameters and contributed to measurements of the development of clouds with high temporal and spatial resolution in the HOPE area. KITcube was the main facility at the supersite Hambach (HAM) and consists of a surface-based network with meteorological stations and a 30-m tower measuring the standard parameters of temperature, humidity, air pressure, wind speed and direction, sensible heat fluxes, the energy balance components at the Earth's surface (Kalthoff et al., 2006) as well as soil moisture and

soil temperature profiles (Krauss et al., 2010). These stations in general are distributed over the whole area of KITcube to account for surface inhomogeneity. For instance, KIT operated two Eddy-Covariance stations – one at the main site HAM, and a second one at the outpost WAS, approximately 2.5 km to the west. KITcube also includes scanning Doppler wind lidars to measure wind speed, wind direction, and turbulence characteristics in the convective boundary layer. One WindTracer was installed at supersite HAM, a second WindTracer at the outpost WAS (see Fig 2b) to allow Dual-Doppler applications. Both

were installed together with a Windcube. Additionally, a Doppler lidar of KIT IMK-IFU (HALO Streamline) was operated at the TERENO site Selhausen. These instruments were complemented by a microwave radiometer which determines temperature and humidity profiles, a scanning cloud radar monitoring the development of clouds, a vertically pointing micro rain radar and disdrometers providing information about precipitation, and a ceilometer for cloud base height detection. At a second KITcube outpost denoted KiXPol, approximately 7.5 km southwest of HAM, an X-band rain radar was operated. In-

situ vertical profiles of temperature, humidity, and wind profiles as well as convective indices were gathered by radiosondes launched regularly every 6th full hour at the KITcube main site. Land and full-sky images were taken by S14 camera systems at HAM and WAS.

Also at supersite HAM, two lidar systems from the Institute for Physics and Meteorology (IPM) of the University of Hohenheim observed 3D thermodynamic fields of temperature and moisture including their turbulent fluctuations. A





temperature rotational Raman lidar (TRRL) measured temperature profiles (Radlach et al., 2008;Hammann et al., 2015) and a differential absorption lidar (DIAL) measured absolute humidity profiles (Behrendt et al.;Späth et al., 2016;Wagner et al., 2013). Both systems have scanning capability and an intrinsic high spatial and temporal resolution of 1-10 s and 15-100 m up to a range of about 5 km. Consequently, both systems are capable of resolving turbulent fluctuations in the convective boundary

layer from the surface to the entrainment zone. Derived products include statistical moments of moisture and temperature turbulent fluctuations (Behrendt et al., 2015;Wulfmeyer et al., 2010), profiles of stability variables such as buoyancy (Behrendt et al., 2011) and the boundary layer depth, aerosol backscatter fields and cloud boundaries. The self-calibrating DIAL technique has excellent absolute accuracy (Bhawar et al., 2011) and has been acknowledged as water-vapour reference standard of WMO. Continuous observations with the TROPOS mobile facility LACROS (Leipzig Aerosol and Cloud Remote Observations

System) (Bühl et al., 2013) were performed at the supersite KRA. LACROS employs a 35-GHz cloud radar, a multi-wavelength Raman polarization lidar, a ceilometer, a Doppler lidar, a microwave radiometer, an optical disdrometer, as well as an all-sky imager. The Raman polarization lidar Polly$^{XT}$ (Engelmann et al., 2016),  deployed at supersite KRA, is part of the lidar network PollyNet (Baars et al., 2016a) and provides automatically derived profiles of aerosol scattering properties and water vapour mixing ratio. Observations of the vertical velocity in the boundary layer and at cloud bases were provided

by the Doppler Wind lidar WiLi (Bühl et al., 2012). The focus of the LACROS observations was set on the continuous vertical profiling of the full tropospheric column to derive aerosol and cloud microphysical properties and cloud droplet dynamics (Bühl et al., 2016). LACROS at supersite KRA as well as JOYCE at supersite JUE are part of Cloudnet (Illingworth et al., 2007) providing a target categorization mask and microphysical parameters of clouds based, amongst others, on co-located vertically pointing observations of at least a cloud radar, a lidar and a microwave radiometer.

Beside the supersite observations at JUE, KRA, and HAM, also different instrument networks were distributed in the vicinity of the three supersites. The PYR network of 99 autonomous meteorological stations including pyranometers developed by TROPOS (Madhavan et al., 2016b) was deployed within a radius of about 5 km around the supersite JUE to capture the broadband downwelling solar irradiance with high spatial and temporal resolution. The Meteorological Institute of the University of Bonn (MIUB) coordinated the operation of 6 sky imagers within the SKY network that were provided by several

partner institutes to obtain imagery for cloud classification and the determination of cloud morphology (Beekmans et al., 2016). Three scanning polarimetric X-band rain radars jointly operated within the XRD network by the University of Bonn (BoXPol), the Jülich Research Centre (JuXPol) (Diederich et al., 2015) and KIT (KiXPol) provided 3D fields of polarimetric moments over the domain and precipitation estimates (Heinze et al., 2016;Trömel et al., 2013;Xie et al., 2016). Within the Sun photometer network (SUN), the vertically integrated aerosol characteristics and water vapour field at the three HOPE-Jülich

supersites as well as at two more-remote sites (Aachen and Insel Hombroich, see Table 1) were derived. Except for the one operated within JOYCE at supersite JUE, all Sun photometers were provided by NASA Goddard Space Flight Center (GSFC), Langley, USA, and operated by MPIM.

Additionally, two ground-based scanning spectral radiometers SpecMACS, from the Munich Institute for Meteorology (MIM) of the Ludwig-Maximilians Universität Munich (Ewald et al., 2015), and EAGLE from Leipzig Institute of Meteorology (LIM)





of the University of Leipzig (Jäkel et al., 2013) participated in the campaign. These instruments provide the solar radiation reflected at cloud sides from which vertical profiles of cloud microphysical properties shall be inferred.

### 2.1.2 HOPE-Melpitz

The HOPE-Melpitz campaign basically combined the remote sensing of aerosol and cloud properties based on the LACROS supersite with the helicopter-borne Airborne Cloud Turbulence Observation System ACTOS (Siebert et al., 2013) (see Figure 1b). The follow-up campaign HOPE-Melpitz has become necessary because of problems with the availability of a helicopter carrying ACTOS during HOPE-Jülich.

The Melpitz site (12.928° E, 51.525° N, 86 m asl) is the TROPOS research station for the continuous physical and chemical in-situ aerosol characterization of background aerosol characteristics in central Germany (Spindler et al., 2012). The site is located in a rural area, 40 km northeast of Leipzig (Figure 3). The topography around the Melpitz site is rather flat over an area of several hundred square kilometres, ranging between 100 m asl and 250 m asl. Melpitz is part of the European Monitoring and Evaluation Programme (EMEP) (Tørseth et al., 2012) and provides a comprehensive set of in-situ observed

chemical, microphysical and optical aerosol properties. Based on the co-location of the ground-based aerosol instrumentation, the airborne ACTOS platform, and the remote-sensing facility LACROS, the HOPE-Melpitz campaign thus provides the opportunity to investigate the relationship between tropospheric aerosols and clouds and aerosol conditions.

Similar to HOPE-Jülich, during HOPE-Melpitz the LACROS instrumentation comprised the polarization Raman lidar Polly[XT]-OCEANET (Engelmann et al., 2016) with near-range capabilities, a Humidity-Temperature Profiler (HATPRO) microwave

radiometer, the Doppler Wind lidar WiLi, 50 pyranometers, an all-sky imager, and a radiosonde station (provided from KITcube, see Table 2). Two sun photometers were installed, one at the site of Melpitz and one at TROPOS in Leipzig (51.3° N, 12.4° E, 120 m asl) in order to distinguish rural and urban aerosol conditions.

Measurements of the broadband irradiances at the surface were carried out with a mobile station following the recommendations of the Baseline Surface Radiation Network (McArthur, 2005), and can serve as high-quality reference for

the pyranometer network. In addition, spectral irradiances were observed with a rotating shadowband radiometer of type GUVis-3511 (Witthuhn et al., 2016).

Detailed information on the ACTOS setup are given in Siebert et al. (2013). ACTOS provides dynamic, thermodynamic as well as cloud and aerosol microphysical properties of warm shallow boundary layer clouds. The standard ACTOS instrumentation comprises sensors for the wind vector, temperature, and humidity under clear and cloudy conditions. Observed

microphysical parameters of liquid clouds include the cloud droplet number-size distribution in the range from 1 to 180 µm as well as the integral properties of this cloud droplet spectrum, e.g., liquid water content and effective radius. Aerosol number-size distributions for the size range from 8 nm to 2.8 µm are obtained with a resolution of 2 minutes. The total aerosol number concentration was recorded in the aerosol particle size range from 8 nm to 2 µm with 1 Hz resolution (Düsing et al., 2016) and





with 50 Hz resolution (Wehner et al., 2011). Additionally, a mini-CCNC (Cloud Condensation Nuclei Counter) was used for measuring the CCN number concentration at different supersaturations.

The two ground-based spectral radiometers EAGLE and SpecMACS from LIM and LMU, respectively, that were operated during HOPE-Jülich, were also deployed during HOPE-Melpitz. Besides ACTOS, airborne observations with spectral
radiometers for cloud remote sensing from the Freie Universität Berlin (Schröder et al., 2004) were performed on some days.

## 2.2. Datasets

### 2.2.2 HOPE-Jülich

HOPE-Jülich was conducted from 3 April to 31 May 2013 as this period in the year favours low-level cloud formation. Only the measurements of the pyranometer network PYR continued until end of July to capture high-sun conditions. An extensive
operation plan, documenting the daily availability of all central instruments of HOPE-Jülich can be found in the supplementary material to this article.

The weather conditions during the campaign varied from several warm and cold front passages interrupted by a few high pressure systems with high-level cirrus clouds at the beginning of the campaign and more low-level convective clouds later on. Since the campaign focused on the onset of clouds and precipitation, IOPs have been called out whenever clear skies,
boundary layer clouds, or precipitation developing clouds were forecast. During IOPs, instruments requiring continuous human control were measuring in addition to autonomously operating instruments. Furthermore, radiosondes were launched more frequently at supersite Hambach, depending on the weather situation and its variability. Table 3 summarizes the IOPs during HOPE-Jülich and the corresponding weather conditions. IOPs with especially well suited weather conditions have been labelled as "Golden Days" and have been more deeply analysed by all participating groups.
As an example, a detailed depiction of IOP7 consisting of a turbulently driven boundary layer development topped with afternoon single cumulus clouds in the afternoon can be found in Löhnert et al. (2015). There, it is demonstrated that a holistic view of the daily development of the boundary layer is only possible through the synergetic treatment of different ground-based remote sensors.

### 2.2.2 HOPE-Melpitz

Weather conditions have not been optimal for the helicopter operations due to problems with low-level overcast clouds (no flight permit inside clouds) and icing conditions. During the three weeks of the campaign, five IOPs have been performed on which 10 ACTOS flights were performed, covering 15 hours of measurements (Table 4).  However, the helicopter flights captured a spectrum of different meteorological conditions as can be seen from Table 4.





### 2.2.3 Data availability

All officially participating partners have submitted their quality controlled data and in a common format to the HD(CP)$^2$ Data Archive Center. Data processing of specific sensors (i.e. microwave radiometer, cloud radar, ceilometer) deployed by different supersites was made uniform. All the data processing is documented by means of metadata. See Stamnas et al. (2016) for a

detailed overview on the data format and data base. All data will be public available by the end of 2016.

## 3 Results

### 3.1 Near-surface wind field and energy budget

An essential regime that was observed during HOPE is the turbulent structure of the atmospheric boundary layer (ABL). To capture this regime both the surface energy budget components and the near-surface and lower boundary layer wind fields are

required. The set of instruments available during HOPE-Jülich provided a unique opportunity to compare and to correlate vertical-velocity variances from different locations. Maurer et al. (2016) made use of a triangular set-up of three KITcube Doppler lidar systems deployed approximately 3 km apart from each other. This distance was assumed to be sufficient to ensure that the lidars do not monitor the same convective cells at the same time. Nevertheless, they found persistent similar statistical properties of velocity variances measured along the wind direction in contrast to measurements across the wind

direction. This indicates that local organized structures of turbulence can dominate turbulence characteristics and that single turbulence measurements may not be representative for a larger domain.

In a similar approach Träumner et al. (2015) investigated correlation patterns of near-surface wind fields from a Dual Doppler lidar set-up scanning at low elevation angles together with available in-situ wind vectors from ground-based stations. As a measure for anisotropy, integral length scales were defined for the along-stream and the cross-stream wind components.

Integral scales provide a measure of the spatial or temporal dimension of turbulent eddies (Wyngaard, 2004). The authors confirmed previous findings of streak-like structures elongated and aligned in the wind direction. Also periodic behaviour in the horizontal wind fields has been identified occasionally. Interestingly, the mean structural pattern could be related to the background wind speed and the atmospheric stability. Still, individual wind fields can vary strongly for the same external forcing. Thus, a characterization of coherence pattern in the otherwise turbulent boundary layer requires extensive

spatiotemporal averaging.

Eder et al. (2015) investigated the complete surface energy budget and tested the hypothesis whether so-called turbulent organized structures (TOS), low-frequency structures that fill the entire atmospheric boundary layer, are a major cause for the frequent unclosed surface energy balances as they contribute to the vertical energy fluxes. In fact, by means of data from horizontally and vertically scanning Doppler lidars the authors could show that TOS with time scales larger than 30 minutes

extend deep into the surface layer. This finding implies that future turbulent energy exchange studies require the full 3D field of humidity, temperature and velocity in high spatio-temporal resolution, which was also pointed out and elaborated in Wulfmeyer et al. (2016).





Based on the autonomous pyranometer network described in Madhavan et al. (2016b), the representativeness of a single station measurement for spatially extended domains with different area sizes has been investigated. This is an important aspect for the evaluation of model results with observations, where point measurements are mostly compared to grid-box means, and are thus implicitly assumed to have similar statistical properties. Spatial and temporal smoothing have been quantified which limit

the representativeness of a point measurement for its surrounding domain size and period. Spatial averaging acts as a low-pass filter and reduces or even completely removes high-frequency spatio-temporal variations. This is illustrated in Figure 4(a), which shows a wavelet-based power spectrum obtained from 99 pyranometer stations, and corresponding estimates of the power spectra for three areas ranging from 1x1 km$^2$ to 10x10 km$^2$ in size under broken-cloud conditions. Figure 4(b) shows the explained variance (square of Pearson correlation coefficient) of temporal fluctuations of a point measurement and a spatial

domain as a function of frequency. It demonstrates the second effect, which describes that the correlation of temporal fluctuations decreases with increasing frequency. The combination of both effects adds up to the total deviation of a point measurement from the spatial mean of an extended domain, which is presented in Figure 4(c). The magnitude of this deviation depends on the domain size, the averaging period, and the synoptic conditions. Broken clouds cause the largest deviations, reaching about 30 and 80 W m$^{-2}$ for 3 hourly and second-resolution observations, respectively, and a 10x10 km$^2$-sized domain.

Also based on the highly-resolved spatiotemporal pyranometer measurements performed by TROPOS, Lohmann et al. (2016) analysed the statistics of spatiotemporal irradiance fluctuations with a strong application-oriented focus on photovoltaic power systems. They specifically calculated single-point statistics and two-point correlation coefficients for clear, overcast and mixed skies. The statistics for clear and overcast skies show similar behaviour as in previously published work, see Lohmann et al. (2016) for references. In order to account for conditions for a distributed PV system, they defined so-called irradiance

increments as changes in transmissivities over specified intervals of time, and showed that these increments are more strongly averaged out in space than the transmissivities themselves. By conditioning the sky type - which can easily be done from the irradiance measurements themselves - they demonstrated that the probability for strong irradiance increments is twice as high compared to increment statistics computed without distinguishing between different sky types.

As clouds impose the largest short-term variability in solar irradiance at the surface the analysis of cloud advection and

subsequent extrapolation represents a reasonable approach for short-term irradiance forecasts. Schmidt et al. (2016) made use of time series of hemispheric sky images to predict the surface irradiance by means of mapping the cloud position, which in turn is translated into shadow maps at the surface. The temporal evolution of such shadow maps is calculated from cloud motion vectors that were calculated from subsequent sky images. Irradiance forecasts of up to 25 minutes have been produced and were validated against the network of pyranometers described in Madhavan et al. (2016b). Although these sky-imager

based forecasts do not outperform a simple persistence forecast on average, improved forecast skill was found for convective cloud conditions with high cloud and irradiance variability. This finding may provide useful application in photovoltaic electricity production.





### 3.2 The turbulence structure of the boundary layer and clouds

The goal of the HD(CP)[2] project was to realize and to evaluate a model run spanning the area of whole Germany at the horizontal scale of 100 m. At such a small scale certain parameterizations for organized turbulent motions such as those that define the atmospheric boundary layer, and areas of shallow convection are supposed to be not required anymore and the

model setup is comparable to the one of a large-eddy-simulation (LES), wherein the sub-grid parameterizations are simpler and have less impact on the model performance (Bryan et al., 2003;Deardorff, 1970).

The increased model resolution puts new requirements on evaluation techniques. The HOPE experiments provided an optimum test bed for novel applications to derive boundary layer fluxes and turbulence characteristics. Observations of the turbulent fluxes of thermodynamic properties in the PBL, such as of temperature and water vapour, provide detailed information on the

minimum resolution required by a model to capture the turbulence spectrum down to the inertial sub-range and consequently to resolve the major part of the turbulent fluctuations. This value is in here introduced as the integral scale. During HOPE-Jülich, it was possible to derive the statistics of turbulent temperature fluctuations and thus of the integral scale of this parameter in the PBL with lidar (Behrendt et al., 2015). In addition to commercially available Doppler lidar systems, which provide turbulent wind fluctuations, three water vapour research lidars were deployed during HOPE-Jülich, which provide turbulent

humidity fluctuations that were documented by Di Girolamo et al. (2016), as well as Muppa et al. (2016). As the authors of the above-mentioned studies note, HOPE-Jülich provided for the first time data to observe the turbulence characteristics of the PBL up to the fourth statistical moment, i.e., the mean, standard deviation, variance, skewness, and kurtosis of the spatiotemporal water vapour and temperature fields were derived. Examples of the relationship between the integral scales of humidity and temperature fluctuation and height above ground within the convective boundary layer for the 20 April 2013

(IOP 5), 11:30-13:30 UTC , (only temperature fluctuations, see Di Girolamo et al. (2016)) and 24 April 2013 (IOP 6), 10:00-11:00 UTC (temperature and humidity fluctuation, see Behrendt et al. (2015) and Muppa et al. (2016)), respectively, are depicted in Figure 5. A general feature that was found during the investigated clear-sky days and which can also be seen in Figure 5 was that the integral scale decreases from the ground towards the top of the convective boundary layer. This is due to the decrease in the size of the turbulent eddies with height which is a result of the entrainment of dry free-tropospheric air at

the PBL top (Couvreux et al., 2005) which is also characterized by an increase in the variance of the temperature or water vapour toward PBL top. Converting the observed time scales shown in Figure 5 to spatial scales assuming horizontal and vertical wind velocities of 5 m s$^{-1}$ and 1 m s$^{-1}$, respectively, results in horizontal and vertical integral length scales of 100-1000 m and 20-200 m, respectively. Thus, in order to capture the full turbulence spectrum in the PBL, also a numerical model simulation should be run at temporal and spatial resolutions that are higher resolved than the observed values.

Detailed convective boundary layer (CBL) turbulence characteristics from HOPE and further field campaigns (Wulfmeyer et al., 2016) showed that the combination of active temperature-, humidity- and wind-profiling applied during HOPE-Jülich sufficiently resolves the turbulence structure of the CBL and lays the ground for new boundary layer turbulence parameterizations.



In addition to turbulent fluxes in the cloud-free planetary boundary layer, the turbulence characteristics of a stratocumulus layer were investigated simultaneously with ACTOS and the Doppler lidar WiLi of the LACROS site on 22 September 2013 during HOPE Melpitz (Seifert et al., 2016). The inter-comparison shown in Figure 6 presents a sequence and a histogram of the vertical velocities observed with ACTOS (red) and WiLi (blue). Thus, vertical velocities in the stratocumulus are similar

at the cloud base (observed with the Doppler lidar) and cloud top (observed with ACTOS). This is an important fact for Doppler-lidar studies of stratocumulus clouds, because it infers that Doppler lidars are suitable to characterize the turbulence characteristics of entire stratocumulus cloud layers. From the vertical-velocity observations of WiLi and ACTOS also integral length scales were derived which were in the range of 150 to 250 m. Thus, these cloud observations provide confidence that running a model in the 100-m range is sufficient to resolve also the energy-containing eddies in stratocumulus layers.

Furthermore, a combination of lidar and microwave radiometer data has been used to infer the height of the stable nocturnal boundary layer from aerosol-induced lidar backscatter variance and microwave radiometer derived potential temperature profiles (Saeed et al., 2016).

### 3.3 Thermodynamic properties of the atmosphere

Besides wind vectors, profiles of atmospheric temperature and humidity are the main drivers of numerical weather forecast-models and key for the verification of climate and Earth system models. An overview of their importance and the requirements set to observing systems is presented in Wulfmeyer et al. (2015). For models explicitly resolving turbulent processes (such as the HD(CP)² model), it is important to capture small-scale water vapour and thermodynamic stability fluctuations, which can trigger convection. Evaluation as well as data assimilation procedures for these models require advancements in measurement

accuracy as well as in spatial and temporal resolution.

From the multi-sensor observations available for the HOPE-Jülich experiment, Steinke et al. (2015) investigated the comparability and range of applicability of various sensors for the determination of the integrated water vapour (IWV). As can be seen in Figure 7, in general a good agreement was found between the IWV observations from Global Positioning System (GPS) stations (Gendt et al., 2001), microwave radiometer, Sun photometer, radiosonde. The systematic difference and

standard deviation were derived to be approximately 0.4 kg m$^{-2}$ and 1 kg m$^{-2}$, respectively, but the performance and availability of each technique varies by means of meteorological conditions and time of the day. Spaceborne observations of the IWV from MODIS generally showed a bias toward lower values, which most probably results from difficulties in the discrimination of clear and cloudy scenes from the satellite data. IWV observations are compared to ICON simulations with 156 m horizontal resolution. A case study reveals that the diurnal cycle of IWV variability of the model matches well with the high-temporal-

resolution microwave radiometer measurements, given a slight bias toward lower values in the model simulations, and that the spatial covariances for distances on the km scale are comparable in observations and model.

A technique that is considered to provide accurate, continuous, highly-resolved observations of the water vapour mixing ratio is the Raman lidar. Nevertheless, the stability of the system calibration is still subject of research and may depend on the design





of specific systems. Based on observations with the Raman polarization lidar Polly[XT] at supersite KRA and of BASIL at supersite JUE, Foth et al. (2015) presented a calibration technique that uses the integrated water vapour of a co-located microwave radiometer to provide calibration data for the lidar observations. The result is an automatically generated time-height cross section of the water vapour mixing ratio, as it is shown in Figure 8 for KRA for the April 2013 during HOPE-

Jülich. As can be seen, lidar observations are only available at nighttime and only from the ground to the base of optically thick clouds. In a sophisticated approach, these data gaps will in future be filled with values obtained from an optimal-estimation scheme that considers the spatio-temporal evolution of both the integrated water vapour from the microwave radiometer and the vertical profiles of water vapour mixing ratio (Foth et al., 2016). A similar methodology was also applied to the JUE BASIL and microwave radiometer data by Barrera-Verdejo et al. (2016) who could show the benefits of sensor

synergy in terms of an increase in information content in the regions where lidar data is not available. Barrera-Verdejo et al. (2016) similarly showed the positive impact of combining Rotational Raman Lidar measurements of BASIL with microwave radiometer observations for improving the temperature profile above the boundary layer.

A self-calibrating technique to observe the spatiotemporal distribution of water vapour is the DIAL technique. Such a system was operated during HOPE Jülich by IPM at supersite HAM. Exploiting the scanning capabilities of the DIAL system Späth

et al. (2016) presented detailed insights into the spatial inhomogeneity of the water vapour field around the supersites HAM, KRA, and JUE. Such observations provide valuable information that are required to improve our understanding of land–atmosphere exchange processes.

### 3.4  Microphysical properties of aerosols and clouds

The retrieval and evaluation of microphysical properties of aerosols, clouds, and precipitation from ground-based remote sensing observations is a crucial task. In-situ observations do provide much higher accuracy but for the long-term evaluation of the performance of operational weather forecast models and the microphysical parameterizations therein continuous datasets are required.

In particular the HOPE-Melpitz campaign provided the opportunity to relate in-situ observations of warm-cloud microphysical

properties and aerosol properties from ACTOS to the respective parameters observed with ground-based observations of the LACROS facility. Case studies are presented in the following that document the simultaneous ground-based and in-situ observation of a stratocumulus layer and the aerosol properties in the lower troposphere, respectively.

Aerosol particles act as nuclei for cloud droplets and ice crystals and are thus a prerequisite for the formation of clouds. Lidar is a promising tool to provide estimates of the concentration of cloud droplet condensation nuclei (CCN) and ice nucleating

particles (INP) (Mamouri and Ansmann, 2016). During HOPE-Jülich and HOPE-Melpitz the Raman polarization lidar Polly[XT] was continuously operated to provide information on the vertical aerosol structure in the planetary boundary layer and the troposphere. HOPE-Jülich was the first time a Raman polarization lidar provided a continuous data set of the calibrated attenuated backscatter coefficient at three wavelengths. Amongst other parameters, the dominating type of aerosol particles





present in each observed volume was derived by a newly developed target classification, as is explained by Baars et al. (2016b). Figure 9 exemplary shows the aerosol target classification for three consecutive days from 24 to 26 April 2013 (IOPs 6-8) during HOPE-Jülich. Frequently large, non-spherical particles, probably dust or pollen particles that were emitted in the vicinity of the site have been monitored. With increasing distance from ground, the particles grow by hygroscopic growth,

leading to the presence of large, spherical particles, as it was the case on 25 and 26 April. The mask also helps to identify whether a cloud layer was within or detached from the planetary boundary layer aerosol. Overall, the classification of cloud particles solely on the lidar observations is difficult. This will be overcome in a future step, by merging the multi-wavelength aerosol classification with the Cloudnet target classification presented in Illingworth et al. (2007).

Retrievals of microphysical aerosol properties, such as CCN concentration, from lidar observations, as well as retrievals of the

ambient scattering properties of an aerosol population measured in-situ are still subject to large uncertainties. In-situ observations of aerosol properties are usually performed under dry conditions and inlets are limited by a maximum cut-off size of an aerosol distribution. During HOPE-Melpitz, both in-situ aerosol observations as well as lidar observations of Polly$^{XT}$ were available. Figure 10 presents the relationship of the backscatter coefficient observed with Polly$^{XT}$ and the respective extinction coefficient obtained from the in-situ aerosol observations of ACTOS as derived by Düsing et al. (2016). Based on

the dry in-situ measurements of ACTOS, the ambient extinction coefficient was obtained at wavelengths of 355, 532, and 1064 nm using a Mie model and a hygroscopic-growth correction. A linear relationship with significant R² values was derived. This shows that the ambient aerosol extinction coefficient can well be derived from in-situ measurements given the extensive instrumentation for microphysical and chemical aerosol characterization that is available at the Melpitz field site.

During HOPE-Jülich the availability of CCN was investigated using an aerosol model. The approach presented by Hande et

al. (2016) used the COSMO-MUSCAT model to simulate the generation and transportation of aerosols over Germany during the campaign. From the simulation results, a parameterisation of the CCN concentration was derived which can be applied also to other climatological regions and different aerosol regimes. Even though the simulated aerosol properties were evaluated against in-situ observations of aerosol particle size distributions at Melpitz, no evaluation of the CCN parameterisation against measurements was performed. This emphasizes the need to improve remote-sensing techniques for the retrieval of CCN

profiles as the one of Mamouri and Ansmann (2016).

At the beginning of the first phase of HD(CP)² no operational microphysical retrieval of the effective radius of cloud droplets from ground-based remote sensing observations was available. As a first step towards an evaluation dataset for numerical weather forecasts, it was decided to apply the retrieval technique of Frisch et al. (2002) to the LACROS observations by implementing it into the processing framework of Cloudnet. The technique is based on vertically pointing measurements from

a millimetre-wavelength cloud radar and a microwave radiometer and produces height-resolved estimates of cloud particle effective radius and liquid water content. In addition, liquid water content profiles are produced operationally within Cloudnet (Illingworth et al., 2007), assuming either adiabatic profiles of liquid water content between the lidar-derived cloud base and the radar-derived cloud top or scaled-adiabatic profiles for which the adiabatic liquid water content is scaled to fit the liquid water path observed with the microwave radiometer.



The implemented Frisch-2002 retrieval of cloud droplet effective radius and the Cloudnet retrieval of the adiabatically scaled liquid water content were evaluated against in-situ observations of ACTOS for a stratocumulus deck observed simultaneously by ACTOS and LACROS on 22 September 2013 during the HOPE-Melpitz campaign (Seifert et al., 2016). The ACTOS flight on 22 September is denoted as IOP 22 in Table 3. The comparison of the average vertical profiles of liquid water content and

cloud droplet effective radius is shown in Figure 11. Although the vertical profile of LWC agrees well with the in-situ and the remote-sensing observations, the deviation of the effective radius profile increases towards the cloud top. The main reason for this deviation was found to be the assumption of a certain shape of the size distribution in the ground-based retrieval.

The accurate representation of the ice phase in numerical models is a crucial task since cold rain is the main driver of precipitation formation at midlatitudes (Mülmenstädt et al., 2015). The continuous observations of the LACROS supersite

during HOPE-Jülich enabled to obtain statistical information about the primary ice production in stratiform midlevel mixed-phase cloud layers. Figure 12 shows an overview about the ice water content and ice-to-total mass ratio of all mixed-phase cloud layers that were identified from the HOPE-Jülich observations with the scheme described in Bühl et al. (2016). The ice water content of clouds with top temperatures above -10 °C was in general lower than $10^{-4}$ g m$^{-3}$. At temperatures below -15 °C, values of the ice water content vary around $10^{-3}$ g m$^{-3}$. The ice-to-liquid mass ratio decreases from $10^{-2}$ to $10^{-5}$ for

temperatures increasing from -40 to 0 °C. This indicates that the liquid water content of the clouds increases with decreasing cloud top, which can be explained with the increasing amount of adiabatically available liquid water with increasing temperature. The color-coded data points in Figure 12 provide in addition the radar-observed linear depolarization ratio of the observed ice particles, which is a proxy for the particle shape. Values of around -20 dBZ (-10 °C <T<-5 °C), -30 dBZ (-20 °C <T<-10°C), and -25 dBZ (T<-20 °C) indicate needle-like, dendritic, and bullet-rosette-like shapes, respectively (Bühl et al.,

2016;Myagkov et al., 2016). Knowing about the relationship between ice water content, liquid water content, temperature, and shape of freshly formed ice crystals is an important step towards new approaches for the evaluation of ice formation schemes in numerical weather forecast models. This will also be a task of the second phase of HD(CP)².

### 3.5 Macrophysical cloud & precipitation properties

The combination of scanning polarimetric X-band Doppler rain radars, vertically pointing micro rain radars (MRR) and a ground-based network of disdrometers and rain gauges provided an excellent opportunity to validate the Doppler rain radar ability to infer the spatial variability of quantitative precipitation properties from polarimetric radar reflectivities. Xie et al. (2016) performed a detailed analysis of all precipitation observations under different synoptic conditions. As an example Figure 13 shows a time series of the surface rain rates estimated from three Doppler rain radar measurements compared to the

in-situ observations from seven disdrometers (partly from TR32 and TERENO projects), averaged over the disdrometer locations. In general, the rain radars reproduce the disdrometer observations well. The two near-by radars (KiXPol and JuXPol) showed better agreement than the 50 km remote radar BoXPol, which is explained by its correspondingly larger field of view





and associated beam-filling errors. Interestingly, daily accumulated precipitation from rain radars and disdrometers agree very well and independent of distance, which indicates that no systematic errors affect the conversion of observed radar reflectivity into near-surface precipitation properties. Xie et al. (2016) also managed to associate distinct microphysical processes for rain formation like coalescence, size-sorting and riming/aggregation with the state of the measured polarization state. These

polarimetric fingerprints serve as very useful information for process understanding of rain formation and model validations (Trömel and Simmer, 2012)

Ground-based cloud photography provides the most detailed qualitative information on cloud patterns at high spatial and temporal resolution. Consequently, up to six sky imagers were used during HOPE. The combination of several imagers allows also for a quantitative retrieval of the spatial cloud structure. Beekmans et al. (2016) presented an approach for a spatial cloud

reconstruction by using two hemispheric sky imagers in a stereoscopic setup. They combined a dense stereo correspondence technique and a large-scale stereo setup to derive 3D cloud geometries. Obviously, such stereoscopic cloud reconstruction is best suited for convective clouds that exhibit strong 3D spatial features. Important aspects of such a technique include an accurate camera calibration (internal projection and camera orientation in space), precise synchronization, similar radiometric properties, and successful stereo matching on the rather fuzzy (diffuse) cloud images. As an example, Figure 14 shows a cross

section (right panel) of a reconstruction from a cumulus cloud (left panel). It was found that the cloud base height is very well reproduced in comparison to lidar observations. In general, Beekmans et al. (2016) provided a complete approach including geometric and radiometric corrections to obtain the spatial cloud envelope geometry for the cloud sides facing the sky imagers. Together with 3D cloud information from scanning active systems such data will be very valuable for cloud reconstruction and radiation closure studies.

## 4 Examples of model-observation inter-comparisons

The above-mentioned studies demonstrate well that large efforts are being taken to make observations suitable for initialization and evaluation of numerical weather prediction (NWP) models and to provide process studies that are essential for their improvement. Nevertheless, operating a forecast model at scales that are small enough to resolve the different supersites of the

HOPE-Jülich campaign puts certain requirements on the capabilities of the model. When the model resolution is high enough to be in the range of the inertial scale of the turbulent eddies (see Section 3.2), the model is operating in the so-called "grey zone" where the parameterization of turbulence in the convective ABL may not be necessary. To what extent the parameterization of turbulence and shallow convection is still necessary is one of the key subjects of HD(CP)². Based on HOPE-Jülich observations, the grey zone was investigated in a study of Barthlott and Hoose (2015) who run model simulations

at horizontal resolutions between 250 m and 2.8 km. From the turbulence spectra derived from the model output, however, it was found that the effective resolution (the minimum size of resolvable eddies) is about 3-5 times lower than the resolution at which the model was initialized. For one case with isolated showers the amount of formed precipitation was found to vary by





48% for the different model resolutions. In fact, model runs with the highest resolution showed best agreement to the observed precipitation amounts.

A major goal of HD(CP)[2] is to use high-resolution modelling to derive parametrizations for climate models. In this respect the vertical cloud overlap parametrization is of high interest as it strongly influences the distribution of energy. In the past such parametrizations have only been tested against observations on a global scale or for deep convective clouds. For the first time, Corbetta et al. (2015) investigated cumuliform cloud overlap for several boundary layer cloud cases including HOPE and compared it with the results from LES runs. Gridded time-height data from Cloudnet were used to derive cloud fraction masks at various temporal and vertical resolutions. They could show good agreement between observations and models in the overlap ratio, i.e., ratio of cloud fraction by volume and by area, as a function of vertical resolution. Interestingly, the simulated and observed decorrelation lengths found for this type of clouds are smaller ($\sim$300 m) than previously reported ($>$1 km).

The evaluation of actual LES simulations of the HOPE-Jülich area was done by Heinze et al. (2016b) who performed simulations at up to 50 m horizontal resolution over the HOPE domain for a 19-day time period in order to capture a variety of different atmospheric and especially boundary layer conditions. The general weather pattern where reproduced in 80% of the cases. Also cloud types usually agree well with observations. Resulting turbulence characteristics and boundary layer heights have been compared to observations from active remote sensing (Doppler lidar and aerosol lidar) and from in-situ radiosonde observations as proposed by (Schween et al., 2014). Figure 15 demonstrates that the high resolution LES runs qualitatively reproduce the observed boundary layer heights within the observation uncertainties. Nevertheless, the differences are pointing to problems in the representation of ABL features in the LES simulations, and should be subject of further investigations. Please note that the criterion of model based ABL depth is also subject of uncertainties which are explained in Milovac et al. (2016) where similar deviations between measurements and observations were found. Heinze et al. (2016b) further showed that the turbulence characteristics of the ABL are captured satisfactorily with the LES model. Significant differences with respect to results from coarser resolved COSMO simulations were not reported. This might in part be due to the so-called semi-idealized set up with periodic boundary conditions and a homogeneous surface forcing. In general, the LES reproduce the observed surface fluxes when averaging over the HOPE domain.

Furthermore, within the Synthesis Module of HD(CP)[2] high-resolution ICON runs with 156-m resolution were extensively evaluated against datasets collected during HOPE-Jülich and from other sources (Heinze et al., 2016a). It was found that the high-resolved ICON-LES model much better matches the variability at small- to meso-scales than a coarser-resolved reference model. Simulated turbulence profiles are realistic, and the integrated water vapour matches the observed temporal variability at short timescales. From direct comparisons between modelled and continuous ground-based observations of the cloud field during HOPE-Jülich it was however found that convective boundary layer clouds are under-represented in the model, even though the evaluation of the cloud fields on a larger scale, i.e. in comparison to satellite observations, showed that clouds are well represented in the model. Heinze et al. (2016a) concluded that, despite the given potential for further improvement of the ICON-LES model, it already fits well to the purpose of using its output for parameterisation development.





## 5    Summary & conclusions

The HD(CP)$^2$ Observational Prototype Experiment HOPE provided an unprecedented data set on the spatiotemporal structure of surface and boundary layer energy fluxes, temperature, humidity, aerosols, clouds and precipitation fields along a variety of weather situations.

All data that have been measured by the official HD(CP)$^2$ partner institutes are stored in the HD(CP)$^2$ Data Archive Centre, and will be public available by the end of 2016. Currently, evaluation of the ICON model is performed both on small spatiotemporal scale based on the HOPE data and over the entire domain of Germany exploiting supersite, satellite and radar data. The extensive data base enable studies beyond pure model evaluation with a large potential for process studies on boundary layer fluxes, the formation of clouds and precipitation, cloud-aerosol interaction and on many more aspects.

With the large number of in-situ and Doppler wind lidar instruments coherent structures in the surface-near boundary layer wind fields and characteristic integral scales have been identified, and have been related to the type of external forcing. For the first time to our knowledge, TTRL demonstrated its capability to resolve the temperature inversion layer at the top of the ABL during daytime, which is a key information for future process studies. Similarly, vertical temperature fluctuations have been observed for the first time by means of rotational Raman lidar measurements. It turned out that a temporal resolution of

10 s was sufficient to resolve turbulence structures down to the inertial sub-range from the mixed layer to the entrainment zone. Observed statistics of vertically resolved temperature fluctuations up to the forth-order moment provide important information on boundary layer dynamics and thermodynamics. The combination of daytime temperature and humidity profiles from Raman lidar measurements was used to obtain turbulent flux profiles in the convective boundary layer. In general, the combination of vertically resolved (lidar) and vertically integrated (microwave radiometer) and in-situ (radiosondes)

measurements of the atmospheric humidity has produced a unique 3D humidity field that together with wind and temperature measurements will serve as a solid constrain for the evaluation of high-resolution models. These results confirm the unique importance of high-resolution thermodynamic profiles for weather and climate research as demonstrated in Wulfmeyer et al. (2015). Surface solar and thermal radiation budget measurements complement the energy budget observations. A high-resolution pyranometer network produced statistics on spatiotemporal solar irradiance correlations for different sky conditions.

A comparison of turbulence measurements near cloud top from aircraft in situ measurements and from cloud base by lidar measurements revealed similar statistical properties, which points to a vertically homogeneous turbulence structure inside stratocumulus clouds.

Continuous operations of most of the instruments for two months made it possible to identify atmospheric variability from the micro- to the mesoscale. A long-term comparison of integrated water vapour from radiosondes and from ground-based and

satellite remote sensing shows a generally good agreement but also revealed a bias of the spaceborne measurements towards lower values.  Lidar observations of the aerosol profiles have been translated into the dominant aerosol type within each measurement volume. Such aerosol target classifications showed the hygroscopic growth of spherical aerosol particles under humid conditions as well as the presence of large non-spherical dust particles that were emitted from near-by sources. It turned





out that the closure of in-situ observations and remote sensing of aerosol microphysical properties is feasible when an extensive aerosol in-situ characterisation is available. A respective closure of cloud microphysical properties remains challenging due to uncertainties stemming from required assumptions on the particle size distribution and from spatiotemporal averaging. Cloud liquid water content profiles derived in-situ and with remote sensing, however, were found to agree well. Continuous

observations of mixed phase clouds from a combination of active and passive remote sensing shows that the ratio of ice to liquid water decreases with decreasing cloud top temperature, which serves as an important information for the evaluation of ice formation parameterizations in cloud modelling.

Macrophysical cloud structures like cloud vertical dimension, cloud cover, cloud type, precipitation fields have been continuously observed with lidar, radar and sky imager. Large-scale precipitation pattern together with the dominant process

type for rain formation were observed with polarimetric Doppler precipitation radars. Three-dimensional cloud morphology has been retrieved from sky imagers in a stereoscopic setup. Thus, a uniquely high resolved data set on cloud structural properties has been achieved during HOPE.

First model evaluation studies based on HOPE data have shown general agreement between observed and modelled boundary layer height and turbulence characteristics, but also point to significant differences that deserve further investigations.

With the completion of the high-resolution ICON LES model a vast number of model evaluation work is currently in progress. The supersite LACROS that has been deployed at KRA during HOPE-Jülich continues its long-term measurements at its base institute in Leipzig and will contribute to further process and model evaluation studies.

Future work will take advantage of the synergy of the different active and passive remote sensing measurements. For instance, Doppler lidar and polarimetric radar measurements may link dynamical forcing (up and downdrafts) with microphysical

processes (riming, coagulation, ice formation). The cloud radars of JOYCE, KITcube and LACROS were occasionally operating in a synchronized scan mode. Together with vertically pointing and scanning microwave radiometer data three-dimensional distributions of cloud liquid water may be constructed, and may get even further refined from cloud structure stereoscopy from synchronized sky imager data. Radiation closure studies will be performed based on observed and modelled spatial cloud structures and observed surface radiation budget measurements. High-resolution irradiance data can be used to

build stochastic irradiance simulators for specific cloudy sky conditions, which in turn can be used to construct realistic cloud induced solar radiation variability. Combined measurements of temperature, humidity, and vertical wind fluctuations in the PBL under different meteorological conditions will provide important statistical information for improved turbulence parameterizations. HOPE also demonstrated the future potential of the synergy of scanning wind, temperature, and water-vapour lidar systems for 3D studies of land-atmosphere exchange and ABL entrainment in heterogeneous terrain. HOPE data

may also reveal to what extend variations in aerosol concentrations and thus in CCN and IN concentrations have an effect on cloud and ice formation compared to dynamical forcing.



**Acknowledgements**

The work summarized in this review was mainly carried out in the project HD(CP)$^2$ funded by the German Ministry for Education and Research. We specifically acknowledge the HD(CP)$^2$ projects 01LK1212A (University of Hohenheim), 01LK1209D (University of Leipzig), 01LK1209B (FZJ), 01LK1209C and 01LK1212C (TROPOS), 01LK1212F and 01LK1204B (KIT), 01LK1219A and 01LK1210A (University of Bonn), 01LK1203B (University of Hannover), 01LK1203A (MPI Hamburg). We also refer to all acknowledgements in the publications cited in section 3.

HOPE is particularly grateful to the Research Center Jülich that provided generous logistic support during the Jülich campaign. We thank the Transregional Collaborative Research Centre 32 "Patterns in Soil-Vegetation-Atmosphere Systems - Monitoring, Modelling and Data Assimilation" for contributing their valuable rain observation research infrastructures to the Jülich campaign.

The universities of Cologne and Bonn as well as TROPOS secured intense radiosonding observations from internal budgets. Raman lidar system BASIL were funded on the basis of a specific cooperation agreement between Scuola di Ingegneria - Università degli Studi della Basilicata, TROPOS and MPI Hamburg.

We appreciated the provision of four Sun photometers for HOPE-Jülich and one device for HOPE-Melpitz by Goddard Space Flight Center, Greenbelt, MD, USA.

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



**Figures and Tables**

**Table 1: Sites and networks deployed during HOPE-Jülich. Information on the individual instruments are given in Table 2. For details on the affiliations see Sec. 2.1.1. as well as the title page of this article.**

| Supersite or network | Abbrev-iation | Location | Instruments |
|---|---|---|---|
| Krauthausen | KRA | 50.8797° N, 6.4145° E, 99 m asl | **TROPOS:** LACROS supersite with Mira-35, PollyXT, CHM15kx, WiLi, HATPRO, Parsivel2, Pyranometer, all-sky imager<br>**MPIM:** Cimel |
| Jülich | JUE | 50.909° N, 6.4139° E, 111 m asl | **IGMK/FZJ**: JOYCE with Mira-35, CHM15k, HALO Streamline, HATPRO, Parsivel2, all-sky imager<br>**MPIM:** ARL-2, Cimel<br>**UniBas:** BASIL<br>**TROPOS:** Pyranometer |
| Hambach | HAM | 50.897° N, 6.463° E, 114 m asl | **KIT:** KITCube with Mira-35, Wind-Tracer, HALO Streamline, CHM15k, HATPRO, radiosonde station, Parsivel2, energy balance stations (at TERENO sites, see Fig. 1), wind mast<br>**IPM:** DIAL, TRRL |
| Pyranometer network | PYR | | Area enclosed by 50.846° N, 6.379° E and 50.945° N, 6.485° E.<br>All pyranometers operated by TROPOS. |
| Sky imager network | SKY | | **KRA:** 50.897° N, 6.463° E, 99 m asl<br>**JUE:** two instruments within 500 m of 50.909° N, 6.4139° E, 111 m asl |
| X-band radar network | XRD | | **KIT:** KiXPol at 50.8566° N, 6.3799° E, 114 m asl<br>**MIUB:** BoXPol at 50.7312° N, 7.071 24° E, 99.5 m asl<br>**FZJ:** JuXPol at 50.932° N, 6.455° E, 300 m asl<br>All Instruments operated by the individual institutions. |
| Sun photometer network | SUN | | **Aachen:** 50.777° N, 6.0606° E, 230 m asl<br>**KRA:** 50.879° N, 6.4145° E, 99 m asl<br>**Hombroich:** 51.151° N, 6.6436° E, 70 m asl<br>**HAM:** 50.897° N, 6.4630° E, 114 m asl<br>**JUE:** 50.909° N, 6.4139° E, 111 m asl<br>All instruments, except for JUE, provided by NASA/GSFC and operated by MPIM. |





**Table 2: Details of instruments deployed during HOPE-Jülich and (in part, see Sec. 2.1.2) during HOPE-Melpitz.**

| Instrument | type | reference | sites / networks | measured quantities | atmospheric parameters | resolution |
|---|---|---|---|---|---|---|
| **Lidar remote sensing** | | | | | | |
| Polly[XI] | multiwavelength Raman polarization lidar | Engelmann et al. (2016) | KRA | backscattered signal from molecules and particles | particle backscatter coefficient and extinction coefficient; linear depolarization ratio; water vapor mixing ratio | 30 m; 30 s |
| BASIL | multiwavelength Raman polarization lidar | Di Girolamo et al. (2009) | JUE | backscattered signal from molecules and particles | profiles of particle backscatter coefficient and extinction coefficient, linear depolarization ratio, water vapour mixing ratio, temperature | 7.5 m; 10 s |
| ARL-2 | multiwavelength Raman polarization lidar | Wandinger et al. (2016) | JUE | backscattered signal from molecules and particles | profiles of particle backscatter coefficient and extinction coefficient, linear depolarization ratio, water vapour mixing ratio, temperature | 7.5 m; 10 s |
| DIAL | differential absorption lidar | (Späth et al., 2016;Wagner et al., 2013) | HAM | backscattered signal from molecules and particles | absolute humidity 3D fields, particle backscatter 3D fields at 820 nm | 15 m; 1 s |
| RRL | Rotational Raman temperature lidar | (Hammann et al., 2015;Radlach et al., 2008) | HAM | backscattered signal from molecules and particles | 3D fields of temperature, water vapor mixing ratio, particle backscatter coefficient at 355 nm, particle extinction coefficient at 355 nm | 3.75 m, 10 s |
| CHM15k(x) | lidar ceilometer | Heese et al. (2010) | KRA, HAM, JUE | backscattered signal from molecules and particles | cloud boundaries | 10-30 s; 15 m |
| WiLi | Doppler lidar | Engelmann et al. (2008) | KRA | Doppler-shift along line-of-sight | profiles of vertical air velocity and horizontal wind | 1-2 s; 75 m |
| HALO Streamline | Doppler lidar | Pearson et al. (2009) | HAM, JUE | Doppler-shift along line-of-sight | profiles of vertical air velocity and horizontal wind | 1-2 s; 15 m |
| WindTracer | Doppler lidar | Gatt et al. (2015) | HAM | Doppler-shift along line-of-sight | Doppler-shift along line of sight, SNR, vertical air velocity, radial air velocity, profiles of horizontal and vertical wind velocity | 0.1/1 s (radial / vertical); 25-70 m |
| Windcube | Doppler lidar | Gottschall and Courtney (2010) | HAM | Doppler-shift along line-of-sight | Doppler-shift along line of sight, SNR, vertical air velocity, radial air velocity, profiles of horizontal and vertical wind velocity | 1.6 s; 25 m |
| **Radar remote sensing** | | | | | | |
| Mira-35, Mira-36S | 35/36-GHz cloud radar | Görsdorf et al. (2015) | KRA,HAM, JUE | Radar reflectivity, Doppler spectrum, linear depolarization ratio | Cloud boundaries, cloud structure, contributes to cloud liquid water profiles | 15-30 m; 1-30 s |
| X-band radar | 10-GHz precipitation radar | (Borowska et al., 2011;Kalthoff et al., 2013) | XRD | reflectivity, differential reflectivity, diff. phase, Doppler vel. and width, correlation coeff. | Horizontal precipitation and boundary layer wind field | 1 min, 100 m |
| **Passive remote sensing** | | | | | | |
| HATPRO | Microwave radiometer | Rose et al. (2005) | KRA, HAM, JUE | Atmospheric brightness temperatures from 22 to 58 GHz | Temperature and humidity profile; liquid water path | 1 s, 100-1000 m |
| CIMEL CE318 | Sun photometer | Holben et al. (2001) | SUN | sky radiances | aeorsol optical depth and volume size distribution, integrated water vapor | 15 min |
| All-sky imager / Mobotix S14 | Fisheye camera | (Beekmans et al., 2016;Kalisch and Macke, 2008) | SKY | Full sky images | Cloud morphology | 15 s, 120 s (S14) |



**Table 2, continued.**

| Instrument | type | reference | sites / networks | measured quantities | atmospheric parameters | resolution |
|---|---|---|---|---|---|---|
| **Ground-based and in-situ observations** | | | | | | |
| Parsivel2 | optical disdrometer | Tokay et al. (2014) | Bonn, HAM, KRA | Size and velocity distribution of hydrometeors | precipitation rate, rain drop size distribution | 30 s (Bonn, KRA), 60 s (HAM) |
| DFM-09 | radiosonde | Bock et al. (2016) | HAM | pressure, humidity, temperature, gps position | atmospheric pressure, temperature, humidity, wind vector | 1 s |
| Pyranometer | | Madhavan et al. (2016b) | PYR | Photodiode voltage, bimetal voltage | Broadband solar and thermal downward radiation fluxes, temperature | 1 km, 1 s |
| Surface meteorology | energy balance stations and masts | Kalthoff et al. (2013) | HAM, JUE | temperature, humidity, pressure, wind vector, precipitation rate, radiation | Surface and soil latent and sensible heat flux | 0.05 s (turbulent fields), 1 s (met. data), 30 min (fluxes) |



**Table 3: Summary of Intensive Observation Periods during HOPE-Jülich. Coloured lines denote "Golden Days".**

| IOP No. | Date | sky situation |
|---|---|---|
| 1 | April 13 | broken convective clouds |
| 2 | April 14 | low-cloud deck until noon, broken cirrus in the afternoon |
| 3 | April 15 | convective clouds, precipitation |
| 4 | April 18 | few PBL clouds, broken cirrus |
| 5 | April 20 | clear |
| 6 | April 24 | clear |
| 7 | April 25 | PBL clouds |
| 8 | April 26 | frontal clouds, precipitation |
| 9 | April 29 | weak convection |
| 10 | May 2 | strong aerosol, cumulus |
| 11 | May 4 | clear |
| 12 | May 5 | PBL clouds |
| 13 | May 18 | scattered clouds |
| 14 | May 19 | scattered clouds |
| 15 | May 24 | PBL convection in cold air mass |
| 16 | May 25 | convective clouds, warm-front and precipitation in the evening |
| 17 | May 27 | scattered clouds |
| 18 | May 28 | scattered clouds, complex scenario |

**Table 4: Summary of Intensive Observation Periods during HOPE-Melpitz. On these days a total of 15 hours of observations with ACTOS were performed.**

| IOP No. | Date | sky situation |
|---|---|---|
| 19 | September 13 | Cu clouds |
| 20 | September 14 | polluted air, clear skies, Cu |
| 21 | September 17 | clean air, Cu |
| 22 | September 22 | Sc, decoupled from PBL |
| 23 | September 27 | Cu convection, very low PBL |

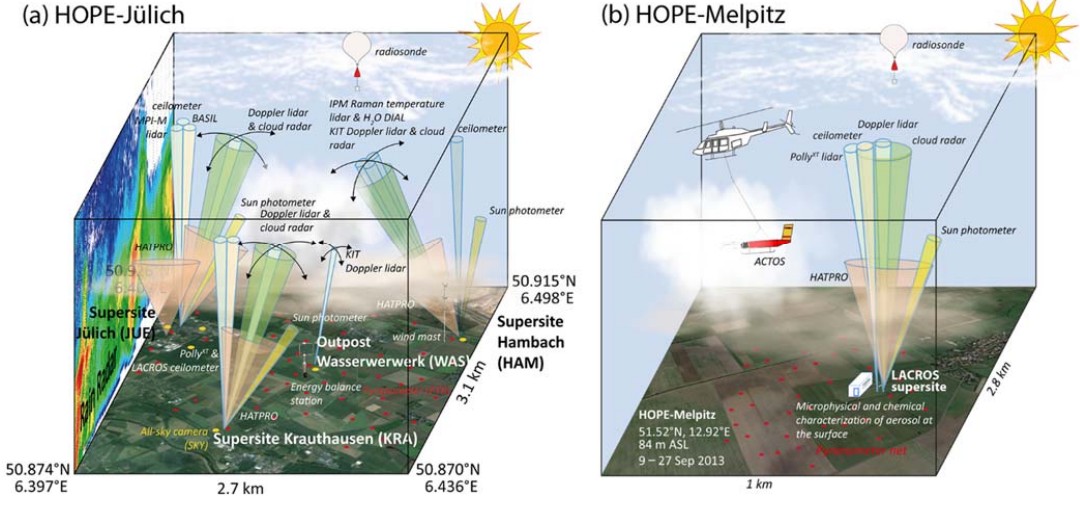

**Figure 1: Setup of the HOPE campaign: (a) Location of the three supersites Jülich (JUE), Hambach (HAM), and Krauthausen (KRA) as well as the outpost Wasserwerk (WAS) with their main instrumentation during HOPE-Jülich. (b) Location of the instrumentation during HOPE-Melpitz.**





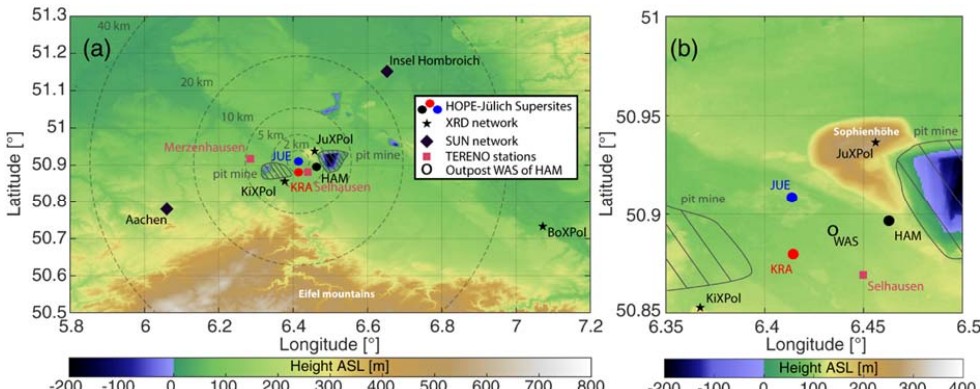

**Figure 2: Map of the spatial distribution of the sites and networks deployed according to Table 1 (left) and a zoomed-in view centered at supersite Jülich (right). Background colors indicate the topography and dashed lines denote circles of constant distance from supersite Jülich (JUE). Dashed lines denote circles of constant distance from supersite JUE. Shaded areas denote open-pit mines, for which the elevation map is not up to date.**

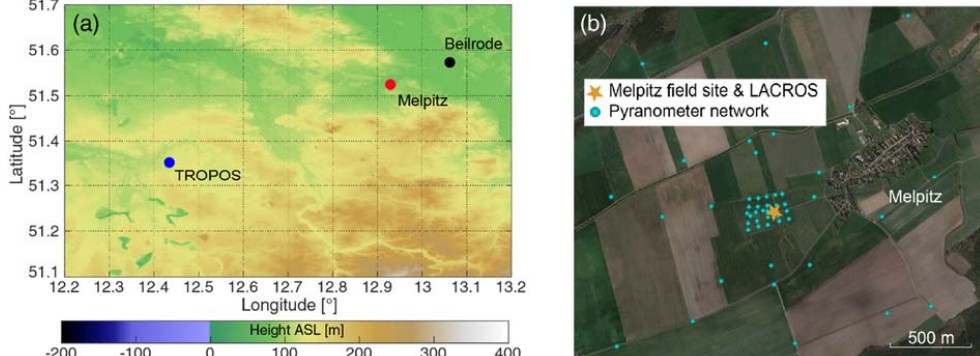

**Figure 3: Topography around the location of the HOPE-Melpitz campaign. (a) large-scale topography; (b) aerial photograph of the Melpitz field site with the locations of the pyranometers.**





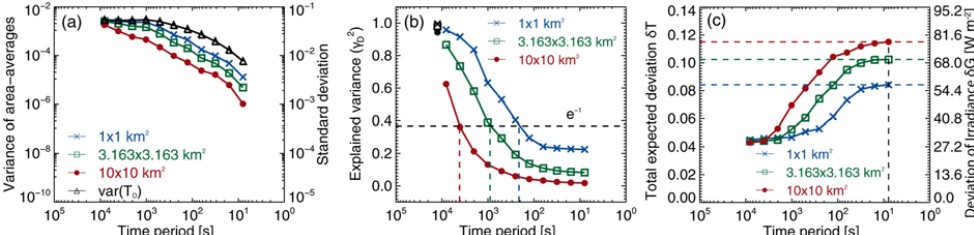

**Figure 4: Spatiotemporal characteristics derived from the pyranometer network under broken-cloud conditions during HOPE-Jülich. This figure illustrates the origin of deviations between a point measurement and a domain-averaged value (representativeness error) for broadband solar atmospheric transmittance and irradiance for different domain sizes. (a) Power spectra of transmittance for a point measurement and domains with different sizes; (b) Explained variance of temporal fluctuations in a point measurement and a domain average as function of period, and (c) total expected deviation between a point measurement and a domain average for transmittance and irradiance as a function of averaging , assuming a value of 680 W m$^{-2}$ for the incoming solar irradiance at the top of atmosphere. Adapted from (Madhavan et al., 2016a).**

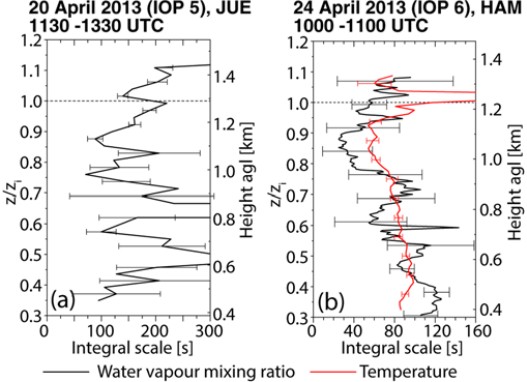

**Figure 5: Integral scales of the temperature fluctuations (black) and humidity fluctuations (red) in the convective boundary layer derived from high-resolved temperature observations obtained between 1130 and 1330 on 20 April 2013 (IOP 5) and 1100 and 1200 UTC on 24 April 2013 (IOP 6) during HOPE-Jülich. Heights are normalized with respect to the height of the convective boundary layer height $z_i$. Adapted from Behrendt et al. (2015), Muppa et al. (2016), and from Di Girolamo et al. [2016].**



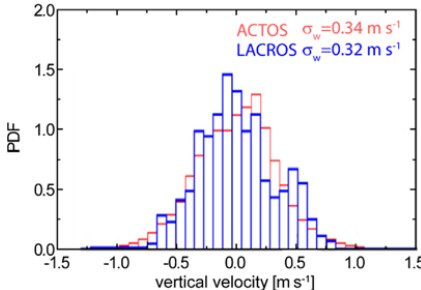

**Figure 6: Simultaneous observation of the vertical velocity in a stratocumulus layer performed in-cloud with ACTOS (red) and at cloud base with Doppler lidar (blue) on 22 September 2013 during HOPE-Melpitz. Adapted from Seifert et al. (2016).**

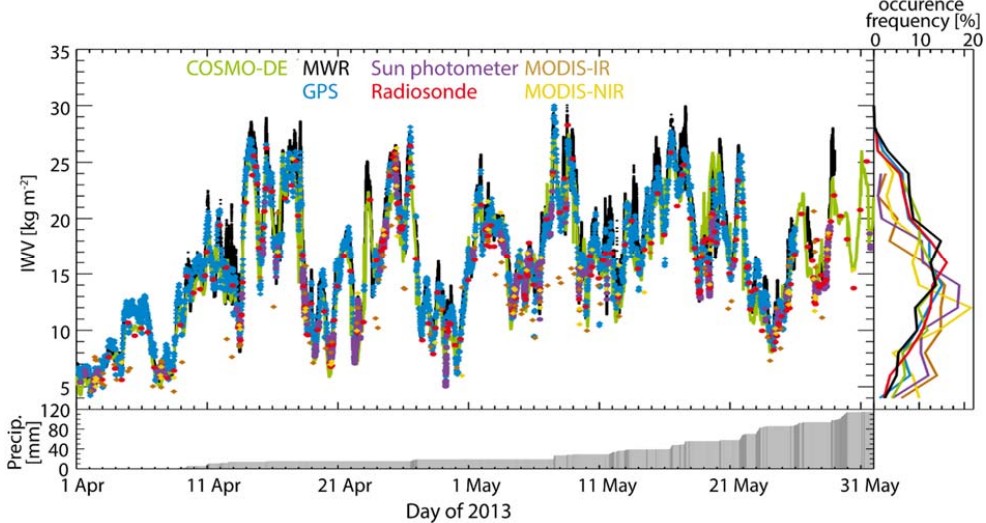

**Figure 7: Observation of the integrated water vapour (IWV) during HOPE-Jülich for a large suite of different instruments. Adapted from Steinke et al. (2015).**




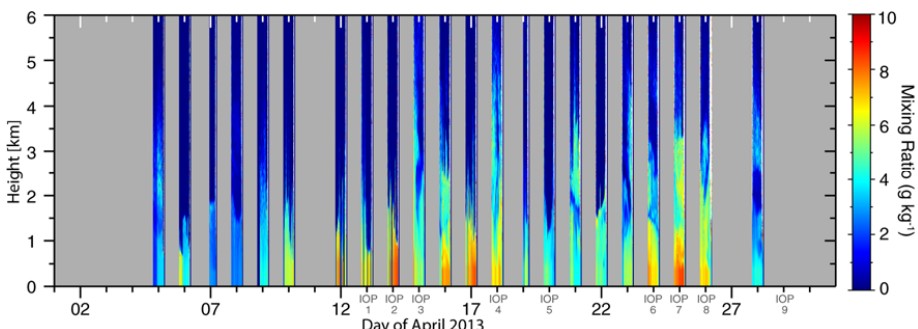

**Figure 8: Calibrated nighttime observations at KRA of the water vapour mixing ratio for April 2013 during HOPE-Jülich obtained from Polly$^{XT}$ data that were calibrated automatically with the integrated water vapour provided by a co-located microwave radiometer. Adapted from Foth et al. (2015).**

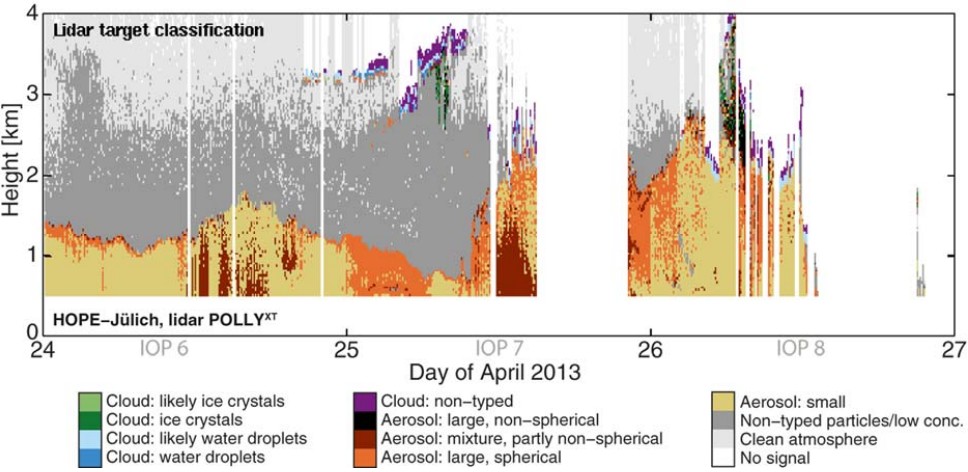

**Figure 9: Aerosol target classification for the HOPE-Jülich period from 24 to 26 April 2013 (IOPs 6-8) based on continuous observations of the multi-wavelength polarization lidar Polly$^{XT}$. The methodology is described in Baars et al. (2016b).**





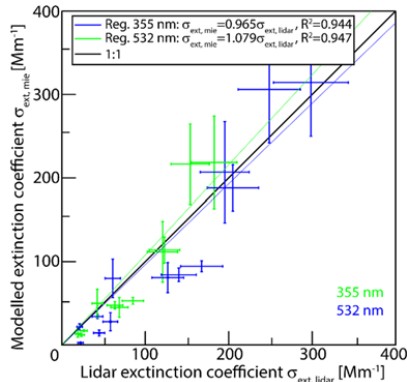

**Figure 10: Correlation between the particle extinction coefficient derived from Mie modelling and hygroscopic-growth correction of in-situ measurements of ACTOS with the respective ones measured with Polly^XT. Adapted from Düsing et al. (2016).**

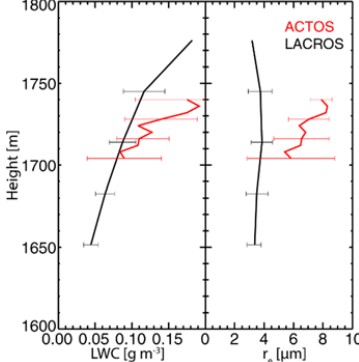

**Figure 11: Stratocumulus observation on 22 September 2013. Vertical profiles of (a) liquid water content and (b) cloud droplet effective radius as observed with ACTOS (red) and retrieved from LACROS observations (black). Error bars show the standard deviation. Adapted from Seifert et al. (2016).**



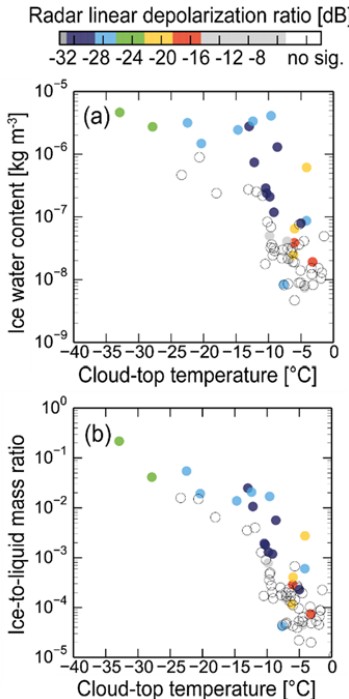

**Figure 12: Relationship between mean ice water content (IWC) and ice-to-liquid mass ratio as a function of cloud top temperature of all thin supercooled midlevel clouds detected during HOPE-Jülich.**

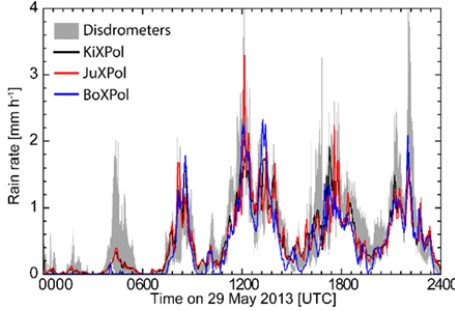

**Figure 13: Time series of rain rates derived from observations of seven disdrometers (including those from the TR32 program) and the three polarimetric radars on 29 May 2013. The shaded grey area indicates the range of rain rates observed by the disdrometers with 1 min temporal resolution in the HOPE area, while the rain rate from the three polarimetric radar observations is calculated at the radar gates that are coincident with disdrometer locations and also averaged over the disdrometer locations. From Xie et al. (2016).**


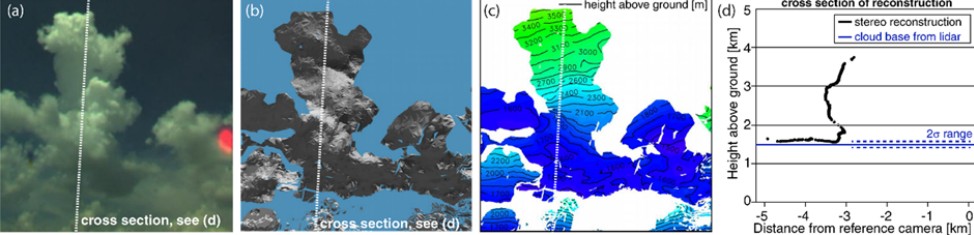

**Figure 14: 3D reconstruction of a cumulus tower from a stereographic photograph from 24 July 2014, 11:32:00 UTC. Shown are (a) subsection of the image obtained from the reference camera, (b) the reconstruction as an untextured triangulated surface mesh, (c) the color-coded height of the reconstruction with contour lines, and (d) reconstructed distance of the cloud edges from the reference camera obtained along the cross-section (dashed line) shown in (a), (b), and (c) as well as a comparison of the cloud base with the one observed with lidar ceilometer (blue line). Adapted from Beekmans et al. (2016).**

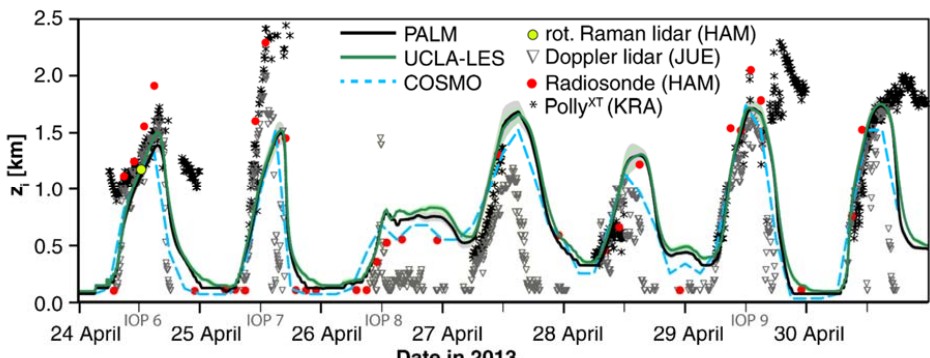

**Figure 15: Temporal evolution of the boundary layer depth $z_i$ for the period from 24 to 30 April 2013. $z_i$ is determined by means of the bulk-Richardson number criterion in all three models (PALM, UCLA-LES and COSMO) and in the radiosonde data. A criterion based on the vertical velocity variance and detected aerosol layers is used for the wind lidar and aerosol lidar PollyXT, respectively. The data point obtained from the temperature rotational Raman lidar (TRRL, rot. Raman lidar) is based on Behrendt et al. (2015). Radiosondes were launched at the KITcube site, the Doppler lidar and PollyXT took measurements at sites JUE and KRA, respectively. Grey and green shading denote twice the standard deviation of $z_i$ in PALM and UCLA-LES, respectively. Adopted from Heinze et al. (2016).**