# Peer review of "The HD(CP)2 Observational Prototype Experiment HOPE – An Overview"

_Atmospheric Chemistry and Physics, 2016_

## Referee Comment (RC1) · Anonymous Referee #1 · 23 Dec 2016

General Comments

(Note, I have referenced sections in the paper using Page (P) and Line (L) position throughout this review).

This article provides an overview of the HD(CP)2 Observational Prototype Experiment (HOPE). This paper does not appear to present new work but rather, provides an extensive summary of work that has been done using the HOPE experiment to date coupled with the experiment description. Given the complexity and breadth of HOPE, there is definitely value in an overview like this to help the science community understand the experiment and to make good use of the data it produced. Generally, I like the structure with a description of the physical layout of the campaign followed by science results and application to model evaluation. I have a few relatively minor concerns.

However, I think there are certain aspects of the paper that could be improved to make it more useful.

The two most significant issues I have with the paper are regarding the motivation and some of the description in section 2. At a high level, the authors have explained the goal of HOPE and what measurements will be provided. However, ultimately, there is little motivation provided beyond stating that the measurements from HOPE will be used for the initialization and evaluation of the ICON model. It would have been helpful, for example, to hear if there were particular areas of concern – or types of numerical experiments that were planned for this coupled observation/model system. Also, while I understand that this is an observation paper, not a modeling paper, I would have expected a bit more with regard how these data are intended to be used for model evaluation. At least pointers to model planning. In the introduction (P3,L1) it is stated that "Key to this effort was the provision of modeled scenarios at 100 m grid resolution over thousands of kilometers, . . ." But I do not subsequently find any mention of those modeled scenarios or work being done toward their development. There is a modeling section (4) but it is mainly a list of a few evaluation studies. It is not even clear if all of these studies make use of the ICON model which the introduction indicates is the target for this work. ICON is mentioned toward the end of section 4 (P17,L25) but it is not clear if the other studies mentioned earlier in that section are using ICON or some other more established model(s). In short, I don't get a sense of an overall set of goals either for the science applications or for linking the observations to models.

My other main issue is with the description of the measurements in section 2. Throughout, the authors are trying to present a lot of material and I appreciate that. As noted earlier, I think this will provide a valuable reference. And for the most part, I think the structure works well for the discussion of Results (section 3). But in section 2, particularly the discussion of HOPE-Julich in section 2.1.1, I was struck by the very long lists of instruments presented in text without any sub-organization. I think this section would be easier to read and to use if it were organized in some way – either with subheadings for sites – or sub-headings for major classes of instruments. As it is – it is difficult to take away a clear picture of how the instruments are coordinated.

I would also say that similar to my earlier comments about motivation for science and the modeling – more could be done here to explain why this particular set of measurements was chosen. It is stated at a high-level that there is a goal "to derive the atmospheric state of water vapor, temperature, wind and cloud and precipitation properties with 100-m resolution for an area of about 10x10x10 km . . ." (P4,L6-7). The list of instruments is impressive and certainly it provides some spatial representation of these parameters. However, I am sure that these instruments cannot provide all of those parameters over the full domain. It would be helpful to understand how close to achieving this goal HOPE came, how instrument configurations were chosen to get as close to this goal as possible, and if there are any thoughts regarding how to get closer to the goal of full sampling of the stated domain. Why was this particular set of instruments chosen – and how do they enable achieving core goals of the activity?

Finally, before I move on to mentioning more specific issues, I suggest it would be helpful to comment somewhere in the Summary and conclusions if there are any specific plans for using the observation data set or particularly, for applying the data set to the ICON model.

More specific science issues

P1,L38: what does it mean to operate the sun photometers in synergy? Or does the operation in synergy refer to all the previously mentioned instruments (lidars, radars etc)? Either way – does this just refer to the spatial placement of instruments? Or are instruments being scanned in such a way to optimize their co-collection of observations?

P2,L16: The text says "It is a coordinated initiative . . .". This is confusing because the previous sentence was specifically talking about the HD(CP)2 model – but I presume the sentence beginning on line 16 is referring to the larger HD(CP)2 initiative – not

specifically the model. In any case, I found the sentence to be confusing.

P2,L29-32 – the newly developed ICON model is mentioned. Is this the same as the HD(CP)2 model mentioned on line 15 of this page? Not clear.

P2,L30-31 – mentions that the observation datasets are intended to provide both initialization and evaluation of the ICON model. This is mentioned again on P16,L22-23. I can see that these data would be excellent for model evaluation – and a few example are given in section 4 (p16-17); however, I don't see any discussion of using these data for model initialization or how that might be done. I could imagine that these data could be used for that as well – but given that the point is made that these data are ideal for that purpose – it would be preferable to have some discussion on what the authors have in mind for that application – or what others have done or are planning in this area.

P5-6 – in this portion of section 2, the instrumentation of HOPE-Julich are described. I think this section really needs some sub-sections. This could be done by supersite or by instrument class or both. But as it is – this section comes across as a very long list of instruments that is very difficult to digest. Some structure would help get clear how these instruments are arranged.

P9, L2-3 – the HD(CP)2 data archive center is mentioned. I think it would be good to provide a link to this center. I think I was able to find it – but it would be helpful to provide the actual link here.

P9,L8 – indicates that an "essential regime" observed during HOPE was the turbulent structure of the atmosphere. I don't think I have ever seen the word "regime" used this way before. I think of "regime" as referring to a meteorological state – not the general distribution of a physical attribute.

P15,L5: The text indicates that the "LWC agrees well with the in-situ and remote sensing observations". First of all, this should be reworded along the lines of "the LWC de-

rived from remote sensing observations agree well with in-situ measurements". However, I also have some concerns about this statement. The phrase "agrees well" is generally subjective, but in this case, I would argue that the two are diverging near cloud top and that deserves some mention.

P16,L26-30. The text refers to activities related to the grey zone and characterizes this as conditions "where the parameterization of turbulence in the convective ABL may not be necessary". I think this is not a good representation of the issues posed by the grey zone. The issue as I understand it – is that in this spatial resolution zone (∼1km – ∼10 or 20 km) the resolution is too coarse to explicitly resolve certain features (e.g. eddies associated with shallow convection) but the resolution is too fine for traditional parameterizations to work. So I don't think the issue is so much that the parameterizations aren't necessary is that traditional parameterizations break down – because the assumption that the domain is much larger than the phenomena being sampled is no longer true.

P17,L16-17. The text states that the LES simulations "qualitatively reproduce the observed boundary layer heights within the observation uncertainties". This is not obvious. First of all – saying they agree within the uncertainties is a quantitative statement. And it appears that while they do agree well at sometimes, at others they clearly diverge. So – indicating that they agree seems like a simplification of what is going on.

Typographic/syntax

P3,L15: builts should be "builds"

P11, L29: the wording "that are highly resolved" is awkward or incorrect. I suggest changing this to something like "that are more highly resolved"

P11, L32: I think "lays the ground for . . ." should be "lays the groundwork for . . ."

P12, L6: I think that "it infers that Doppler lidars. . ." should be "it implies that Doppler lidars . . ."

P14,L2: I think the wording "Figure 9 exemplary shows the aerosol. . ." should be re-worded/reordered as "Figure 9 shows an exemplary aerosol . . ."

P16,L4: "state of the measured polarization state" seems redundant. I would think you could just say "measure

---

## Referee Comment (RC2) · Anonymous Referee #2 · 23 Dec 2016

**Review of Macke et al., ACPD, 2016**

This paper presents an overview of the HD(CP)[2] experiment and summaries of the associated papers that report on results from the campaign. It provides a useful overview of the available data and model runs that were performed. The paper is suitable for publication provided that a few corrections are made. In places the summaries of the papers do not give enough detail about the work being talked about (see line-by-line comments below). Some suggestions for additions are a table of the model runs performed detailing type of model, resolution, etc. Also, it would be useful to quote the uncertainties of the instruments listed in Table 1. There are a lot of statements along the lines of "xx agrees well with yy", but little quoting of quantitative agreement (e.g. within x %).

**Line-by-line comments**

Abstract

HOPE-Jülich instrumentation included a radio sounding station, 4 Doppler lidars, 4 Raman lidars (3, 3, and 4 of these provide temperature, water vapor, and particle backscatter data, respectively),

**The "3, 3 and 4" part of this sentence doesn't make sense to me since there is only one radio sounding station mentioned.**

p.2, L15

The newly developed convection-resolving HD(CP)2 model will be used to develop new convection parameterizations for large-scale eddy simulation models.

**The term "large-scale eddy simulation models" could lead to confusion with "large eddy simulation" models, or is this what you mean? Although, in that case might it not also be used to develop parameterizations for coarser resolution models? If you mean lower resolution models then it would be better to describe the types of model that you are talking about in a different way - e.g. mesoscale models, GCMS, or what the resolution range is perhaps. Or maybe this sentence is not even needed here since there is perhaps better explained at the start of p.3.**

p.3, L26

(Buehler and Russchenberg, 2016)

**Can't find this reference in the references.**

p.5, L27

These instruments were complemented by a microwave radiometer which determines temperature and humidity profiles,

**Does it also give the cloud liquid water path? This would be worth mentioning if so.**

p.5, L29

an X-band rain radar was operated

**Since descriptions of what observations the other instruments can give are listed here it would be good to do the same for the X-band radar.**

p.7, L7

The follow-up campaign HOPE-Melpitz has become necessary…
**Should be "The follow-up campaign HOPE-Melpitz became necessary…"**

p.10, L14
for 3 hourly and second-resolution observations, respectively
**Please make it clear whether you mean 1-second or 3-second resolution for the latter part of the sentence.**

p. 10, L20
"and showed that these increments are more strongly averaged out in space than the transmissivities themselves"
**It's not quite clear what you mean by "more strongly averaged out".**

p.11, L18
"the integral scales"
**It would be useful to refer back to section 3.1 where "integral scales" are explained.**

p.11, L3 – "resolution" would be better than "scale" here

p.11, L22 – "A general feature… decreases from the ground towards the top"
**This does not seem to be the case for Fig. 5a, only 5b. Can more detail be given about how common this was – e.g. in x % of the periods observed (since only two periods are shown in Fig. 5).**

p.12, L3
The inter-comparison shown in Figure 6 presents a sequence and a histogram of the vertical velocities observed with ACTOS (red) and WiLi (blue). Thus, vertical velocities in the stratocumulus are similar at the cloud base (observed with the Doppler lidar) and cloud top (observed with ACTOS).
**There is only a histogram (no sequence as mentioned). Also, it would be good to do a statistical test to determine how similar the two distributions are. Also, what are the means values for each distribution?**

p.13, L 13-17
**Can more detail be given about the type of measurement this is? I.e. is it a profile, or a single surface measurement?**

p.14, L3-5 –
**How is it known that the large aerosol were emitted in the vicinity of the supersite? Also, how is it known that particle growth was observed rather than large aerosol being advected in? A little more detail of the evidence is required here.**

p.15, L4 – "Although the vertical profile of LWC agrees well with the in-situ and the remote-sensing observations, the deviation of the effective radius profile increases towards the cloud top"
**It seems that the LWC profile deviates more than the reff at cloud top.**

p.15, L5 – **How many profiles go into Fig. 11?**

p.15, L11-12 – **What instruments were used to determine the ice water content and ratios? More detail is needed here, rather than just citing Buhl (2016)**. Also, what location within the cloud are these values representative of? From the Figure 12 caption it seems they may be vertical averages over thin cloud layers. What criteria is used to determine thin clouds? What is the horizontal averaging period and what is the time period covered by the data?

p.15, L16-17 – "can be explained with the increasing amount of adiabatically available liquid water with increasing temperature"
    **This could also be due to the increased concentrations of ice nucleating particles at colder temperatures, which should be mentioned. Can the analytically predicted change in the adiabatic condensation rate explain these large differences?**

p.15, L31 – **Fig. 13 only shows data from one day - is it possible to quote statistics from more days, or the whole campaign. Correlation coefficients between the two measurements would be useful.**

p.16, L26-27 – "the model is operating in the so-called "grey zone" where the parameterization of turbulence in the convective ABL may not be necessary"
    **Will it not be the case that some form of sub-grid turbulence parameterization will always to be necessary even at fairly high resolution? What types of turbulence parameterizations are you talking about here? Also, the term "grey zone" is usually used to refer to the resolutions where convective parameterizations are no longer needed, but that are still too coarse to resolve the important eddies. Maybe this refers to a "grey zone" for the boundary layer parameterizations, or something else? But this needs to be made clear.**

p.17, L9 "ratio of cloud fraction by volume and by area"
    **Not quite sure what is meant here – can you please explain this a bit more?**

p.17, L21 "the turbulence characteristics of the ABL were captured satisfactorily"
    **Can you please give more detail? Captured to within what margin of bias?**
p. 17 L24 – **as above – surface fluxes were reproduced to within what range?**
p. 17, L27 – **what was the resolution of the coarser model?**
p. 17, L28-29 – **more detail is needed again – how closely did the turbulence profiles and integrated water vapour match and what timescales do you mean?**

p.18, L12 – **TTRL has not been defined yet.**

p.19, L6 – "ratio of ice to liquid water decreases with decreasing cloud top temperature"
    **As the temperature got lower (colder) the ratio went up**

**Figures**

Fig. 1
    **A higher resolution image would be better if possible. Also, the rain radar label is a bit unclear on left panel.**
Fig. 2
    Dashed lines denote circles of constant distance from supersite JUE
    **This is repeated in the caption.**

Fig. 4

Need to make it clear in the caption what the var($T_D$) line is in Fig. 4a. Also explain that the x-axis is the time period (or the inverse of the frequency).

Fig.5

It looks like the line colour labels are the wrong way around for the water vapour and temperature.

Fig. 6

In the caption, please mention LACROS (as in the legend) and WiLi (as in the manuscript text) for clarity.
Also, what are the mean values for the two distributions?
It would also be good to have a statistical test to show how similar the distributions are.

Fig. 11

You should mention that ACTOS is in-situ data in the caption.

Fig. 12

It would be better to plot this with temperature on the y-axis if possible.

**Typos**

p.4, L19 – "than it was the case for HOPE" - remove "it"
p.9, L5 - "All data will be public available by the end of 2016." – "All data will be publicly available by the end of 2016."
p.11, L28 – move "also" to after "should"
    L29 – replace "higher resolved" with "more highly resolved" or "better resolved"
p. 14, L2 – "Figure 9 exemplary" – better as "Figure 9 shows an example of…"
p. 16, L2 – add "are" between "and" and "independent"
p.17, L4 – add comma after "in the past"

p.17, L13 – "general weather patter where"
        -> general weather pattern was
p.17, L16 – Schween reference incorrectly all in brackets.
p.17, L19 – "subject of uncertainties" -> subject to uncertainties.
p.18, L13 – "a key information" -> key information
p.18, L21  - "constrain" -> constraint

---

## Author Comment (AC1) · 7 Feb 2017

We thank the reviewer for the constructive and helpful comments.

**General Comments**

(Note, I have referenced sections in the paper using Page (P) and Line (L) position throughout this review).

This article provides an overview of the HD(CP)$^2$ Observational Prototype Experiment (HOPE). This paper does not appear to present new work but rather, provides an extensive summary of work that has been done using the HOPE experiment to date coupled with the experiment description. Given the complexity and breadth of HOPE, there is definitely value in an overview like this to help the science community understand the experiment and to make good use of the data it produced. Generally, I like the structure with a description of the physical layout of the campaign followed by science results and application to model evaluation. I have a few relatively minor concerns. However, I think there are certain aspects of the paper that could be improved to make it more useful.

We appreciate the suggestions given by Referee #1 to straighten the representation of the HOPE Overview Paper and to put it into a clearer context. In order to extract the main messages from the general comment of the Referee we decided to introduce an enumeration. We will provide replies to each item. We have also dedicated a reply to each specific comment.

The two most significant issues I have with the paper are regarding the motivation and some of the description in section 2.

1.

At a high level, the authors have explained the goal of HOPE and what measurements will be provided. However, ultimately, there is little motivation provided beyond stating that the measurements from HOPE will be used for the initialization and evaluation of the ICON model.

It would have been helpful, for example, to hear if there were particular areas of concern – or types of numerical experiments that were planned for this coupled observation/model system.

The overall motivation of HD(CP)$^2$ is the development of a large-domain large eddy simulation (LES) model based on ICON for process studies to provide a temporally and spatially highly resolved realistic virtual reality of physical processes (e.g., turbulent fluxes, convection, clouds) which are commonly unresolved in numerical weather prediction models. Such an approach is required for the verification of mesoscale models, and improved weather and climate simulations. In this connection, the objective of HOPE was "…the observation of the spatio-temporal variability of water vapor, temperature and cloud and precipitation properties with unprecedented detail. The experiment will resolve all relevant spatial sub-grid scale processes that are subject to parameterizations even at the resolution of the HD model; these scales are believed to be most critical for the onset of clouds and precipitation. ..."

Thus the main "area of concern" was the degree of realism of the high-resolution LES runs when compared to observed atmospheric properties, in particular in relation to land-atmosphere exchange, atmospheric turbulence as well as cloud and precipitation formation and evolution.

In order to improve the description of this motivation, we re-formulated the HOPE-specific part of Section 1 and added the following text:

*P3, L17-20: "When compared to model results, the high resolution HOPE data could elucidate to*

*what extent a pure increase in model resolution improves model skills in the atmospheric boundary layer, and to what extent unavoidable parameterizations of physical processes - essentially turbulence and cloud microphysics - require new approaches."*

*P4, L1ff: "Although phase 1 of HD(CP)[2], lasting from 2012 to 2014, was mainly devoted to establish a scalable high-resolution ICON model and to obtain data for model evaluation at various scales, first high resolution ICON-based LES have been performed to evaluate the effect of resolution on reproducing boundary layer fluxes and heights as well as on cloud formation. First results are reported in this overview. …"*

In addition, we added a new paragraph to the "summary & conclusions" section (Sect. 5) to provide information about how and where HOPE observations will be used in the future for the advancement of ICON:

*P24, L30-P25,L8: "In future, HOPE data will continue to contribute to the development, evaluation and improvement of high-resolution NWP and LES models because the data will be available via the Standardized Atmospheric Measurement Data (SAMD) data base which fulfills the needs of in particular model experts. Focused on the ICON development and on the collection of observational data for model evaluation, phase 1 of HD(CP)² set the starting point for an ongoing, synergistic use of HOPE and other observational data by the modelling community. In phase 2 of HD(CP)², which started in 2015, HD(CP)[2] participants are already making use of these observations. For instance, a project on boundary layer clouds will confront ICON with HOPE data for different cloud regimes at different spatiotemporal scales. A project addressing fast cloud adjustment to aerosols will exploit remote-sensing and in-situ observations of aerosol and cloud properties to evaluate the susceptibility of the model performance to different representations of aerosol in the model, e.g., to variations in the concentration of nuclei for cloud droplets or ice crystals. A project on the effects of surface heterogeneity e.g. uses the HOPE observations to challenge the applicability of the Monin-Obukhov Simularity Theory (MOST) and the reproduction of the vertical boundary layer structure and turbulence on small scales. Other projects apply the observations of the 3D water vapour fields and the cloud microphysical properties derived with Cloudnet for the development of convection parameterizations, just to mention a few."*

2.

> Also, while I understand that this is an observation paper, not a modeling paper, I would have expected a bit more with regard how these data are intended to be used for model evaluation. At least pointers to model planning. In the introduction (P3,L1) it is stated that "Key to this effort was the provision of modeled scenarios at 100 m grid resolution over thousands of kilometers, . . ." But I do not subsequently find any mention of those modeled scenarios or work being done toward their development. There is a modeling section (4) but it is mainly a list of a few evaluation studies. It is not even clear if all of these studies make use of the ICON model which the introduction indicates is the target for this work. ICON is mentioned toward the end of section 4 (P17,L25) but it is not clear if the other studies mentioned earlier in that section are using ICON or some other more established model(s). In short, I don't get a sense of an overall set of goals either for the science applications or for linking the observations to models.

The actual model evaluation as well as the development and implementation of evaluation techniques within HD(CP)² is still ongoing, see Muppa et al 2017. This HOPE overview paper aims at providing the current state of these activities. The application of the observations for the

model evaluation is mostly presented in Section 4 of the manuscript. Nevertheless, we also highlight several times in the text, that the usage of HOPE data will be ongoing in the future. As already stated in the response to the above comment, we currently just provide examples for the future applications of HOPE data in a new paragraph in the "summary & conclusions" section, We also added an extensive introduction to Section 4 (P19-P22) in which the available ICON runs are documented in order to provide the reader with an overview on the already performed model runs.

Reference: Shravan Kumar Muppa, Andreas Behrendt, Kirsten Warrach Sagi, Florian Späth, Hans-Stefan Bauer, Norbert Kalthoff, Vera Maurer, Rieke Heinze, Christopher Moseley, Roeland A.J. Neggers and Volker Wulfmeyer: Characterizing turbulent processes in the convective boundary layer: Evaluation of large eddy simulations with high-resolution lidar observations, in preparation, 2017.

3.

My other main issue is with the description of the measurements in section 2. Through- out, the authors are trying to present a lot of material and I appreciate that. As noted earlier, I think this will provide a valuable reference. And for the most part, I think the structure works well for the discussion of Results (section 3). But in section 2, particularly the discussion of HOPE-Jülich in section 2.1.1, I was struck by the very long lists of instruments presented in text without any sub-organization. I think this section would be easier to read and to use if it were organized in some way – either with sub-headings for sites – or sub-headings for major classes of instruments. As it is – it is difficult to take away a clear picture of how the instruments are coordinated.

In order to improve the readability of the description of the instrumentation during HOPE-Jülich, we followed the advice of the reviewer and restructured section 2.1.1 by adding dedicated headings for each site and the networks, including more extensive descriptions.

4.

I would also say that similar to my earlier comments about motivation for science and the modeling – more could be done here to explain why this particular set of measurements was chosen.
It is stated at a high-level that there is a goal "to derive the atmospheric state of water vapor, temperature, wind and cloud and precipitation properties with 100-m resolution for an area of about 10x10x10 km . . ." (P4,L6-7). The list of instruments is impressive and certainly it provides some spatial representation of these parameters. However, I am sure that these instruments cannot provide all of those parameters over the full domain. It would be helpful to understand how close to achieving this goal HOPE came, how instrument configurations were chosen to get as close to this goal as possible, and if there are any thoughts regarding how to get closer to the goal of full sampling of the stated domain. Why was this particular set of instruments chosen – and how do they enable achieving core goals of the activity?

We agree with the referee that during HOPE this goal was yet not fully reached. However, the synergy of scanning water-vapor DIAL, temperature RRL, and Doppler lidar systems demonstrated the excellent progress in this area. During HOPE, it was possible to reach a range of several km (Späth et al. 2016) and with further advancements of sensor sensitivity in the near future, we reach ranges of more than 10 km (e.g., Wulfmeyer et al. 2015). We added a paragraph at the beginning of Section 2 in which we state why the specific instrumentation was deployed.

*P4, L14-23: "The technological aspect of HOPE was to unite most of the mobile ground-based remote sensing and surface flux observations available in Germany within a single domain in order to capture the vertical structure and horizontal variability of wind, temperature, humidity as well as aerosol and cloud condensate with the best possible temporal, and spatial resolution. Thus we were able to accommodate active remote sensing from lidar and radar, passive remote sensing from microwave radiometer and Sun photometer, whenever possible with scanning capabilities. During HOPE, 3D water-vapour, temperature, and wind measurements were possible with unprecedented spatiotemporal resolution in the boundary layer. In order to understand the forcing of and the response to surface properties, distributed surface flux and surface standard meteorological observations were deployed as well. Of course, it is not possible to obtain an instantaneous 3D picture of the atmosphere from a limited number of directional observations. However, ongoing improvements in sensor detection accuracy and optimized scanning strategies will capture the 4D boundary layer properties even better in the future."*

5.

Finally, before I move on to mentioning more specific issues, I suggest it would be helpful to comment somewhere in the Summary and conclusions if there are any specific plans for using the observation data set or particularly, for applying the data set to the ICON model.

Similar to the response given to discussion points 1 and 2 above, we would like to advert the referee to the new paragraph that we added at the very end of the "summary & conclusions" section (P25).

**More specific science issues**

P1,L38: what does it mean to operate the sun photometers in synergy?  Or does the operation in synergy refer to all the previously mentioned instruments (lidars, radars etc)? Either way – does this just refer to the spatial placement of instruments? Or are instruments being scanned in such a way to optimize their co-collection of observations?

We rephrased the sentence (P2, L1) to specify more clearly that we (a) operated instruments at different sites of which (b) some were operated in synergy.

Actually, scanning measurements were largely performed in synergy during HOPE-Jülich. These measurements comprised simultaneous synchronized scans of the cloud radar systems of KRA and JUL, the Doppler lidar systems of KRA, JUL, and HAM (as well as other sites of KIT), as well as of the microwave radiometers of KRA, JUL, and HAM. So far, only studies based on the synchronized Doppler lidar scans are available, documenting the properties of the horizontal wind field in the HOPE Jülich area (Träumner et al., 2015; Maurer et al., 2016). To date, no quantitative results (i.e., publications) were published from the synchronized scans of the cloud radars and the microwave radiometers. Nevertheless, software has been developed mandatory to operate the instruments in synergy. We decided not to discuss these measurements in the manuscript because, as mentioned above, no results were published from the observations yet.

P2,L16: The text says "It is a coordinated initiative . . .". This is confusing because the previous sentence was specifically talking about the HD(CP)2 model – but I presume the sentence beginning on line 16 is referring to the larger HD(CP)2 initiative – not specifically the model. In any case, I found the sentence to be confusing.

We actually addressed ICON at the position in the text highlighted by the referee. To avoid confusion, we rephrased the beginning of the sentence to specify more clearly what we mean with initiative and also added minor modifications to P2,L15-18.

> P2,L29-32 – the newly developed ICON model is mentioned. Is this the same as the HD(CP)2 model mentioned on line 15 of this page? Not clear.

Yes, at both instances we indeed refer to the same model. The ICON model is now already introduced in the text at the position suggested by the referee (see previous comment).

> P2,L30-31 – mentions that the observation datasets are intended to provide both initialization and evaluation of the ICON model. This is mentioned again on P16, L22-23. I can see that these data would be excellent for model evaluation – and a few example are given in section 4 (p16-17); however, I don't see any discussion of using these data for model initialization or how that might be done. I could imagine that these data could be used for that as well – but given that the point is made that these data are ideal for that purpose – it would be preferable to have some discussion on what the authors have in mind for that application – or what others have done or are planning in this area.

We are glad that the referee pointed us to the missing reference to model-initialization activities. Actually, no many studies did so far use HOPE observations for the assimilation into NWP models. One recent attempt was added to the end of Section 4.1 (P22, L13-24):

*Regarding the application of HOPE observations for the initialization of NWP models, a first attempt was recently reported by Adam et al. (2016) who concentrated on the 24 April 2013. In their study the authors assimilated lower-tropospheric temperature profiles from the temperature rotational Raman lidar, reaching from about 500 to 3000 m above ground, into the Weather Research and Forecasting (WRF; Skamarock et al., (2008)) model using a 3D-variational method (Barker et al., 2004). The WRF model was covering Central Europe with 57 vertical levels and 3-km horizontal resolution. The assimilation of the temperature profiles from the TRRL in addition to the conventional assimilation of conventional data of including zenith total delay from Global Navigation Satellite System and operational radiosonde data were found to improve the agreement of measured boundary layer height and temperature gradient to the modelled values. Nine hours after the assimilation of TRRL data was initialized, already an area of 100 km in radius around the HOPE-Jülich area was affected, showing a temperature deviation from the conventional run using radiosonde data only of up to 2.5 K at 2.5 km height above sea level. Similar impacts can also be expected for the assimilation of continuous profiles of water vapour mixing ratio from continuous lidar observations, as was found in an earlier study of Grzeschik et al. (2008).*

To some extent also other HOPE data actually went into model initialization activities because the records of the atmospheric soundings performed during HOPE-Jülich and HOPE-Melpitz were added to the global data assimilation systems. Since this is a standard procedure, we do not mention this in the manuscript.

> P5-6 – in this portion of section 2, the instrumentation of HOPE-Julich are described. I think this section really needs some sub-sections. This could be done by supersite or by instrument class or both. But as it is – this section comes across as a very long list of instruments that is very difficult to digest. Some structure would help get clear how these instruments are arranged.

We went through section 2.1 in order to improve the readability and to provide a better

description of the instrumentation. We basically introduced headers for each supersite and the networks.

> P9, L2-3 – the HD(CP)2 data archive center is mentioned. I think it would be good to provide a link to this center. I think I was able to find it – but it would be helpful to provide the actual link here.

We updated the date when the archive went online and do now also provide the link to the data archive, see Sect. 2.2.3 (P10).

> P9,L8 – indicates that an "essential regime" observed during HOPE was the turbulent structure of the atmosphere. I don't think I have ever seen the word "regime" used this way before. I think of "regime" as referring to a meteorological state – not the general distribution of a physical attribute.

We apologize for the confusion and rephrased the sentence to: P10,L15: "*A central goal of HOPE was the characterization of the turbulent structure of the atmospheric boundary layer (ABL).*"

> P15,L5: The text indicates that the "LWC agrees well with the in-situ and remote sensing observations".  First of all, this should be reworded along the lines of "the LWC derived from remote sensing observations agree well with in-situ measurements".  However, I also have some concerns about this statement. The phrase "agrees well" is generally subjective, but in this case, I would argue that the two are diverging near cloud top and that deserves some mention.

We reformulated the corresponding paragraph and now provide more-detailed information on the intercomparison between in-situ and ground-based observations of the cloud microphysical parameters. In this manner, the corresponding Figure 12 was updated. In the revised version, Figure 12 also contains the time-height cross sections from the ground-based observations for the investigated time period. The profiles of liquid water content (lwc) and effective radius (re) were extended by the data points that were used to derive the statistics. The revised text (P17, L4-19) now presents more details on the flight pattern of ACTOS. Concerning the interpretation of the observed differences between in-situ and remote sensing, we provide possible reasons but leave a detailed analysis to an upcoming publication.

*"The implemented Frisch-2002 retrieval of cloud droplet effective radius and the Cloudnet retrieval of the adiabatically scaled LWC were evaluated against in-situ observations of ACTOS for a stratocumulus deck observed simultaneously by ACTOS and LACROS during the HOPE-Melpitz campaign on 22 September 2013 (IOP 22) from 09:59-10:16 UTC, as is shown in Figure 12. During the time period, ACTOS constantly flew horizontal legs of 2 km length in cross-wind direction in a distance of about 500-m upwind of the LACROS site. Time-height cross-sections from the continuous LACROS observations as shown in Figure 12 (a) and (b) will be available in the SAMD database (Sect. 2.2.3) for entire HOPE-Jülich and HOPE-Melpitz. The comparisons of the average vertical profiles of LWC and cloud droplet effective radius observed with ACTOS and retrieved with LACROS are shown in Figure 12. It can be seen that ACTOS probed mainly the mid-upper part of the cloud layer. Both, the observations of the LWC of the cloud droplet effective radius of ACTOS and LACROS (Figure 12a) are within the range of one standard deviation, as is shown by the horizontal error bars. Beside the found absolute differences, the profiles of LWC and effective radius retrieved from the LACROS observations deviate more strongly from those of ACTOS toward cloud top. A possible explanation for the observed discrepancies is the temporal variability of the LWC and effective radius in the cloud-top region as is shown in Figure 12 (a) and*

*(b). Also, ACTOS was not flying directly above the LACROS site. Considering the applied retrieval of Eq. 5 in Frisch et al. (2002), also the assumption of a certain shape of the size distribution and of a cloud droplet number concentration can introduce biases. The application of the co-located observations of ACTOS and LACROS for the evaluation of ground-based retrievals will be discussed in an upcoming publication (Seifert et al., 2017)."*

P16,L26-30. The text refers to activities related to the grey zone and characterizes this as conditions "where the parameterization of turbulence in the convective ABL may not be necessary". I think this is not a good representation of the issues posed by the grey zone. The issue as I understand it – is that in this spatial resolution zone (~1km ~10 or 20 km) the resolution is too coarse to explicitly resolve certain features (e.g. eddies associated with shallow convection) but the resolution is too fine for traditional parameterizations to work. So I don't think the issue is so much that the parameterizations aren't necessary is that traditional parameterizations break down – because the assumption that the domain is much larger than the phenomena being sampled is no longer true.

We agree with the referee that the sentence with „where the parameterization of turbulence in the convective ABL may not be necessary" is misleading. The grey zone is the region where turbulence still needs to be parameterized (but in a resolution-adaptive manner), in contrast to large-eddy simulations, which resolve the energy-containing turbulence. The referee is also right about the fact that traditional parameterizations used in mesoscale modelling might not be appropriate for the grey zone. We rephrased this part of the manuscript.

P17,L16-17. The text states that the LES simulations "qualitatively reproduce the observed boundary layer heights within the observation uncertainties". This is not obvious. First of all – saying they agree within the uncertainties is a quantitative statement. And it appears that while they do agree well at sometimes, at others they clearly diverge. So – indicating that they agree seems like a simplification of what is going on.

We extended the description of the results in Section 4.1 and provide now also quantitative information about the correlation between models and observations. The statement now reads as follows:
*P21, L15-18: "Figure 16 exemplarily shows the temporal evolution of the boundary layer height as derived from different model runs and from observations. The 2-hour (12-14 UTC) mean boundary layer depth derived by the PALM model agreed within 400 m with the different observation methods and with the COSMO-DE run at 2.8-km resolution."*

**Typographic/syntax**

P3,L15: builts should be "builds"

corrected

P11, L29: the wording "that are highly resolved" is awkward or incorrect. I suggest changing this to something like "that are more highly resolved"

We could not find the occurrence of that phrase in the manuscript. At the position to which the referee pointed, we wrote '…that are higher resolved than…' and we consider this phrasing accurate. Still, we decided to formulate more clearly what we mean with "high resolution" when we mention this phrase in the text. (e.g., P13,L15-17)

11, L32: I think "lays the ground for . . ." should be "lays the groundwork for . . ."

Corrected

P12, L6: I think that "it infers that Doppler lidars. . ." should be "it implies that Doppler lidars . ."

Corrected

P14,L2:  I think the wording "Figure 9 exemplary shows the aerosol. . ." should be reorded/reordered as "Figure 9 shows an exemplary aerosol . . ."

Corrected

P16,L4: "state of the measured polarization state" seems redundant. I would think you could just say "measure

We rephrased this sentence to "…aggregation with the measured polarimetric properties of the hydrometeors."

**The HD(CP)² Observational Prototype Experiment HOPE – An Overview**

[revised manuscript text omitted]

The O4 project in the O module of HD(CP)[2] was devoted to HOPE and has been designed to provide a critical model evaluation at the scale of the model simulations and further to provide information on sub-grid variability and microphysical properties that are subject to parameterizations even at high-resolution simulations such as planned with ICON. Even for LES, unresolved sub-grid scale processes are believed to be in particular critical for cloud formation and the onset of precipitation, and thus built the central focus of HOPE. In order to derive the atmospheric state and the 3D fields of water vapour, temperature, wind and cloud and precipitation properties at the scale of 100-m resolution for an area of about $10x10x10$ km$^3$ three close-by supersites, separated by a distance of approximately 4 km, complemented by larger networks were deployed. The instrumentation was selected in order to allow for detailed observations of the onset of clouds and precipitation in the convective atmospheric boundary layer. When compared to model results, the high-resolution HOPE data could elucidate to what extent a pure increase in model resolution improves model skills in the atmospheric boundary layer, and to what extent unavoidable parameterizations of physical processes - essentially turbulence and cloud microphysics - require new approaches.

HOPE complements the larger spatiotemporal Full-Domain (O2) and Supersites (O1) activities in the observations module in HD(CP)[2] of which O2 provides continuous time series of 2D fields across the HD(CP)[2] domain and O1 is devoted to the provision of 1D profiles at four dedicated locations in Germany and the Netherlands, respectively. The scope of Module O3 was to establish a data flow from the observation modules to the model and synthesis modules. In 2016, HD(CP)[2] entered its second phase, which puts a much stronger effort on the synthesis part.

HOPE builds on the experience gained in previous field campaigns like the Convective and Orographically-induced Precipitation Study (COPS) (Wulfmeyer et al., 2011), however, with a stronger focus on multi-sensor synergy covering a micro- to meso-scale domain. COPS and the associated General Observation Period (GOP) that was prepared in the context of the Quantitative Precipitation Forecasting priority program (SPP1167) of the German Science Foundation (DFG) (Crewell et al., 2008) aimed at the observation of orographically driven initiation of convection with supersites several tens of km apart in strongly structured terrain. Complementary to COPS, HOPE is covering a smaller domain with higher resolution, and is accompanied by long-term supersite observations within the framework of the Terrestrial Environmental Observatoria (TERENO) Programme (Simmer et al., 2015) around the ground-based remote sensing supersite JOYCE (Löhnert et al., 2015), and the TROPOS long-term aerosol observatory in Melpitz (Spindler et al., 2012).

Although phase 1 of HD(CP)$^2$, lasting from 2012 to 2015, was mainly devoted to establish a scalable high-resolution ICON model and to obtain data for model evaluation at various scales, first highly resolved ICON-based LES have been performed to evaluate the effect of resolution on reproducing boundary layer fluxes and heights, as well as on cloud formation. First results are reported in this overview.

5    This article mainly serves as a guide through the sites and instrumentation used during the HOPE campaigns and it is aiming on giving a motivation to learn about the details and specific conclusions described in the individual publications this overview is built upon. The structure is as follows. Section 2 describes the site setups and measurements performed during HOPE including information about the meteorological conditions and data availability. Examples from each of the research topics are presented in section 3. In section 4, first comparisons between models and observations are discussed. A summary and

10   conclusions on the further applications of the HOPE data as well as designs for future observational strategies are presented in section 5. Individual work performed during HOPE is published in this ACP/AMT HOPE special issue or, in part, in other journals and is cited in the present overview correspondingly.

**2 Description of the HOPE field campaigns**

The technological aspect of HOPE was to unite most of the mobile ground-based remote sensing and surface flux observations

15   available in Germany within a single domain in order to capture the vertical structure and horizontal variability of wind, temperature, humidity as well as aerosol and cloud condensate with the best possible temporal and spatial resolution. Thus we were able to accommodate active remote sensing from lidar and radar, and passive remote sensing from microwave radiometer and Sun photometer, whenever possible with scanning capabilities. During HOPE, 3D water-vapour, temperature, and wind measurements were possible with unprecedented spatiotemporal resolution in the boundary layer. In order to understand the

20   forcing of and the response to surface properties, distributed surface flux and surface standard meteorological observations were deployed as well. Of course, it is not possible to obtain an instantaneous 3D picture of the atmosphere from a limited number of directional observations. However, ongoing improvements in sensor detection accuracy and optimized scanning strategies will capture the 4D boundary layer properties even better in the future.

[revised manuscript text omitted]

field for the HOPE-Jülich area every 5 minutes and with 250-m radial resolution. In-situ vertical profiles of temperature, humidity, and wind profiles as well as convective indices were gathered by radiosondes launched regularly every $6^{th}$ full hour at the KITcube main site. Land and full-sky images were taken by S14 camera systems at HAM and WAS.

Also at supersite HAM, two lidar systems from the Institute for Physics and Meteorology (IPM) of the University of
5  Hohenheim observed 3D thermodynamic fields of temperature and moisture including their turbulent fluctuations. A temperature rotational Raman lidar (TRRL) measured temperature profiles (Radlach et al., 2008;Hammann et al., 2015) and a water-vapour differential absorption lidar (DIAL) measured absolute humidity profiles (Behrendt et al.;Späth et al., 2016;Wagner et al., 2013). In contrast to the Raman lidar technique, the DIAL technique, which is based on the alternating emission of laser pulses at frequencies strongly and weakly absorbed by water vapour, does not require calibration. By sending
10  out the laser beam vertically into the atmosphere, high-resolution observations of the convective boundary layer and the lower free troposphere can be made with the instrument (Muppa et al., 2016;Wagner et al., 2013). But the same system also allows for observations in any direction of interest and thus to map the structure of the water vapour field and its development (Milovac et al., 2016). Like the DIAL also the TRRL of IPM has scanning capabilities 
[revised manuscript text omitted]
 in the 10x10 km²-sized domain, reaching about 30 W m$^{-2}$ for 3-hourly and 80 W m$^{-2}$ for 1-s-resolution observations.

Also based on the horizontally high resolved measurements of the irradiance from the pyranometer network (PYR) performed by TROPOS, Lohmann et al. (2016) analysed the statistics of spatiotemporal irradiance fluctuations with a strong applicationoriented focus on photovoltaic power systems. They specifically calculated single-point statistics and two-point correlation coefficients for clear, overcast and mixed skies. The statistics for clear and overcast skies show similar behavior as in previously published work, see Lohmann et al. (2016) for references. In order to account for conditions for a distributed PV system, they defined so-called irradiance increments as changes in transmissivities over specified intervals of time, and showed that the magnitude of increments is more strongly reduced by spatial averaging than that of the fluctuations. By conditioning the sky type - which can easily be done from the irradiance measurements themselves - they demonstrated that the probability for strong irradiance increments is twice as high compared to increment statistics computed without distinguishing between different sky types.

As clouds impose the largest short-term variability in solar irradiance at the surface the analysis of cloud advection and subsequent extrapolation represents a reasonable approach for short-term irradiance forecasts. Schmidt et al. (2016) made use of time series of hemispheric sky images to predict the surface irradiance by means of mapping the cloud position, which in turn is translated into shadow maps at the surface. The temporal evolution of such shadow maps is calculated from cloud motion vectors that were calculated from subsequent sky images. Irradiance forecasts of up to 25 minutes have been produced and were validated against the network of pyranometers described in Madhavan et al. (2016b). Although these sky-imager-based forecasts do not outperform a simple persistence forecast on average, improved forecast skill was found for convective cloud conditions with high cloud and irradiance variability. This finding may provide useful application in photovoltaic electricity production.

**3.2 The turbulence structure of the boundary layer and clouds**

The goal of the HD(CP)$^2$ project was to realize and to evaluate a model run spanning the area of whole Germany at the horizontal resolution of 100 m. At such a small scale, certain parameterizations for organized turbulent motions such as those that define the atmospheric boundary layer, and areas of shallow convection are supposed to be not required anymore. Hence, the setup of the envisioned model is comparable to the one of a large-eddy-simulation (LES), wherein the sub-grid parameterizations are simpler and have less impact on the model performance (Bryan et al., 2003;Deardorff, 1970).

The increased model resolution puts new requirements on evaluation techniques. The HOPE experiments provided an optimum test bed for novel applications to derive boundary layer fluxes and turbulence characteristics. Observations of the turbulent fluxes of thermodynamic properties in the PBL, such as of temperature and water vapour, provide detailed information on the minimum resolution required by a model to capture the turbulence spectrum down to the inertial sub-range and consequently to resolve the major part of the turbulent fluctuations. This value is in here introduced as the integral scale. During HOPE-Jülich, it was possible to derive based on lidar observations the statistics of turbulent temperature fluctuations and thus of the integral scale of this parameter in the PBL (Behrendt et al., 2015). In addition to commercially available Doppler lidar systems, which provide turbulent wind fluctuations, three water vapour research lidars were deployed during HOPE-Jülich, which provide turbulent humidity fluctuations that were documented by Di Girolamo et al. (2016), as well as Muppa et al. (2016).

As the authors of the above-mentioned studies note, HOPE-Jülich provided for the first time data to observe the turbulence characteristics of the PBL, more specifically the convective boundary layer (CBL), up to the fourth statistical moment, i.e., the mean, standard deviation, variance, skewness, and kurtosis of the spatiotemporal water vapour and temperature. Examples of the relationship between the integral scales (introduced in section 3.1) of humidity and temperature fluctuation and height above ground within the CBL for the 20 April 2013 (IOP 5), 11:30-13:30 UTC, (only temperature fluctuations; see Di Girolamo et al. (2016)) and 24 April 2013 (IOP 6), 11:00-12:00 UTC (temperature and humidity fluctuation; see Behrendt et al. (2015) and Muppa et al. (2016)), respectively, are depicted in Figure 6. A decrease in the integral length scale of the water vapour mixing ratio with height in the upper part of the CBL was found at the HAM site similar to previous observations (Couvreux et al., 2005;Wulfmeyer et al., 2010). A similar decrease was found for temperature at the same site. The temperature observations from JUE site show a more complex structure. The reasons for this are still under investigation. The decrease of the integral length scale toward the top of the CBL can be explained by the decrease in the size of the turbulent eddies with height resulting from the entrainment of dry free-tropospheric air at the CBL top (Couvreux et al., 2005) which is also characterized by an increase in the variance of the temperature or water vapour toward CBL top. Converting the observed time scales shown in Figure 6 to spatial scales assuming horizontal and vertical wind velocities of 5 m s$^{-1}$ and 1 m s$^{-1}$, respectively, results in horizontal and vertical integral length scales of 100-1000 m and 20-200 m, respectively. Thus, in order to capture the full turbulence spectrum in the CBL, a numerical model simulation should also be run at temporal and spatial resolutions that are better resolved than the observed values.

Detailed convective boundary layer turbulence characteristics from HOPE and further field campaigns (Wulfmeyer et al., 2016) showed that the combination of active temperature-, humidity- and wind-profiling applied during HOPE-Jülich sufficiently resolves the turbulence structure of the CBL and lays the groundwork for new boundary layer turbulence parameterizations.

In addition to turbulent fluxes in the cloud-free planetary boundary layer, the turbulence characteristics of a stratocumulus layer were investigated simultaneously with ACTOS and the Doppler lidar WiLi of the LACROS site on 22 September 2013 during HOPE Melpitz. The inter-comparison shown in Figure 7 presents a histogram of the vertical velocities observed with ACTOS (red) and WiLi (blue), further insights into the microphysical properties of the cloud layer are given in Sec. 3.4 and Figure 12. The variability of the vertical velocities (with the mean adjusted to 0 m s$^{-1}$ and corrected for large-scale trends) during the cloud observation time of 16 minutes were found to be similar at the stratocumulus cloud base (observed with the Doppler lidar) and top (observed with ACTOS), with standard deviations of 0.23 m s$^{-1}$ for ACTOS and 0.21 m s$^{-1}$ for WiLi. This is an important fact for Doppler-lidar studies of stratocumulus clouds, because it implies that Doppler lidars are suitable to characterize the turbulence characteristics of entire stratocumulus cloud layers. From the vertical-velocity observations of WiLi and ACTOS also integral length scales were derived which were in the range from 38 m (ACTOS) to 45 m (WiLi). The observations will be further discussed in an upcoming publication (Seifert et al., 2017)

[revised manuscript text omitted]

Based on scanning measurements with the water-vapour DIAL of IPM made during HOPE-Jülich, Späth et al. (2016) (see Sect. 2.1.1) presented a detailed study of the 3-dimensional structure of the water vapour field between the supersites HAM, KRA, and JUE with a range resolution of 30-300 m and a temporal resolution in the range of 10 s for each profile. Full conical scans (360° in azimuth) around the site to characterize the water vapour field at a defined elevation angle took 15 min. Such observations provide valuable information for improving our understanding of land–atmosphere exchange processes as different types of land cover results in different evapo-transpiration and thus moisture in the CBL.

**3.4 Microphysical properties of aerosols and clouds**

The retrieval and evaluation of microphysical properties of aerosols, clouds, and precipitation from ground-based remote sensing observations is a crucial task. In-situ observations do provide much higher accuracy but for the long-term evaluation of the performance of operational weather forecast models and the microphysical parameterizations therein continuous datasets are required. In particular the HOPE-Melpitz campaign provided the opportunity to relate in-situ observations of warm-cloud microphysical properties and of aerosol properties from ACTOS to the respective parameters observed with ground-based observations of the LACROS facility. Case studies are presented in the following that document the simultaneous ground-based remote-sensing and in-situ observations of a stratocumulus layer and of the aerosol properties in the lower troposphere, respectively.

Aerosol particles act as nuclei for cloud droplets and ice crystals and are thus a prerequisite for the formation of clouds. Lidar is a promising tool to provide estimates of the concentration of cloud droplet condensation nuclei (CCN) and ice nucleating particles (INP) (Mamouri and Ansmann, 2016). During HOPE-Jülich and HOPE-Melpitz the Raman polarization lidar Polly[XT] was continuously operated to provide information on the vertical aerosol structure in the planetary boundary layer and the troposphere. HOPE-Jülich was the first time a Raman polarization lidar provided a continuous data set of the calibrated attenuated backscatter coefficient at three wavelengths. Amongst other parameters, the dominating type of aerosol particles present in each observed volume was derived by a newly developed target classification, as is explained by Baars et al. (2017). Figure 10 shows an example of the aerosol target classification for three consecutive days from 24 to 26 April 2013 (IOPs 6-8) during HOPE-Jülich. Frequently large, non-spherical particles, probably dust or pollen particles that were emitted in the vicinity of the site have been monitored. The occurrence of these aerosol types is correlated with the development of the planetary boundary layer and they first appear close to the ground and are slowly dispersed into the boundary layer in the course of the day, as can be seen for 24 and 25 April in Figure 10. Baars et al. (2017) in addition present a case study that shows visual evidence of the dispersion of dust from the near-by open pit coal mine of Inden, west of the KRA site. With increasing distance from ground, the particles frequently grow by hygroscopic growth, leading to the presence of large,

spherical particles, as it was the case on 25 and 26 April. The mask also helps to identify whether a cloud layer was within or detached from the planetary boundary layer aerosol. Overall, the classification of cloud particles solely on the lidar observations is difficult. This will be overcome in a future step, by merging the multi-wavelength aerosol classification with the Cloudnet target classification presented in Illingworth et al. (2007).

5   Retrievals of microphysical aerosol properties, such as CCN concentration, from lidar observations, as well as retrievals of the ambient scattering properties of an aerosol population measured in-situ are still subject to large uncertainties. In-situ observations of aerosol properties are usually performed under dry conditions and inlets are limited by a maximum cut-off size of an aerosol distribution. During HOPE-Melpitz, both in-situ aerosol observations as well as lidar observations of Polly[XT] were available. Figure 11 presents the relationship of the backscatter coefficient observed with Polly[XT] and the respective

10   extinction coefficient obtained from the in-situ aerosol observations of ACTOS as derived by Düsing et al. (2017). Based on the low-humidity (dry-state) in-situ aerosol measurements of ACTOS, the ambient extinction coefficient was obtained at wavelengths of 355, 532, and 1064 nm using a Mie model and a hygroscopic-growth correction. 13 data points derived for different altitudes and conditions on 14 and 17 Sept. 2013 (IOPs 20 and 21; see Table 4) are included in Figure 11. Each in-situ data point is based on all (and at least one) 120-s aerosol particle number size distributions recorded during a period of

15   flight at a constant height. Averaging times for the lidar observations varied between 30 and 60 minutes. A linear relationship with significant $R^2$ values was derived between the modelled in-situ and remote-sensing extinction coefficients. For 355 nm 54% of all cases agree within the uncertainties and for 532 nm 55% of the cases. On average, the model underestimates the measured extinction coefficients for 355 nm by 3.5% and overestimates the measurements by 7.9% at a wavelength of 532 nm. Correlation coefficients are 0.944 and 0.947, respectively. This shows that the ambient aerosol extinction coefficient can

20   well be derived from in-situ measurements given the extensive instrumentation for microphysical and chemical aerosol characterization that is available at the Melpitz field site.

During HOPE-Jülich the availability of CCN was investigated using an aerosol model. The approach presented by Hande et al. (2016) used the COSMO-MUSCAT model to simulate the generation and transportation of aerosols over Germany during the campaign. From the simulation results, a parameterisation of the CCN concentration was derived which can be applied

25   also to other climatological regions and different aerosol regimes. Even though the simulated aerosol properties were evaluated against in-situ observations of aerosol particle size distributions at Melpitz, no evaluation of the CCN parameterisation against measurements was performed. This emphasizes the need to improve remote-sensing techniques for the retrieval of CCN profiles as the one of Mamouri and Ansmann (2016).

At the beginning of the first Phase of HD(CP)² no operational microphysical retrieval of the effective radius of cloud droplets

30   from ground-based remote sensing observations was available within the project. As a first step towards an evaluation dataset for numerical weather forecasts, it was decided to apply the retrieval technique of Frisch et al. (2002) to the LACROS observations by implementing it into the processing framework of Cloudnet. The technique is based on vertically-pointing measurements from a millimetre-wavelength cloud radar and a microwave radiometer and produces height-resolved estimates of cloud particle effective radius and liquid water content. In addition, liquid water content profiles are produced operationally

within Cloudnet (Illingworth et al., 2007), assuming either adiabatic profiles of liquid water content (LWC) between the lidar-derived cloud base and the radar-derived cloud top or scaled-adiabatic profiles for which the adiabatic liquid water content is scaled to fit the liquid water path observed with the microwave radiometer (Merk et al., 2016).

The implemented Frisch-2002 retrieval of cloud droplet effective radius and the Cloudnet retrieval of the adiabatically scaled

5   LWC were evaluated against in-situ observations of ACTOS for a stratocumulus deck observed simultaneously by ACTOS and LACROS during the HOPE-Melpitz campaign on 22 September 2013 (IOP 22) from 09:59-10:16 UTC, as is shown in Figure 12. During the time period, ACTOS constantly flew horizontal legs of 2 km length in cross-wind direction in a distance of about 500-m upwind of the LACROS site. Time-height cross-sections from the continuous LACROS observations as shown in Figure 12 (a) and (b) will be available in the SAMD database (Sect. 2.2.3) for entire HOPE-Jülich and HOPE-Melpitz. The

10  comparisons of the average vertical profiles of LWC and cloud droplet effective radius observed with ACTOS and retrieved with LACROS are shown in Figure 12. It can be seen that ACTOS probed mainly the mid-upper part of the cloud layer. Both, the observations of the LWC of the cloud droplet effective radius of ACTOS and LACROS (Figure 12a) are within the range of one standard deviation, as is shown by the horizontal error bars. Beside the found absolute differences, the profiles of LWC and effective radius retrieved from the LACROS observations deviate more strongly from those of ACTOS toward cloud top.

15  A possible explanation for the observed discrepancies is the temporal variability of the LWC and effective radius in the cloud-top region as is shown in Figure 12 (a) and (b). Also, ACTOS was not flying directly above the LACROS site. Considering the applied retrieval of Eq. 5 in Frisch et al. (2002), also the assumption of a certain shape of the size distribution and of a cloud droplet number concentration can introduce biases. The application of the co-located observations of ACTOS and LACROS for the evaluation of ground-based retrievals will be discussed in an upcoming publication (Seifert et al., 2017).

20  The accurate representation of the ice phase in numerical models is a crucial task since cold rain is the main driver of precipitation formation at midlatitudes (Mülmenstädt et al., 2015). The continuous observations of the LACROS supersite during HOPE-Jülich enabled to obtain statistical information about the primary ice production in stratiform midlevel mixed-phase cloud layers. Figure 13 shows an overview about the ice water content and ice-to-total mass ratio of all mixed-phase cloud layers that were identified from the HOPE-Jülich observations. In these plots the method for measurement of ice

25  formation efficiency of Bühl et al. (2016) is used, which selects supercooled thin stratiform cloud layers with a turbulent mixed-phase (liquid-dominated) cloud top of a vertical extent of less than 380 m. In this way, non-linear ice formation effects like ice multiplication or splintering are avoided and, thus, do not affect the statistics. IWC is measured 60 m below the base of the mixed-phase layer, where an observation of the falling ice particles is possible without influence of water droplets or turbulent motions. LWC are mean values of the scaled-adiabatic approach (Merk et al., 2016) averaged over the complete

30  height of the shallow mixed-phase top layer of the cloud where liquid water is present. As shown in Figure 13, the ice water content of clouds with top temperatures above -10 °C was in general lower than $10^{-4}$ g m$^{-3}$. At temperatures below -15 °C, values of the ice water content vary around $10^{-3}$ g m$^{-3}$. The ice-to-liquid mass ratio decreases from $10^{-2}$ to $10^{-5}$ for temperatures increasing from -40 to 0 °C. The plots thus quantify how ice formation becomes more efficient with decreasing temperature. The colour-coded data points in Figure 13 provide in addition the radar-observed linear depolarization ratio of the observed

ice particles, which is a proxy for the particle shape. Values of around -20 dBZ (-10 °C <T<-5 °C), -30 dBZ (-20 °C <T<-10°C), and -25 dBZ (T<-20 °C) indicate needle-like, dendritic, and bullet-rosette-like shapes, respectively (Bühl et al., 2016;Myagkov et al., 2016). Knowing about the relationship between ice water content, liquid water content, temperature, and shape of freshly formed ice crystals is an important step towards new approaches for the evaluation of ice formation schemes

5    in numerical weather forecast models. This will also be a task of the second phase of HD(CP)[2].

**3.5  Macrophysical cloud & precipitation properties**

The combination of scanning polarimetric X-band Doppler rain radars, vertically pointing micro rain radars (MRR) and a ground-based network of disdrometers and rain gauges provided an excellent opportunity to validate the Doppler rain radar

10    ability to infer the spatial variability of quantitative precipitation properties from polarimetric radar reflectivities. Xie et al. (2016) performed a detailed analysis of all precipitation observations under different synoptic conditions. As an example, Figure 14 shows a time series of the surface precipitation rates estimated from measurements of three Doppler rain radar compared to the in-situ observations from seven disdrometers (partly from TR32 and TERENO projects), averaged over the disdrometer locations. The authors note that rainfall accumulations at the daily and even hourly scale were surprisingly

15    consistent between the different observations of rain gauges, disdrometers and X-band radar, at least for the low-intensity rainfall events (of $0.5 - 20$ mm day$^{-1}$) prevalent during HOPE-Jülich. The correlation was found to be better than 0.93.  The two near-by radars (KiXPol and JuXPol) showed slightly better agreement than the 50 km remote radar BoXPol, which is explained by its correspondingly larger field of view and associated beam-filling errors. Xie et al. (2016) also managed to associate distinct microphysical processes for rain formation like coalescence, size-sorting and riming/aggregation with the

20    measured polarimetric properties of the hydrometeors. These polarimetric fingerprints serve as very useful information for process understanding of rain formation and model validation (Trömel and Simmer, 2012)

Ground-based cloud photography provides the most detailed qualitative information on cloud patterns at high spatial and temporal resolution. Consequently, up to six sky imagers were operated in the SKY network during HOPE-Jülich. The combination of several imagers allows also for a quantitative retrieval of the spatial cloud structure. Beekmans et al. (2016)

25    presented an approach for a spatial cloud reconstruction by using two hemispheric sky imagers in a stereoscopic setup. They combined a dense stereo correspondence technique and a large-scale stereo setup to derive 3D cloud geometries. Obviously, such a stereoscopic cloud reconstruction is best suited for convective clouds that exhibit strong 3D spatial features. Important aspects of such a technique include an accurate camera calibration (internal projection and camera orientation in space), precise synchronization, similar radiometric properties, and successful stereo matching on the rather fuzzy (diffuse) cloud images. As

30    an example, Figure 15 shows the determination of a cross section (panel d) from a reconstruction from a cumulus cloud (panel a). It was found that the near-zenith cloud base height is very well reproduced in comparison to lidar observations, yielding errors between five to ten percent for low to mid-altitude cumuliform clouds. In general, Beekmans et al. (2016) provided a

complete approach including geometric and radiometric corrections to obtain the spatial cloud envelope geometry for the cloud sides facing the sky imagers. Together with 3D cloud information from scanning active systems such data will be very valuable for cloud reconstruction and radiation closure studies.

**5  4    Application of HOPE observations in modelling activities**

In the previous section, results of the HOPE observations were presented by means of a summary of the different studies covering a large range of meteorological processes from land-surface-atmospheric boundary layer exchange, cloud and precipitation processes to the sub-grid variability and microphysical properties of clouds and precipitation. Within this section the application of these results for the evaluation of the newly developed ICON model in LES mode but also to other LES and

10  small-scale GCMs will be summarized. A detailed overview about the setup of the different models can be found in Heinze et al. (2017). In general, ICON was run in LES mode on a daily basis. Thus, usually the model was initialized at 00 UTC and calculations were performed for a period of 24 hours. The lateral boundaries for the ICON runs were provided by the COSMO-DE model (Baldauf et al., 2011), which is one of the operational models of the German Meteorological Service (DWD). Within the boundaries of COSMO-DE, covering full Germany and the Netherlands as well as parts of the other neighbouring countries,

15  three ICON domains, only slightly smaller than the COSMO-DE domain (47.6° N –54.6° N, 4.5° E –14.5° E), are nested, having horizontal resolutions of 625 m, 312 m, and 156 m, respectively, and a vertical resolution of 150 layers within 21 km of height above ground. The simulation of 1 day takes approximately 12 days when run on 7200 computing cores and creates 50 TB of output data. LES runs of other models at spatial resolutions in the range of 50 m were reduced to smaller areas around the HOPE-Jülich region and periodic boundary conditions were applied to these models. Those were the models ICON-SI

20  (ICON semi idealized), PALM (PArallelized Large-eddy simulation Model 4.0; Maronga et al. (2015)) and DALES (Dutch Atmospheric LES; (Heus et al., 2010)).

Given the requirements on computational time and storage space the simulation days were chosen according to the appropriateness of the present weather conditions for the evaluation goals. A list of the HOPE days for which ICON runs are already available is provided in Table 5. It should be noted that the number of modelled HOPE days is subject to change in the

25  future and that ICON runs for dates not covered by HOPE were also already performed but are not shown in here. The HOPE days selected for ICON runs cover a wide range of meteorological conditions, from clear-sky days for the evaluation of convective processes in the planetary boundary-layer to days on which frontal passages accompanied by large-scale precipitation occurred. Most evaluation efforts were so far performed in a study of Heinze et al. (2017), but also others already made use of the extensive observational dataset. The studies available so far are discussed below.

**4.1 Examples of model-observation inter-comparisons**

The observational studies presented in Section 3 demonstrate well that large efforts are being taken to make observations suitable for the initialization and the evaluation of numerical weather prediction (NWP) models and to provide process studies that are essential for their improvement. The high temporal resolution of the HOPE dataset allows an analysis beyond the mean, which offers new opportunities to improve the simulation of boundary layer dynamics. Vertical profiles of higher-order moments (variances and turbulent fluxes) can be derived (Behrendt et al., 2015; Van Weverberg et al., 2016) which are essential to advance higher-order closure parameterizations of turbulent transport schemes in numerical models. Recent large-eddy simulation studies analysed the underlying sources and sinks of such prognostic higher-order moment equations for the cloud topped boundary layer (Heinze et al., 2015) and precipitating shallow cumulus regime (Schemann and Seifert, 2017). While these studies underline the importance, more robust conclusions are achieved by combining synoptically realistic model simulations with accompanying observational studies.

Nevertheless, operating a forecast model at scales that are small enough to resolve the different supersites of the HOPE-Jülich campaign puts certain requirements on the capabilities of the model. When the model resolution is between large-eddy simulations (with resolved energy-containing turbulence) and mesoscale simulations (no turbulence resolved), the model is operating in the so-called "grey zone" where more-sophisticated physical parameterizations (e.g. for boundary-layer turbulence or cloud microphysics) might be needed. To what extent the parameterization of turbulence and shallow convection is still necessary has been one of the key subjects of HD(CP)². Based on HOPE-Jülich observations, the grey zone was investigated in a study of Barthlott and Hoose (2015) who performed simulations with the COSMO model at horizontal resolutions ranging between 250 m and 2.8 km for six HOPE IOPs and one additional summertime case of the same year of 2013. From the kinetic energy spectra derived from the model output, it was found that the effective resolution (the minimum size of resolvable eddies) lies between 6 and 7 times the nominal resolution. Finer resolutions improved the representation of boundary-layer thermals, low-level convergence zones and gravity waves, but the effect on the temporal evolution of mean precipitation was rather weak. However, due to sensitivities of the rain intensities to model resolution, differences in the total rain amount of up to +48% occurred. Whereas the location of rain was rather similar at all model resolutions for the springtime cases of HOPE with moderate to strong synoptic forcing, the summertime case with airmass convection showed strong differences between the different resolutions with better agreement to the observed precipitation amount at the highest resolution of 250 m.

A major goal of HD(CP)² has been to use high-resolution modelling to derive parametrizations for climate models and general circulation models. In this respect the vertical cloud overlap parametrization is of high interest as it strongly influences the distribution of energy. In the past, such parametrizations have only been tested against observations on a global scale or for deep convective clouds. For the first time, Corbetta et al. (2015) investigated cumuliform cloud overlap for several boundary layer cloud cases including HOPE and compared it with the results from LES runs of the DALES model. Gridded time-height data from Cloudnet were used to derive cloud fraction masks at various temporal and vertical resolutions. The authors investigated the overlap ratio, i.e., the ratio of the cloud fraction by volume to the vertically averaged cloud fraction by area of

a grid box, as a function of the vertical resolution of the grid box. Cumuliform-cloud overlap ratios were found considerably underestimated by the LES model. For model-layer depths of less than 100 m, the modelled cloud overlap deviated by less than 7% from the observed one. The difference gradually increases to 15% for layer depths of 500 m and approached 20% for larger layer depths. Stratiform clouds were found to be better reproduced by the model, compared to cumuliform clouds.

5    Interestingly, the simulated and observed decorrelation lengths found for this type of clouds are smaller (~300 m) than previously reported (>1 km). The authors conclude that the inefficient overlap found at large vertical scales has the potential of significantly affecting the vertical transfer of radiation in large-scale GCM, because usually volume and area cloud fractions are assumed to be identical. The study can thus help to improve corresponding sub-grid parametrizations.

The evaluation of actual LES simulations of the HOPE-Jülich area was done by Heinze et al. (2016) who performed simulations

10    with PALM and UCLA-LES (University of California, Los Angeles Large-Eddy Simulation model, Stevens et al. (2005)) at up to 50 m horizontal resolution over the HOPE domain for a 19-day time period in order to capture a variety of different atmospheric and especially boundary layer conditions. The general weather pattern was reproduced in 80% of the cases. Also cloud types usually agree well with observations. Resulting turbulence characteristics and boundary layer heights have been compared to observations from active remote sensing (Doppler lidar and aerosol lidar) and from in-situ radiosonde

15    observations as proposed by Schween et al. (2014). Figure 16 exemplarily shows the temporal evolution of the boundary layer height as derived from different model runs and from observations. The 2-hour (12-14 UTC) mean boundary layer depth derived with the PALM model agreed within 400 m to the different observation methods and to the COSMO-DE run at 2.8-km resolution. The found differences are pointing to problems in the representation of ABL features in the LES, and should be subject of further investigations. Please note that the criterion of model-based ABL depth is also subject to uncertainties

20    which is explained further by Milovac et al. (2016) who found similar deviations between measurements and observations as found by Heinze et al. (2016). Heinze et al. (2016) further compared the observed turbulence characteristics of the ABL with the LES model. Observed and modelled profiles of the vertical-velocity variance agreed in their shape with the modelled values being in the range of uncertainty of the observations and showing slightly higher values throughout the boundary layer. Modelled profiles of potential temperature variances were found to be lower than the TRRL observations. For humidity

25    variance, agreement within the uncertainty range was found in the lower and mid-CBL between measurements and LES models. But the modelled variance peaks at the CBL top showed an under-estimation when compared with observations. Significant differences with respect to results from coarser resolved COSMO simulations were not reported. This might in part be due to the so-called semi-idealized set up with periodic boundary conditions and a homogeneous surface forcing. The authors also conclude that the simulated longwave and shortwave surface fluxes simulated with the LES model can be seen as

30    representative in comparison to respective observations at 5 different sites in the HOPE area. The peak shortwave heat flux in the LES and COSMO-DE tends to be overestimated compared to the weighted average, whereas the longwave heat flux tends to be underestimated.

Furthermore, within the Synthesis Module of HD(CP)², high-resolution ICON runs with 625-m, 312-m, and 156-m resolution were extensively evaluated against datasets collected during HOPE-Jülich and from other sources (Heinze et al., 2017). It was

found that the highest-resolved ICON-LES model matches much better the observed variability at small- to meso-scales than the coarser-resolved model runs or the reference model COSMO-DE with its 2.8-km horizontal resolution. It was demonstrated that the simulated turbulence profiles of the vertical velocity approach the observed ones for an increase in the ICON horizontal resolution from 625 m to 156 m. Differences between observed and modelled variance profiles of potential temperature and

5    specific humidity were much larger, which was explained by the absence of surface and soil moisture inhomogeneity in the model setup. The integrated water vapour of all models matched the range of values from the observations, but the temporal variability at short timescales as it was observed with microwave radiometer on a 1-s basis was only reproduced by the 156-m resolution run of ICON. From direct comparisons between modelled and continuous ground-based observations of the cloud field during HOPE-Jülich it was however found that convective boundary layer clouds are under-represented in the model,

10   even though the evaluation of the cloud fields on a larger scale, i.e. in comparison to satellite observations, showed that clouds are well represented in the model.  Heinze et al. (2017) concluded that, despite the given potential for further improvement of the ICON-LES model, it already fits well to the purpose of using its output for parameterisation development.

Regarding the application of HOPE observations for the initialization of NWP models, a first attempt was recently reported by Adam et al. (2016) who concentrated on the 24 April 2013 (IOP 6). In their study the authors assimilated lower-tropospheric

15   temperature profiles from the TRRL, reaching from about 500 to 3000 m above ground, into the Weather Research and Forecasting (WRF; Skamarock et al. (2008)) model using a 3D-variational method (Barker et al., 2004). The WRF model was covering Central Europe with 57 vertical levels and 3-km horizontal resolution. The assimilation of the temperature profiles from the TRRL in addition to the assimilation of conventional data including zenith total delay integrated water vapour field from the Global Navigation Satellite System and operational radiosonde data were found to improve the agreement of measured

20   boundary layer height and temperature gradient to the modelled values. Nine hours after the assimilation of TRRL data was initialized, already an area of 100 km in radius around the HOPE-Jülich area was affected, showing a temperature deviation from the conventional run of up to 2.5 K at 2.5 km height above sea level. Similar impacts can also be expected for the assimilation of profiles of water vapour mixing ratio from continuous lidar observations, as was found in an earlier study of Grzeschik et al. (2008).

25   **5    Summary & conclusions**

[revised manuscript text omitted]

5 First evaluation studies based on HOPE data have shown general agreement between observed and modelled boundary layer height, turbulence characteristics, and cloud coverage, but also point to significant differences that deserve further investigations, both from the observational and from the modelling perspective. Although the meteorological conditions which were prevalent during HOPE-Jülich and HOPE-Melpitz enabled the collection of a broad set of observations, it is obvious that the experimental coverage of the atmospheric boundary layer requires ongoing measurement efforts. In particular the

10 continuous observations from the German supersites will contribute to these efforts. The supersites JOYCE, KIT, and LACROS that have been deployed during HOPE-Jülich continue their long-term measurements at their base institutes and will contribute to further process and model evaluation studies in conjunction with further national and international supersites like Barbados (13.2° N, 59.4° W), Cabauw, the Netherlands, (51.9° N, 4.9 ° E), Lindenberg, Germany, (52.2° N, 14.1° E), Zugspitze mountain, Germany, (47.4° N, 11° E) as well as mobile facilities from the US (ARM) and Germany (mobile deployments of

15 the KIT cube, LACROS) under specific climatological and meteorological conditions.

Future work will take advantage of the synergy of the different active and passive remote sensing measurements. For instance, Doppler lidar and polarimetric radar measurements may link dynamical forcing (up and downdrafts) with microphysical processes (riming, coagulation, ice formation). The cloud radars of JOYCE, KITcube and LACROS were occasionally operating in a synchronized scan mode. Together with vertically pointing and scanning microwave radiometer data, three-

20 dimensional distributions of cloud liquid water may be constructed, and may get even further refined from cloud structure stereoscopy from synchronized sky imager data. Radiation closure studies will be performed based on observed and modelled spatial cloud structures and observed surface radiation budget measurements. High-resolution irradiance data can be used to build stochastic irradiance simulators for specific cloudy sky conditions, which in turn can be used to construct realistic cloud induced solar radiation variability. Combined measurements of temperature, humidity, and vertical wind fluctuations in the

25 PBL under different meteorological conditions will provide important statistical information for improved turbulence parameterizations. HOPE also demonstrated the future potential of the synergy of scanning wind, temperature, and water-vapour lidar systems for 3D studies of land-atmosphere exchange and ABL entrainment in heterogeneous terrain. HOPE data may also reveal to what extent variations in aerosol concentrations and thus in CCN and IN concentrations have an effect on cloud and ice formation compared to dynamical forcing.

30 In future, HOPE data will continue to contribute to the development, evaluation and improvement of high-resolution NWP and LES models because the data will be available via the Standardized Atmospheric Measurement Data (SAMD) data base which fulfils the needs of in particular model experts. Focused on the ICON development and on the collection of observational data for model evaluation, Phase 1 of HD(CP)² set the starting point for an ongoing, synergistic use of HOPE and other observational data by the modelling community. In Phase 2 of HD(CP)², which started in 2016, HD(CP)² participants are

already making use of these observations. For instance, a project on boundary layer clouds will confront ICON with HOPE data for different cloud regimes at different spatiotemporal scales. A project addressing fast cloud adjustment to aerosols will exploit remote-sensing and in-situ observations of aerosol and cloud properties to evaluate the susceptibility of the model performance to different representations of aerosol in the model, e.g., to variations in the concentration of nuclei for cloud

5    droplets or ice crystals. A project on the effects of surface heterogeneity e.g. uses the HOPE observations to challenge the applicability of the Monin-Obukhov Simularity Theory (MOST) and the reproduction of the vertical boundary layer structure and turbulence on small scales. Other projects apply the observations of the 3D water vapour fields and the cloud microphysical properties derived with Cloudnet for the development of convection parameterizations, just to mention a few.

Thanks to the valuable efforts of the community of observers during the HOPE campaigns and given its open-access

10    availability in the SAMD database (See Sect. 2.2.3) the HOPE dataset can serve as excellent tool for the model evaluation and initialization community.

**Acknowledgements**

15    The work summarized in this review was mainly carried out in the project HD(CP)$^2$ funded by the German Ministry for Education and Research. We specifically acknowledge the HD(CP)$^2$ projects 01LK1212A (University of Hohenheim), 01LK1209D (University of Leipzig), 01LK1209B (FZJ), 01LK1209C and 01LK1212C (TROPOS), 01LK1212F and 01LK1204B (KIT), 01LK1219A and 01LK1210A (University of Bonn), 01LK1203B (University of Hannover), 01LK1203A (MPI Hamburg). We also refer to all acknowledgements in the publications cited in section 3.

20    HOPE is particularly grateful to the Research Center Jülich and RWE Power AG (Hambach) that provided generous logistic support during the Jülich campaign. We thank the Transregional Collaborative Research Centre 32 "Patterns in Soil-Vegetation-Atmosphere Systems - Monitoring, Modelling and Data Assimilation" for contributing their valuable rain observation research infrastructures to the Jülich campaign.

The universities of Cologne and Bonn as well as TROPOS secured intense radiosonde observations from internal budgets.

25    Raman lidar system BASIL were funded on the basis of a specific cooperation agreement between Scuola di Ingegneria - Università degli Studi della Basilicata, TROPOS and MPI Hamburg.

We appreciated the provision of four Sun photometers for HOPE-Jülich and one device for HOPE-Melpitz by Goddard Space Flight Center, Greenbelt, MD, USA.

[revised manuscript text omitted]

| Date | IOP | Weather conditions |
|---|---|---|
| 20.04.13 | IOP 5 | Clear sky with only some cirrus clouds in the morning and late afternoon |
| 24.04.13 | IOP 6 | Clear-sky day with only few cirrus clouds in the morning and afternoon |
| 25.04.13 | IOP 7 | Cloudy morning (up to 4/8) until 10 UTC, only few clouds during noon, afterwards again increasing cumulus humilis cloudiness |
| 26.04.13 | IOP 8 | Rapidly increasing cloudiness up to complete overcast situation until noon, several rain showers and light to medium rain, decreasing cloudiness in the late afternoon |
| 02.05.13 | IOP 10 | Broken cumulus mediocris cloudiness, decreasing cloud cover during afternoon |
| 05.05.13 | IOP 12 | Clear-sky conditions until 09 UTC, afterwards slightly increasing cumulus humilis cloudiness up to (2/8) |
| 11.05.13 | - | High cloud cover until noon with several rain showers, afterwards broken cloudiness |
| 28.05.13 | IOP 18 | Clear sky conditions until midday (10 UTC) with only very few cirrus clouds, following low cumulus humilis clouds until 17 UTC, afterwards rapidly increasing cloudiness with rain starting in the evening |

[Figure]

**Figure 1: Setup of the HOPE-Jülich campaign showing the location of the three supersites Jülich (JUE), Hambach (HAM), and Krauthausen (KRA) as well as the outpost Wasserwerk (WAS) with their main instrumentation. The cones and arrows illustrate the field-of-view and scanning capabilities of the specific remote-sensing instruments.**

[Figure]

**Figure 2: Illustration of the setup of the HOPE-Melpitz campaign showing the deployed main instrumentation. The cones illustrate the field-of-view of the specific remote-sensing instruments.**

[Figure]

**Figure 3: Map of the spatial distribution of the measurement sites and networks deployed according to Table 1 (left) and a zoomed-in view centred at supersite Jülich (right). Background colours indicate the topography and dashed lines denote circles of constant distance from supersite Jülich (JUE). Shaded areas denote open-pit mines, for which the elevation map is not up to date.**

[Figure]

**Figure 4: Topography around the location of the HOPE-Melpitz campaign. (a) large-scale topography; (b) aerial photograph of the Melpitz field site with the locations of the pyranometers of the PYR network.**

[Figure]

**Figure 5: Spatiotemporal characteristics derived from the pyranometer network under broken-cloud conditions during HOPE-Jülich.** This figure illustrates the origin of deviations between a point measurement (labelled as var(TD) in the legend) and a domain-averaged value (representativeness error) for broadband solar atmospheric transmittance and irradiance for different domain sizes. (a) Power spectra of transmittance for a point measurement and domains with different sizes; (b) Explained variance of temporal fluctuations in a point measurement and a domain average as function of period, and (c) total expected deviation between a point measurement and a domain average for transmittance and irradiance as a function of averaging, assuming a value of 680 W m$^{-2}$ for the incoming solar irradiance at the top of atmosphere. The time period of fluctuations (inverse of their frequency) is shown logarithmically on the x-axis. Adapted from (Madhavan et al., 2016a).

[Figure]

**Figure 6: Integral scales of the temperature fluctuations (black) and humidity fluctuations (red) in the convective boundary layer derived from high-resolved temperature observations obtained between 1130 and 1330 on 20 April 2013 (IOP 5) and 1100 and 1200 UTC on 24 April 2013 (IOP 6) during HOPE-Jülich. Heights are normalized with respect to the height of the convective boundary layer height $z_i$. Adapted from Behrendt et al. (2015), Muppa et al. (2016), and from Di Girolamo et al. [2016].**

[Figure]

**Figure 7: Simultaneous observation of the vertical velocity variations in a stratocumulus layer performed in-cloud with ACTOS (red) and at cloud base with Doppler lidar WiLi of LACROS (blue) on 22 September 2013 during HOPE-Melpitz. The mean vertical velocity of both observations was set to zero to correct for large-scale vertical motions. Adapted from Seifert et al. (2017).**

[Figure]

**Figure 8: Observation of the integrated water vapour (IWV) during HOPE-Jülich for a large suite of different instruments. Right panel shows the frequency distribution of the IWV values recorded with the different techniques. Bottom panel shows the accumulated amount of precipitation. Adapted from Steinke et al. (2015).**

[Figure]

**Figure 9: Calibrated nighttime observations at KRA of the water vapour mixing ratio for April 2013 during HOPE-Jülich obtained from Polly$^{XT}$ that were calibrated automatically with the integrated water vapour provided by a co-located microwave radiometer. Adapted from Foth et al. (2015).**

[Figure]

**Figure 10: Aerosol target classification for the HOPE-Jülich period from 24 to 26 April 2013 (IOPs 6-8) based on continuous observations of the multi-wavelength polarization lidar PollyXT. The methodology is described in Baars et al. (2017).**

[Figure]

**Figure 11: Correlation between the particle extinction coefficient derived from Mie modelling and hygroscopic-growth correction of in-situ measurements of ACTOS with the respective ones measured with PollyXT. The data set is based on 13 data points obtained at different altitudes during two ACTOS flights on 14 and 17 September 2013 during HOPE-Melpitz. Adapted from Düsing et al. (2017).**

[Figure]

**Figure 12: Stratocumulus observation at the Melpitz site on 22 September 2013. Time-height cross sections of (a) cloud droplet effective radius and (b) liquid water content as observed from ground-based remote sensing with LACROS. (c-d): Profiles of single data points, mean, and standard deviation (horizontal bars) of (c) liquid water content and (d) effective radius as observed in-situ with ACTOS (red) a and retrieved from LACROS (black) for the time period shown in (a) and (b). Scaled-adiabatic method is based on Merk et al. (2016), Frisch-2002 method is based on Eq. 5 of Frisch et al. (2002).**

[Figure]

**Figure 13: Relationship between mean ice water content (IWC) and ice-to-liquid mass ratio as a function of cloud top temperature of all thin supercooled stratiform clouds detected during HOPE-Jülich. The colours represent the different radar linear depolarization ratios.**

[Figure]

**Figure 14: Time series of rain rates derived from observations of seven disdrometers (including those from the TR32 program) and the three polarimetric radars on 29 May 2013. The shaded grey area indicates the range of rain rates observed by the disdrometers with 1 min temporal resolution in the HOPE area, while the rain rate from the three polarimetric radar observations is calculated at the radar gates that are coincident with disdrometer locations and also averaged over the disdrometer locations. From Xie et al. (2016).**

[Figure]

**Figure 15: 3D reconstruction of a cumulus tower from a stereographic photograph from 24 July 2014, 11:32:00 UTC. Shown are (a) subsection of the image obtained from the reference camera, (b) the reconstruction as an untextured triangulated surface mesh, (c) the color-coded height of the reconstruction with contour lines, and (d) reconstructed distance of the cloud edges from the reference camera obtained along the cross-section (dashed line) shown in (a), (b), and (c) as well as a comparison of the cloud base with the one observed with lidar ceilometer (blue line). Adapted from Beekmans et al. (2016).**

[Figure]

**Figure 16: Temporal evolution of the boundary layer depth $z_i$ for the period from 24 to 30 April 2013. $z_i$ is determined by means of the bulk-Richardson number criterion in all three models (PALM, UCLA-LES and COSMO) and in the radiosonde data. A criterion based on the vertical velocity variance and detected aerosol layers is used for the wind lidar and aerosol lidar PollyXT, respectively. The data point obtained from the temperature rotational Raman lidar (TRRL, rot. Raman lidar) is based on Behrendt et al. (2015). Radiosondes were launched at the KITcube site, the Doppler lidar and Polly[XT] took measurements at sites JUE and KRA, respectively. Grey and green shading denote twice the standard deviation of $z_i$ in PALM and UCLA-LES, respectively. Adopted from Heinze et al. (2016).**

---

## Author Comment (AC2) · 7 Feb 2017

**Review of Macke et al., ACPD, 2016**

This paper presents an overview of the HD(CP)$^2$ experiment and summaries of the associated papers that report on results from the campaign. It provides a useful overview of the available data and model runs that were performed. The paper is suitable for publication provided that a few corrections are made. In places the summaries of the papers do not give enough detail about the work being talked about (see line-by-line comments below). Some suggestions for additions are a table of the model runs performed detailing type of model, resolution, etc. Also, it would be useful to quote the uncertainties of the instruments listed in Table 1. There are a lot of statements along the lines of "xx agrees well with yy", but little quoting of quantitative agreement (e.g. within x %).

We thank Referee #2 for his comments and suggestions for improving the manuscript. Below we provide replies separately for every comment of the Referee. First we list below the three comments that appeared in the introductory text to this review.

1)
In places the summaries of the papers do not give enough detail about the work being talked about (see line-by-line comments below).
Thanks for pointing us in the line-by-line comments to the specific positions in the text where more information seems to be necessary. We went through these points and extended the descriptions in the text, where requested.

2)
Some suggestions for additions are a table of the model runs performed detailing type of model, resolution, etc.
The lack of information about the available model data was also addressed by Referee #1. We thus added in cooperation with the HD(CP)² modelling community an extensive introduction to Section 4 (P19) which provides a summary of the currently available model runs for the HOPE periods, concentrating on ICON.

3)
Also, it would be useful to quote the uncertainties of the instruments listed in Table 1. There are a lot of statements along the lines of "xx agrees well with yy", but little quoting of quantitative agreement (e.g. within x %).
The uncertainties of the instruments depend on a multitude of parameters which we are not able to address in detail neither in Table 1 nor in the text of the manuscript. We have to rely on the information provided with the instruments and the applied retrievals given in the references listed in Table 1. In addition, also in Section 2 we already point to related articles that provide details on specific retrievals. To emphasize the importance of the specific description of each instruments, we added a text passage right after the introduction of Table 2 (P5, L28-32):
*"Concerning technical details of the individual instruments, such as instrument calibration and stability, restrictions in the instrument resolution, or the assessment of uncertainties, we refer the reader to the literature cited in Table 2. In addition, results shown in Sect. 3 and 4 of this article are based on already published articles which are cited at the respective positions in text and contain detailed information on the applied instrumentation and methodologies."*

We also admit that we provided too little quantitative information at some positions in the text and (also in agreement to comments of Referee #1) aimed to improve this in the revised version of the manuscript.

**Line-by-line comments**

Abstract
HOPE-Jülich instrumentation included a radio sounding station, 4 Doppler lidars, 4 Raman lidars (3, 3, and 4 of these provide temperature, water vapor, and particle backscatter data, respectively),
**The "3, 3 and 4" part of this sentence doesn't make sense to me since there is only one radio sounding station mentioned.**
The numbers only correspond to the capabilities of the Raman lidar systems. We made this clearer by changing the sentence to (P1-L37ff): *"HOPE-Jülich instrumentation included a radio sounding station, 4*

*Doppler lidars, 4 Raman lidars (3 of them provide temperature, 3 of them water vapour and all of them particle backscatter data), 1 water-vapour differential absorption lidar…"*

p.2, L15
The newly developed convection-resolving HD(CP)$^2$ model will be used to develop new convection parameterizations for large-scale eddy simulation models.

**The term "large-scale eddy simulation models" could lead to confusion with "large eddy simulation" models, or is this what you mean? Although, in that case might it not also be used to develop parameterizations for coarser resolution models? If you mean lower resolution models then it would be better to describe the types of model that you are talking about in a different way - e.g. mesoscale models, GCMS, or what the resolution range is perhaps. Or maybe this sentence is not even needed here since there is perhaps better explained at the start of p.3.**

The corresponding text section led to confusion and we revised it to accentuate that the idea behind ICON is to develop parameterizations for coarser resolution models. We now state: P2, L18-22: *"The newly developed convection-resolving HD(CP)2 Icosahedral non-hydrostatic model (ICON) will be used to develop new convection parameterizations for future application in large-scale general circulation models (GCM) and climate models."*

p.3, L26
(Buehler and Russchenberg, 2016)

**Can't find this reference in the references.**

According to the citation rules of ACP it is apparently not possible to reference a whole (special) issue. We thus removed the reference.

p.5, L27
These instruments were complemented by a microwave radiometer which determines temperature and humidity profiles,

**Does it also give the cloud liquid water path? This would be worth mentioning if so.**

We added this information at the first place where the microwave radiometer is introduced (and also added the information about provided integrated water vapour). See P6, L7-9.

*"Amongst other instruments (see Löhnert et al. (2015)), JOYCE contributed to HOPE with observations of a continuously scanning 35-GHz cloud radar, a Doppler lidar, and three microwave radiometers (one continuously scanning, one vertically pointing, and one continuously obtaining temperature profiles) for the spatiotemporal characterisation of humidity and liquid water fields and for provision of the line-of-sight-integrated amount of water vapour and liquid water (Rose et al., 2005)."*

p.5, L29
an X-band rain radar was operated

**Since descriptions of what observations the other instruments can give are listed here it would be good to do the same for the X-band radar.**

We added that the x-band radar provide information about the precipitation field around the HOPE-Jülich area. P6, L32-P7,L1: *"At a second KITcube outpost denoted KiXPol, approximately 7.5 km southwest of HAM, a polarimetric X-band rain radar was operated, providing volume scans of polarimetric moments, vertical cross-sections (RHI-scans) on demand, as well as the horizontal precipitation field for the HOPE-Jülich area every 5 minutes and with 250-m radial resolution."*

p.7, L7
The follow-up campaign HOPE-Melpitz has become necessary…

**Should be "The follow-up campaign HOPE-Melpitz became necessary…"**

Corrected

p.10, L14

for 3 hourly and second-resolution observations, respectively

**Please make it clear whether you mean 1-second or 3-second resolution for the latter part of the sentence.**

We changed the sentence to (P11,L22-23) *"Broken clouds cause the largest deviations in the 10x10 km²-sized domain, reaching about 30 W m$^{-2}$ for 3-hourly and 80 W m$^{-2}$ for 1-s-resolution observations."*

p. 10, L20

"and showed that these increments are more strongly averaged out in space than the transmissivities themselves"

**It's not quite clear what you mean by "more strongly averaged out".**

We changed the text for clarification (P11, L33): "…*and showed that the magnitude of increments is more strongly reduced by spatial averaging than that of the fluctuations in transmissivity themselves.*"

p.11, L18

"the integral scales"

**It would be useful to refer back to section 3.1 where "integral scales" are explained.**

Done (P13, L4)

p.11, L3 – "resolution" would be better than "scale" here

Done

p.11, L22 – "A general feature… decreases from the ground towards the top"

**This does not seem to be the case for Fig. 5a, only 5b. Can more detail be given about how common this was – e.g. in x % of the periods observed (since only two periods are shown in Fig. 5).**

We thank the referee for the comment and agree that this sentence is not true for Fig. 5a. Hence, we modified the sentence as follows (P13, L7ff):

*"A decrease in the integral length scale of the water vapour mixing ratio with height in the upper part of the CBL was found at the HAM site similar to previous observations (Couvreux et al., 2005;Wulfmeyer et al., 2010). A similar decrease was found for temperature at the same site. The temperature observations from JUE site show a more complex structure. The reasons for this are still under investigation."*

p.12, L3

The inter-comparison shown in Figure 6 presents a sequence and a histogram of the vertical velocities observed with ACTOS (red) and WiLi (blue). Thus, vertical velocities in the stratocumulus are similar at the cloud base (observed with the Doppler lidar) and cloud top (observed with ACTOS).

**There is only a histogram (no sequence as mentioned). Also, it would be good to do a statistical test to determine how similar the two distributions are. Also, what are the means values for each distribution?**

We removed the reference to the sequence (which was removed just before the initial submission of the manuscript). The mean values of both distributions are zero because the vertical velocities were adjusted in such a way that the mean value of the observation period is 0 m/s and also trends over the observation period were corrected. This is a standard procedure when turbulence characteristics are investigated (see e.g., Mallaun and Baumann, 2015). To clarify the applied procedure we updated the text from P13, L22 to L30 and also provide now the information that the cloud observation time was 15 minutes.

Concerning the request for a statistical test we consider that the good agreement with respect to the standard deviation but also regarding the found integral length scales (which requires the analysis of power spectra) sufficiently shows that the distributions agree well. More details on the shown observations will be discussed in an upcoming publication.

Reference:
Mallaun, C. and R. Baumann, 2015: Calibration of 3-d wind measurements on a single-engine research aircraft. Atmos. Meas. Tech., 8, 3177 – 3196, doi:10.5194/amt-8-3177-2015.

p.13, L 13-17

**Can more detail be given about the type of measurement this is? I.e. is it a profile, or a single surface measurement?**

Thanks for this helpful comment. We extended the addressed paragraph by more-quantitative information (See P15, L4-9). In addition, we put more information about the DIAL technique into Section 2.1.1 (subsubsection 'supersite HAM'), P7, L4-14.

p.14, L3-5 –

**How is it known that the large aerosol were emitted in the vicinity of the supersite? Also, how is it known that particle growth was observed rather than large aerosol being advected in? A little more detail of the evidence is required here.**

We included some more information on our reasoning concerning the source of the large non-spherical particles: P15, L29- L33: "*The occurrence of these aerosol types is correlated with the development of the planetary boundary layer and they first appear close to the ground and are slowly dispersed into the boundary layer in the course of the day, as can be seen for 24 and 25 April in Figure 10. Baars et al. (2016b) in addition present a case study that shows visual evidence of the dispersion of dust from the near-by open pit coal mine of Inden, west of the KRA site.*"

p.15, L4 – "Although the vertical profile of LWC agrees well with the in-situ and the remote-sensing observations, the deviation of the effective radius profile increases towards the cloud top"
**It seems that the LWC profile deviates more than the reff at cloud top.**

This concern was also addressed by Referee 1 and we provide here the same reply. We reformulated the corresponding paragraph and now provide more-detailed information on the intercomparison between in-situ and ground-based observations of the cloud microphysical parameters. In this manner, the corresponding Figure 12 was updated. In the revised version, Figure 12 also contains the time-height cross sections from the ground-based observations for the investigated time period. The profiles of liquid water content (lwc) and effective radius (re) were extended by the data points that were used to derive the statistics. The revised text (P17, L4-19) now presents more details on the flight pattern of ACTOS. Concerning the interpretation of the observed differences between in-situ and remote sensing, we provide possible reasons but leave a detailed analysis to an upcoming publication.

"*The implemented Frisch-2002 retrieval of cloud droplet effective radius and the Cloudnet retrieval of the adiabatically scaled LWC were evaluated against in-situ observations of ACTOS for a stratocumulus deck observed simultaneously by ACTOS and LACROS during the HOPE-Melpitz campaign on 22 September 2013 (IOP 22) from 09:59-10:16 UTC, as is shown in Figure 12. During the time period, ACTOS constantly flew horizontal legs of 2 km length in cross-wind direction in a distance of about 500-m upwind of the LACROS site. Time-height cross-sections from the continuous LACROS observations as shown in Figure 12 (a) and (b) will be available in the SAMD database (Sect. 2.2.3) for entire HOPE-Jülich and HOPE-Melpitz. The comparisons of the average vertical profiles of LWC and cloud droplet effective radius observed with ACTOS and retrieved with LACROS are shown in Figure 12. It can be seen that ACTOS probed mainly the mid-upper part of the cloud layer. Both, the observations of the LWC of the cloud droplet effective radius of ACTOS and LACROS (Figure 12a) are within the range of one standard deviation, as is shown by the horizontal error bars. Beside the found absolute differences, the profiles of LWC and effective radius retrieved from the LACROS observations deviate more strongly from those of ACTOS toward cloud top. A possible explanation for the observed discrepancies is the temporal variability of the LWC and effective radius in the cloud-top region as is shown in Figure 12 (a) and (b). Also, ACTOS was not flying directly above the LACROS site. Considering the applied retrieval of Eq. 5 in Frisch et al. (2002), also the assumption of a certain shape of the size distribution and of a cloud droplet number concentration can introduce biases. The application of the co-located observations of ACTOS and LACROS for the evaluation of ground-based retrievals will be discussed*"

*in an upcoming publication (Seifert et al., 2017)."*

        p.15, L5 – **How many profiles go into Fig. 11?**
We added a more detailed description of the observations used for Fig. 11 to the text on P16, L12-19. In addition, we also extended the discussion of the results to provide (as frequently requested by both referees) more quantitative information:
*'13 data points derived for different altitudes and conditions on 14 and 17 Sept. 2013 (see Table 2) are included in **Fehler! Verweisquelle konnte nicht gefunden werden.**. Each in-situ data point is based on all 120-s aerosol particle number size distributions recorded during a period of flight at a constant height. Averaging times for the lidar observations varied between 30 and 60 minutes. A linear relationship with significant R² values was derived between the modelled in-situ and remote-sensing extinction coefficients. For 355 nm 54% of all cases agree within the uncertainties and for 532 nm 55% of the cases. On average, the model underestimates the measured extinction coefficients for 355 nm by 3.5% and overestimates the measurements by 7.9% at a wavelength of 532 nm. Correlation coefficients are 0.944 and 0.947, respectively.'*

        p.15, L11-12 – **What instruments were used to determine the ice water content and ratios? More detail is needed here, rather than just citing Buhl (2016). Also, what location within the cloud are these values representative of? From the Figure 12 caption it seems they may be vertical averages over thin cloud layers. What criteria is used to determine thin clouds? What is the horizontal averaging period and what is the time period covered by the data?**
We now provide more information on the technique in the text.
*P17, L24-30: "In these plots the method for measurement of ice formation efficiency of Bühl et al. (2016) is used, which selects supercooled thin stratiform cloud layers with a turbulent mixed-phase (liquid-dominated) cloud top of a vertical extent of less than 380 m. In this way, non-linear ice formation effects like ice multiplication or splintering are avoided and, thus, do not affect the statistics. IWC is measured 60 m below the base of the mixed-phase layer, where an observation of the falling ice particles is possible without influence of water droplets or turbulent motions. LWC are mean values of the scaled-adiabatic approach (Merk et al., 2016) averaged over the complete height of the shallow mixed-phase top layer of the cloud where liquid water is present."*

        p.15, L16-17 – "can be explained with the increasing amount of adiabatically available liquid water with increasing temperature"
        **This could also be due to the increased concentrations of ice nucleating particles at colder temperatures, which should be mentioned. Can the analytically predicted change in the adiabatic condensation rate explain these large differences?**
This is an interesting suggestion. The authors of the study of Bühl et al. 2016 will in a future step definitely aim to address this question. With respect to the information given by Bühl et al. (2016) we can only speculate about the actual reason for the (rate of) increase of the ice-to-liquid ratio. We therefore decided to reduce the conclusion presented in the manuscript to the following statement (P17, L33): *"The plots quantify how ice formation becomes more efficient with decreasing temperature."*

        p.15, L31 – **Fig. 13 only shows data from one day - is it possible to quote statistics from more days, or the whole campaign. Correlation coefficients between the two measurements would be useful.**
We updated the respective position in the text with some more quantitative information (which we took from the article of Xie et al. 2016). P18,L14-16: *"The authors note that rainfall accumulations at the daily and even hourly scale were surprisingly consistent between the different observations of rain gauges, disdrometers and X-band radar, at least for the low-intensity rainfall events (of 0.5 – 20 mm day-1) prevalent during HOPE-Jülich. The correlation was found to be better than 0.93."*

p.16, L26-27 – "the model is operating in the so-called "grey

zone" where the parameterization of turbulence in the convective ABL may not be necessary"

**Will it not be the case that some form of sub-grid turbulence parameterization will always to be necessary even at fairly high resolution? What types of turbulence parameterizations are you talking about here? Also, the term "grey zone" is usually used to refer to the resolutions where convective parameterizations are no longer needed, but that are still too coarse to resolve the important eddies. Maybe this refers to a "grey zone" for the boundary layer parameterizations, or something else? But this needs to be made clear.**

The referee is right about the fact that a turbulence parameterization is still needed in the grey zone. Only at the scale of large-eddy simulations, the energy-containing turbulence is resolved. We rephrased the part of the text accordingly (P20,L13-18).

p.17, L9 "ratio of cloud fraction by volume and by area"

**Not quite sure what is meant here – can you please explain this a bit more?**

We revised the section addressed by the referee and specified more clearly what is meant by the overlap ratio. We also quantified the differences between modeled and observed overlap ratios and gave further information regarding the conclusions of the study. See P20, L21-P21,L8.

p.17, L21 "the turbulence characteristics of the ABL were captured satisfactorily"

**Can you please give more detail? Captured to within what margin of bias?**

See next comment.

p. 17 L24 – **as above – surface fluxes were reproduced to within what range?**

In agreement to Referee #1, Referee #2 also addressed the lack of quantitative information given in the discussion of the modelling results. We thus revised Section 4.1 and extended the description of the results and provide now also quantitative information about the correlation between models and observations. See, P20-P22.

p. 17, L27 – **what was the resolution of the coarser model?**

We added the name of the model (COSMO-DE) and provided the resolution and a reference to the model, see P21, L17 for Heinze et al. (2016) as well as P22, L2 (Heinze et al., 2017).

p. 17, L28-29 – **more detail is needed again – how closely did the turbulence profiles and integrated water vapour match and what timescales do you mean?**

As stated above, we revised the respective paragraph and now provide more quantitative information. (P22, L4-L12):

*"It was demonstrated that the simulated turbulence profiles of the vertical velocity approach the observed ones for an increase in the ICON horizontal resolution from 625 m to 156 m. Differences between observed and modelled variance profiles of potential temperature and specific humidity were much larger, which was explained by the absence of surface and soil moisture inhomogeneity in the model setup. The integrated water vapour of all models matched the range of values from the observations, but the temporal variability at short timescales as it was observed with microwave radiometer on a 1-s basis was only reproduced by the 156-m resolution run of ICON. From direct comparisons between modelled and continuous ground-based observations of the cloud field during HOPE-Jülich it was however found that convective boundary layer clouds are under-represented in the model, even though the evaluation of the cloud fields on a larger scale, i.e. in comparison to satellite observations, showed that clouds are well represented in the model. Heinze et al. (2017) concluded that, despite the given potential for further improvement of the ICON-LES model, it already fits well to the purpose of using its output for parameterisation development."*

p.18, L12 – **TTRL has not been defined yet.**

That was a typo which is now corrected to 'TRRL'

p.19, L6 – "ratio of ice to liquid water decreases with decreasing cloud top temperature"

**As the temperature got lower (colder) the ratio went up**

We apologize for the typo and replaced ,decreasing cloud top temperature' by 'increasing cloud top temperature'.

**Figures**

Fig. 1

**A higher resolution image would be better if possible. Also, the rain radar label is a bit unclear on left panel.**

We split Figure 1 into two figures to provide a higher resolution. Provision of a scalable vector graphic is not possible in the case of Figure 1 because this would increase the manuscript size too much (given the large amount of vectors used in the figure).

Fig. 2

Dashed lines denote circles of constant distance from supersite JUE

**This is repeated in the caption.**

Corrected. Please note, Figure 2 is now Figure 3.

Fig. 4

**Need to make it clear in the caption what the var($T_D$) line is in Fig. 4a. Also explain that the x-axis is the time period (or the inverse of the frequency).**

Please note, Figure 4 is now Figure 5. We have changed the caption as follows to address this point (insertions marked in bold):

*"Figure 4: Spatiotemporal characteristics derived from the pyranometer network under broken-cloud conditions during HOPE-Jülich. This figure illustrates the origin of deviations between a point measurement and a domain-averaged value (representativeness error) for broadband solar atmospheric transmittance and irradiance for different domain sizes. (a) Power spectra of transmittance for a point measurement* **(labelled as var($T_D$) in the legend)** *and domains with different sizes; (b) Explained variance of temporal fluctuations in a point measurement and a domain average as function of period, and (c) total expected deviation between a point measurement and a domain average for transmittance and irradiance as a function of averaging period, assuming a value of 680 W m-2 for the incoming solar irradiance at the top of atmosphere.* **The time period of fluctuations (inverse of their frequency) is shown logarithmically on the x-axis.** *Adapted from (Madhavan et al., 2016a)."*

Fig.5

**It looks like the line colour labels are the wrong way around for the water vapour and temperature.**

Thanks for the hint. It is now corrected. New Figure number : Fig. 6

Fig. 6

**In the caption, please mention LACROS (as in the legend) and WiLi (as in the manuscript text) for clarity.**

**Also, what are the mean values for the two distributions?**

**It would also be good to have a statistical test to show how similar the distributions are.**

We gave a reply to a similar point to p.12, L3 above.  New Figure number: Fig 7

Fig. 11**You should mention that ACTOS is in-situ data in the caption.**

Done (New Figure number: Fig 12.)

Fig. 12

**It would be better to plot this with temperature on the y-axis if possible.**

We would like to keep the current axis setup of Figure 12 (now Figure 13). The figure is supposed to be published as a 1-column figure and we thus aimed on having the same x-axes for both stacked plots.

**Typos**

p.4, L19 – "than it was the case for HOPE" - remove "it"
Done

p.9, L5 -  "All data will be public available by the end of 2016." – "All data will be publicly available by the end of 2016."
We updated the corresponding paragraph already for another reason (to provide more detailed information about the database)

p.11, L28 – move "also" to after "should"
Done

p.11, L29 – replace "higher resolved" with "more highly resolved" or "better resolved"
Done

p. 14, L2 – "Figure 9 exemplary" – better as "Figure 9 shows an example of…"
Done

p. 16, L2 – add "are" between "and" and "independent"
The corresponding part of the text was already modified for a different reason.

p.17, L4 – add comma after "in the past"
Done

p.17, L13 – "general weather patter where"
        -> general weather pattern was
Done

p.17, L16 – Schween reference incorrectly all in brackets.
Corrected

p.17, L19 – "subject of uncertainties" -> subject to uncertainties.
Done

p.18, L13 – "a key information" -> key information
Done

p.18, L21  - "constrain" -> constraint

Done

**The HD(CP)$^2$ Observational Prototype Experiment HOPE – An Overview**

[revised manuscript text omitted]

Within the M-module (modelling) of HD(CP)[2], the new ICON general circulation model was developed (Zängl et al., 2015) and its performance in LES modelling was evaluated (Dipankar et al., 2015). The O-module (Observations) was defined to provide observational datasets for the initialization and evaluation of the newly developed ICON model and other high-resolved LES models as well as for the development of new parameterizations that are suitable for application in a high-

5   resolution model. The scope of the S-module (synthesis) was to provide first improvements of parameterizations from the use of model and observation results. Key to this effort was the provision of modelled scenarios at 100-m grid resolution over thousands of kilometres, which will be used to analyse, improve or develop parameterizations related to cloud and precipitation development in climate models.

The O4 project in the O module of HD(CP)[2] was devoted to HOPE and has been designed to provide a critical model evaluation

10   at the scale of the model simulations and further to provide information on sub-grid variability and microphysical properties that are subject to parameterizations even at high-resolution simulations such as planned with ICON. Even for LES, unresolved sub-grid scale processes are believed to be in particular critical for cloud formation and the onset of precipitation, and thus built the central focus of HOPE. In order to derive the atmospheric state and the 3D fields of water vapour, temperature, wind and cloud and precipitation properties at the scale of 100-m resolution for an area of about $10x10x10$ km$^3$ three close-by

15   supersites, separated by a distance of approximately 4 km, complemented by larger networks were deployed. The instrumentation was selected in order to allow for detailed observations of the onset of clouds and precipitation in the convective atmospheric boundary layer. When compared to model results, the high-resolution HOPE data could elucidate to what extent a pure increase in model resolution improves model skills in the atmospheric boundary layer, and to what extent unavoidable parameterizations of physical processes - essentially turbulence and cloud microphysics - require new approaches.

20   HOPE complements the larger spatiotemporal Full-Domain (O2) and Supersites (O1) activities in the observations module in HD(CP)[2] of which O2 provides continuous time series of 2D fields across the HD(CP)[2] domain and O1 is devoted to the provision of 1D profiles at four dedicated locations in Germany and the Netherlands, respectively. The scope of Module O3 was to establish a data flow from the observation modules to the model and synthesis modules. In 2016, HD(CP)[2] entered its second phase, which puts a much stronger effort on the synthesis part.

25   HOPE builds on the experience gained in previous field campaigns like the Convective and Orographically-induced Precipitation Study (COPS) (Wulfmeyer et al., 2011), however, with a stronger focus on multi-sensor synergy covering a micro- to meso-scale domain. COPS and the associated General Observation Period (GOP) that was prepared in the context of the Quantitative Precipitation Forecasting priority program (SPP1167) of the German Science Foundation (DFG) (Crewell et al., 2008) aimed at the observation of orographically driven initiation of convection with supersites several tens of km apart

30   in strongly structured terrain. Complementary to COPS, HOPE is covering a smaller domain with higher resolution, and is accompanied by long-term supersite observations within the framework of the Terrestrial Environmental Observatoria (TERENO) Programme (Simmer et al., 2015) around the ground-based remote sensing supersite JOYCE (Löhnert et al., 2015), and the TROPOS long-term aerosol observatory in Melpitz (Spindler et al., 2012).

Although phase 1 of HD(CP)$^2$, lasting from 2012 to 2015, was mainly devoted to establish a scalable high-resolution ICON model and to obtain data for model evaluation at various scales, first highly resolved ICON-based LES have been performed to evaluate the effect of resolution on reproducing boundary layer fluxes and heights, as well as on cloud formation. First results are reported in this overview.

5 This article mainly serves as a guide through the sites and instrumentation used during the HOPE campaigns and it is aiming on giving a motivation to learn about the details and specific conclusions described in the individual publications this overview is built upon. The structure is as follows. Section 2 describes the site setups and measurements performed during HOPE including information about the meteorological conditions and data availability. Examples from each of the research topics are presented in section 3. In section 4, first comparisons between models and observations are discussed. A summary and

10 conclusions on the further applications of the HOPE data as well as designs for future observational strategies are presented in section 5. Individual work performed during HOPE is published in this ACP/AMT HOPE special issue or, in part, in other journals and is cited in the present overview correspondingly.

**2 Description of the HOPE field campaigns**

The technological aspect of HOPE was to unite most of the mobile ground-based remote sensing and surface flux observations

15 available in Germany within a single domain in order to capture the vertical structure and horizontal variability of wind, temperature, humidity as well as aerosol and cloud condensate with the best possible temporal and spatial resolution. Thus we were able to accommodate active remote sensing from lidar and radar, and passive remote sensing from microwave radiometer and Sun photometer, whenever possible with scanning capabilities. During HOPE, 3D water-vapour, temperature, and wind measurements were possible with unprecedented spatiotemporal resolution in the boundary layer. In order to understand the

20 forcing of and the response to surface properties, distributed surface flux and surface standard meteorological observations were deployed as well. Of course, it is not possible to obtain an instantaneous 3D picture of the atmosphere from a limited number of directional observations. However, ongoing improvements in sensor detection accuracy and optimized scanning strategies will capture the 4D boundary layer properties even better in the future.

[revised manuscript text omitted]

As the authors of the above-mentioned studies note, HOPE-Jülich provided for the first time data to observe the turbulence characteristics of the PBL, more specifically the convective boundary layer (CBL), up to the fourth statistical moment, i.e., the mean, standard deviation, variance, skewness, and kurtosis of the spatiotemporal water vapour and temperature. Examples of the relationship between the integral scales (introduced in section 3.1) of humidity and temperature fluctuation and height above ground within the CBL for the 20 April 2013 (IOP 5), 11:30-13:30 UTC, (only temperature fluctuations; see Di Girolamo et al. (2016)) and 24 April 2013 (IOP 6), 11:00-12:00 UTC (temperature and humidity fluctuation; see Behrendt et al. (2015) and Muppa et al. (2016)), respectively, are depicted in Figure 6. A decrease in the integral length scale of the water vapour mixing ratio with height in the upper part of the CBL was found at the HAM site similar to previous observations (Couvreux et al., 2005;Wulfmeyer et al., 2010). A similar decrease was found for temperature at the same site. The temperature observations from JUE site show a more complex structure. The reasons for this are still under investigation. The decrease of the integral length scale toward the top of the CBL can be explained by the decrease in the size of the turbulent eddies with height resulting from the entrainment of dry free-tropospheric air at the CBL top (Couvreux et al., 2005) which is also characterized by an increase in the variance of the temperature or water vapour toward CBL top. Converting the observed time scales shown in Figure 6 to spatial scales assuming horizontal and vertical wind velocities of 5 m s$^{-1}$ and 1 m s$^{-1}$, respectively, results in horizontal and vertical integral length scales of 100-1000 m and 20-200 m, respectively. Thus, in order to capture the full turbulence spectrum in the CBL, a numerical model simulation should also be run at temporal and spatial resolutions that are better resolved than the observed values.

Detailed convective boundary layer turbulence characteristics from HOPE and further field campaigns (Wulfmeyer et al., 2016) showed that the combination of active temperature-, humidity- and wind-profiling applied during HOPE-Jülich sufficiently resolves the turbulence structure of the CBL and lays the groundwork for new boundary layer turbulence parameterizations.

In addition to turbulent fluxes in the cloud-free planetary boundary layer, the turbulence characteristics of a stratocumulus layer were investigated simultaneously with ACTOS and the Doppler lidar WiLi of the LACROS site on 22 September 2013 during HOPE Melpitz. The inter-comparison shown in Figure 7 presents a histogram of the vertical velocities observed with ACTOS (red) and WiLi (blue), further insights into the microphysical properties of the cloud layer are given in Sec. 3.4 and Figure 12. The variability of the vertical velocities (with the mean adjusted to 0 m s$^{-1}$ and corrected for large-scale trends) during the cloud observation time of 16 minutes were found to be similar at the stratocumulus cloud base (observed with the Doppler lidar) and top (observed with ACTOS), with standard deviations of 0.23 m s$^{-1}$ for ACTOS and 0.21 m s$^{-1}$ for WiLi. This is an important fact for Doppler-lidar studies of stratocumulus clouds, because it implies that Doppler lidars are suitable to characterize the turbulence characteristics of entire stratocumulus cloud layers. From the vertical-velocity observations of WiLi and ACTOS also integral length scales were derived which were in the range from 38 m (ACTOS) to 45 m (WiLi). The observations will be further discussed in an upcoming publication (Seifert et al., 2017)

[revised manuscript text omitted]

Based on scanning measurements with the water-vapour DIAL of IPM made during HOPE-Jülich, Späth et al. (2016) (see Sect. 2.1.1) presented a detailed study of the 3-dimensional structure of the water vapour field between the supersites HAM, KRA, and JUE with a range resolution of 30-300 m and a temporal resolution in the range of 10 s for each profile. Full conical scans (360° in azimuth) around the site to characterize the water vapour field at a defined elevation angle took 15 min. Such observations provide valuable information for improving our understanding of land–atmosphere exchange processes as different types of land cover results in different evapo-transpiration and thus moisture in the CBL.

**3.4 Microphysical properties of aerosols and clouds**

The retrieval and evaluation of microphysical properties of aerosols, clouds, and precipitation from ground-based remote sensing observations is a crucial task. In-situ observations do provide much higher accuracy but for the long-term evaluation of the performance of operational weather forecast models and the microphysical parameterizations therein continuous datasets are required. In particular the HOPE-Melpitz campaign provided the opportunity to relate in-situ observations of warm-cloud microphysical properties and of aerosol properties from ACTOS to the respective parameters observed with ground-based observations of the LACROS facility. Case studies are presented in the following that document the simultaneous ground-based remote-sensing and in-situ observations of a stratocumulus layer and of the aerosol properties in the lower troposphere, respectively.

Aerosol particles act as nuclei for cloud droplets and ice crystals and are thus a prerequisite for the formation of clouds. Lidar is a promising tool to provide estimates of the concentration of cloud droplet condensation nuclei (CCN) and ice nucleating particles (INP) (Mamouri and Ansmann, 2016). During HOPE-Jülich and HOPE-Melpitz the Raman polarization lidar Polly[XT] was continuously operated to provide information on the vertical aerosol structure in the planetary boundary layer and the troposphere. HOPE-Jülich was the first time a Raman polarization lidar provided a continuous data set of the calibrated attenuated backscatter coefficient at three wavelengths. Amongst other parameters, the dominating type of aerosol particles present in each observed volume was derived by a newly developed target classification, as is explained by Baars et al. (2017). Figure 10 shows an example of the aerosol target classification for three consecutive days from 24 to 26 April 2013 (IOPs 6-8) during HOPE-Jülich. Frequently large, non-spherical particles, probably dust or pollen particles that were emitted in the vicinity of the site have been monitored. The occurrence of these aerosol types is correlated with the development of the planetary boundary layer and they first appear close to the ground and are slowly dispersed into the boundary layer in the course of the day, as can be seen for 24 and 25 April in Figure 10. Baars et al. (2017) in addition present a case study that shows visual evidence of the dispersion of dust from the near-by open pit coal mine of Inden, west of the KRA site. With increasing distance from ground, the particles frequently grow by hygroscopic growth, leading to the presence of large,

spherical particles, as it was the case on 25 and 26 April. The mask also helps to identify whether a cloud layer was within or detached from the planetary boundary layer aerosol. Overall, the classification of cloud particles solely on the lidar observations is difficult. This will be overcome in a future step, by merging the multi-wavelength aerosol classification with the Cloudnet target classification presented in Illingworth et al. (2007).

5   Retrievals of microphysical aerosol properties, such as CCN concentration, from lidar observations, as well as retrievals of the ambient scattering properties of an aerosol population measured in-situ are still subject to large uncertainties. In-situ observations of aerosol properties are usually performed under dry conditions and inlets are limited by a maximum cut-off size of an aerosol distribution. During HOPE-Melpitz, both in-situ aerosol observations as well as lidar observations of Polly[XT] were available. Figure 11 presents the relationship of the backscatter coefficient observed with Polly[XT] and the respective

10   extinction coefficient obtained from the in-situ aerosol observations of ACTOS as derived by Düsing et al. (2017). Based on the low-humidity (dry-state) in-situ aerosol measurements of ACTOS, the ambient extinction coefficient was obtained at wavelengths of 355, 532, and 1064 nm using a Mie model and a hygroscopic-growth correction. 13 data points derived for different altitudes and conditions on 14 and 17 Sept. 2013 (IOPs 20 and 21; see Table 4) are included in Figure 11. Each in-situ data point is based on all (and at least one) 120-s aerosol particle number size distributions recorded during a period of

15   flight at a constant height. Averaging times for the lidar observations varied between 30 and 60 minutes. A linear relationship with significant $R^2$ values was derived between the modelled in-situ and remote-sensing extinction coefficients. For 355 nm 54% of all cases agree within the uncertainties and for 532 nm 55% of the cases. On average, the model underestimates the measured extinction coefficients for 355 nm by 3.5% and overestimates the measurements by 7.9% at a wavelength of 532 nm. Correlation coefficients are 0.944 and 0.947, respectively. This shows that the ambient aerosol extinction coefficient can

20   well be derived from in-situ measurements given the extensive instrumentation for microphysical and chemical aerosol characterization that is available at the Melpitz field site.

During HOPE-Jülich the availability of CCN was investigated using an aerosol model. The approach presented by Hande et al. (2016) used the COSMO-MUSCAT model to simulate the generation and transportation of aerosols over Germany during the campaign. From the simulation results, a parameterisation of the CCN concentration was derived which can be applied

25   also to other climatological regions and different aerosol regimes. Even though the simulated aerosol properties were evaluated against in-situ observations of aerosol particle size distributions at Melpitz, no evaluation of the CCN parameterisation against measurements was performed. This emphasizes the need to improve remote-sensing techniques for the retrieval of CCN profiles as the one of  Mamouri and Ansmann (2016).

At the beginning of the first Phase of HD(CP)² no operational microphysical retrieval of the effective radius of cloud droplets

30   from ground-based remote sensing observations was available within the project. As a first step towards an evaluation dataset for numerical weather forecasts, it was decided to apply the retrieval technique of Frisch et al. (2002) to the LACROS observations by implementing it into the processing framework of Cloudnet. The technique is based on vertically-pointing measurements from a millimetre-wavelength cloud radar and a microwave radiometer and produces height-resolved estimates of cloud particle effective radius and liquid water content. In addition, liquid water content profiles are produced operationally

within Cloudnet (Illingworth et al., 2007), assuming either adiabatic profiles of liquid water content (LWC) between the lidar-derived cloud base and the radar-derived cloud top or scaled-adiabatic profiles for which the adiabatic liquid water content is scaled to fit the liquid water path observed with the microwave radiometer (Merk et al., 2016).

The implemented Frisch-2002 retrieval of cloud droplet effective radius and the Cloudnet retrieval of the adiabatically scaled

5 LWC were evaluated against in-situ observations of ACTOS for a stratocumulus deck observed simultaneously by ACTOS and LACROS during the HOPE-Melpitz campaign on 22 September 2013 (IOP 22) from 09:59-10:16 UTC, as is shown in Figure 12. During the time period, ACTOS constantly flew horizontal legs of 2 km length in cross-wind direction in a distance of about 500-m upwind of the LACROS site. Time-height cross-sections from the continuous LACROS observations as shown in Figure 12 (a) and (b) will be available in the SAMD database (Sect. 2.2.3) for entire HOPE-Jülich and HOPE-Melpitz. The

10 comparisons of the average vertical profiles of LWC and cloud droplet effective radius observed with ACTOS and retrieved with LACROS are shown in Figure 12. It can be seen that ACTOS probed mainly the mid-upper part of the cloud layer. Both, the observations of the LWC of the cloud droplet effective radius of ACTOS and LACROS (Figure 12a) are within the range of one standard deviation, as is shown by the horizontal error bars. Beside the found absolute differences, the profiles of LWC and effective radius retrieved from the LACROS observations deviate more strongly from those of ACTOS toward cloud top.

15 A possible explanation for the observed discrepancies is the temporal variability of the LWC and effective radius in the cloud-top region as is shown in Figure 12 (a) and (b). Also, ACTOS was not flying directly above the LACROS site. Considering the applied retrieval of Eq. 5 in Frisch et al. (2002), also the assumption of a certain shape of the size distribution and of a cloud droplet number concentration can introduce biases. The application of the co-located observations of ACTOS and LACROS for the evaluation of ground-based retrievals will be discussed in an upcoming publication (Seifert et al., 2017).

20 The accurate representation of the ice phase in numerical models is a crucial task since cold rain is the main driver of precipitation formation at midlatitudes (Mülmenstädt et al., 2015). The continuous observations of the LACROS supersite during HOPE-Jülich enabled to obtain statistical information about the primary ice production in stratiform midlevel mixed-phase cloud layers. Figure 13 shows an overview about the ice water content and ice-to-total mass ratio of all mixed-phase cloud layers that were identified from the HOPE-Jülich observations. In these plots the method for measurement of ice

25 formation efficiency of Bühl et al. (2016) is used, which selects supercooled thin stratiform cloud layers with a turbulent mixed-phase (liquid-dominated) cloud top of a vertical extent of less than 380 m. In this way, non-linear ice formation effects like ice multiplication or splintering are avoided and, thus, do not affect the statistics. IWC is measured 60 m below the base of the mixed-phase layer, where an observation of the falling ice particles is possible without influence of water droplets or turbulent motions. LWC are mean values of the scaled-adiabatic approach (Merk et al., 2016) averaged over the complete

30 height of the shallow mixed-phase top layer of the cloud where liquid water is present. As shown in Figure 13, the ice water content of clouds with top temperatures above -10 °C was in general lower than $10^{-4}$ g m$^{-3}$. At temperatures below -15 °C, values of the ice water content vary around $10^{-3}$ g m$^{-3}$. The ice-to-liquid mass ratio decreases from $10^{-2}$ to $10^{-5}$ for temperatures increasing from -40 to 0 °C. The plots thus quantify how ice formation becomes more efficient with decreasing temperature. The colour-coded data points in Figure 13 provide in addition the radar-observed linear depolarization ratio of the observed

ice particles, which is a proxy for the particle shape. Values of around -20 dBZ (-10 °C <T<-5 °C), -30 dBZ (-20 °C <T<-10°C), and -25 dBZ (T<-20 °C) indicate needle-like, dendritic, and bullet-rosette-like shapes, respectively (Bühl et al., 2016;Myagkov et al., 2016). Knowing about the relationship between ice water content, liquid water content, temperature, and shape of freshly formed ice crystals is an important step towards new approaches for the evaluation of ice formation schemes in numerical weather forecast models. This will also be a task of the second phase of HD(CP)[2].

**3.5 Macrophysical cloud & precipitation properties**

The combination of scanning polarimetric X-band Doppler rain radars, vertically pointing micro rain radars (MRR) and a ground-based network of disdrometers and rain gauges provided an excellent opportunity to validate the Doppler rain radar ability to infer the spatial variability of quantitative precipitation properties from polarimetric radar reflectivities. Xie et al. (2016) performed a detailed analysis of all precipitation observations under different synoptic conditions. As an example, Figure 14 shows a time series of the surface precipitation rates estimated from measurements of three Doppler rain radar compared to the in-situ observations from seven disdrometers (partly from TR32 and TERENO projects), averaged over the disdrometer locations. The authors note that rainfall accumulations at the daily and even hourly scale were surprisingly consistent between the different observations of rain gauges, disdrometers and X-band radar, at least for the low-intensity rainfall events (of $0.5 - 20$ mm day$^{-1}$) prevalent during HOPE-Jülich. The correlation was found to be better than 0.93. The two near-by radars (KiXPol and JuXPol) showed slightly better agreement than the 50 km remote radar BoXPol, which is explained by its correspondingly larger field of view and associated beam-filling errors. Xie et al. (2016) also managed to associate distinct microphysical processes for rain formation like coalescence, size-sorting and riming/aggregation with the measured polarimetric properties of the hydrometeors. These polarimetric fingerprints serve as very useful information for process understanding of rain formation and model validation (Trömel and Simmer, 2012)

Ground-based cloud photography provides the most detailed qualitative information on cloud patterns at high spatial and temporal resolution. Consequently, up to six sky imagers were operated in the SKY network during HOPE-Jülich. The combination of several imagers allows also for a quantitative retrieval of the spatial cloud structure. Beekmans et al. (2016) presented an approach for a spatial cloud reconstruction by using two hemispheric sky imagers in a stereoscopic setup. They combined a dense stereo correspondence technique and a large-scale stereo setup to derive 3D cloud geometries. Obviously, such a stereoscopic cloud reconstruction is best suited for convective clouds that exhibit strong 3D spatial features. Important aspects of such a technique include an accurate camera calibration (internal projection and camera orientation in space), precise synchronization, similar radiometric properties, and successful stereo matching on the rather fuzzy (diffuse) cloud images. As an example, Figure 15 shows the determination of a cross section (panel d) from a reconstruction from a cumulus cloud (panel a). It was found that the near-zenith cloud base height is very well reproduced in comparison to lidar observations, yielding errors between five to ten percent for low to mid-altitude cumuliform clouds. In general, Beekmans et al. (2016) provided a

complete approach including geometric and radiometric corrections to obtain the spatial cloud envelope geometry for the cloud sides facing the sky imagers. Together with 3D cloud information from scanning active systems such data will be very valuable for cloud reconstruction and radiation closure studies.

**4    Application of HOPE observations in modelling activities**

In the previous section, results of the HOPE observations were presented by means of a summary of the different studies covering a large range of meteorological processes from land-surface-atmospheric boundary layer exchange, cloud and precipitation processes to the sub-grid variability and microphysical properties of clouds and precipitation. Within this section the application of these results for the evaluation of the newly developed ICON model in LES mode but also to other LES and small-scale GCMs will be summarized. A detailed overview about the setup of the different models can be found in Heinze et al. (2017). In general, ICON was run in LES mode on a daily basis. Thus, usually the model was initialized at 00 UTC and calculations were performed for a period of 24 hours. The lateral boundaries for the ICON runs were provided by the COSMO-DE model (Baldauf et al., 2011), which is one of the operational models of the German Meteorological Service (DWD). Within the boundaries of COSMO-DE, covering full Germany and the Netherlands as well as parts of the other neighbouring countries, three ICON domains, only slightly smaller than the COSMO-DE domain (47.6° N –54.6° N, 4.5° E –14.5° E), are nested, having horizontal resolutions of 625 m, 312 m, and 156 m, respectively, and a vertical resolution of 150 layers within 21 km of height above ground. The simulation of 1 day takes approximately 12 days when run on 7200 computing cores and creates 50 TB of output data. LES runs of other models at spatial resolutions in the range of 50 m were reduced to smaller areas around the HOPE-Jülich region and periodic boundary conditions were applied to these models. Those were the models ICON-SI (ICON semi idealized), PALM (PArallelized Large-eddy simulation Model 4.0; Maronga et al. (2015)) and DALES (Dutch Atmospheric LES; (Heus et al., 2010)).

Given the requirements on computational time and storage space the simulation days were chosen according to the appropriateness of the present weather conditions for the evaluation goals. A list of the HOPE days for which ICON runs are already available is provided in Table 5. It should be noted that the number of modelled HOPE days is subject to change in the future and that ICON runs for dates not covered by HOPE were also already performed but are not shown in here. The HOPE days selected for ICON runs cover a wide range of meteorological conditions, from clear-sky days for the evaluation of convective processes in the planetary boundary-layer to days on which frontal passages accompanied by large-scale precipitation occurred. Most evaluation efforts were so far performed in a study of Heinze et al. (2017), but also others already made use of the extensive observational dataset. The studies available so far are discussed below.

**4.1 Examples of model-observation inter-comparisons**

The observational studies presented in Section 3 demonstrate well that large efforts are being taken to make observations suitable for the initialization and the evaluation of numerical weather prediction (NWP) models and to provide process studies that are essential for their improvement. The high temporal resolution of the HOPE dataset allows an analysis beyond the mean, which offers new opportunities to improve the simulation of boundary layer dynamics. Vertical profiles of higher-order moments (variances and turbulent fluxes) can be derived (Behrendt et al., 2015; Van Weverberg et al., 2016) which are essential to advance higher-order closure parameterizations of turbulent transport schemes in numerical models. Recent large-eddy simulation studies analysed the underlying sources and sinks of such prognostic higher-order moment equations for the cloud topped boundary layer (Heinze et al., 2015) and precipitating shallow cumulus regime (Schemann and Seifert, 2017). While these studies underline the importance, more robust conclusions are achieved by combining synoptically realistic model simulations with accompanying observational studies.

Nevertheless, operating a forecast model at scales that are small enough to resolve the different supersites of the HOPE-Jülich campaign puts certain requirements on the capabilities of the model. When the model resolution is between large-eddy simulations (with resolved energy-containing turbulence) and mesoscale simulations (no turbulence resolved), the model is operating in the so-called "grey zone" where more-sophisticated physical parameterizations (e.g. for boundary-layer turbulence or cloud microphysics) might be needed. To what extent the parameterization of turbulence and shallow convection is still necessary has been one of the key subjects of HD(CP)². Based on HOPE-Jülich observations, the grey zone was investigated in a study of Barthlott and Hoose (2015) who performed simulations with the COSMO model at horizontal resolutions ranging between 250 m and 2.8 km for six HOPE IOPs and one additional summertime case of the same year of 2013. From the kinetic energy spectra derived from the model output, it was found that the effective resolution (the minimum size of resolvable eddies) lies between 6 and 7 times the nominal resolution. Finer resolutions improved the representation of boundary-layer thermals, low-level convergence zones and gravity waves, but the effect on the temporal evolution of mean precipitation was rather weak. However, due to sensitivities of the rain intensities to model resolution, differences in the total rain amount of up to +48% occurred. Whereas the location of rain was rather similar at all model resolutions for the springtime cases of HOPE with moderate to strong synoptic forcing, the summertime case with airmass convection showed strong differences between the different resolutions with better agreement to the observed precipitation amount at the highest resolution of 250 m.

A major goal of HD(CP)² has been to use high-resolution modelling to derive parametrizations for climate models and general circulation models. In this respect the vertical cloud overlap parametrization is of high interest as it strongly influences the distribution of energy. In the past, such parametrizations have only been tested against observations on a global scale or for deep convective clouds. For the first time, Corbetta et al. (2015) investigated cumuliform cloud overlap for several boundary layer cloud cases including HOPE and compared it with the results from LES runs of the DALES model. Gridded time-height data from Cloudnet were used to derive cloud fraction masks at various temporal and vertical resolutions. The authors investigated the overlap ratio, i.e., the ratio of the cloud fraction by volume to the vertically averaged cloud fraction by area of

a grid box, as a function of the vertical resolution of the grid box. Cumuliform-cloud overlap ratios were found considerably underestimated by the LES model. For model-layer depths of less than 100 m, the modelled cloud overlap deviated by less than 7% from the observed one. The difference gradually increases to 15% for layer depths of 500 m and approached 20% for larger layer depths. Stratiform clouds were found to be better reproduced by the model, compared to cumuliform clouds.

5    Interestingly, the simulated and observed decorrelation lengths found for this type of clouds are smaller (~300 m) than previously reported (>1 km). The authors conclude that the inefficient overlap found at large vertical scales has the potential of significantly affecting the vertical transfer of radiation in large-scale GCM, because usually volume and area cloud fractions are assumed to be identical. The study can thus help to improve corresponding sub-grid parametrizations.

The evaluation of actual LES simulations of the HOPE-Jülich area was done by Heinze et al. (2016) who performed simulations

10    with PALM and UCLA-LES (University of California, Los Angeles Large-Eddy Simulation model, Stevens et al. (2005)) at up to 50 m horizontal resolution over the HOPE domain for a 19-day time period in order to capture a variety of different atmospheric and especially boundary layer conditions. The general weather pattern was reproduced in 80% of the cases. Also cloud types usually agree well with observations. Resulting turbulence characteristics and boundary layer heights have been compared to observations from active remote sensing (Doppler lidar and aerosol lidar) and from in-situ radiosonde

15    observations as proposed by Schween et al. (2014). Figure 16 exemplarily shows the temporal evolution of the boundary layer height as derived from different model runs and from observations. The 2-hour (12-14 UTC) mean boundary layer depth derived with the PALM model agreed within 400 m to the different observation methods and to the COSMO-DE run at 2.8-km resolution. The found differences are pointing to problems in the representation of ABL features in the LES, and should be subject of further investigations. Please note that the criterion of model-based ABL depth is also subject to uncertainties

20    which is explained further by Milovac et al. (2016) who found similar deviations between measurements and observations as found by Heinze et al. (2016). Heinze et al. (2016) further compared the observed turbulence characteristics of the ABL with the LES model. Observed and modelled profiles of the vertical-velocity variance agreed in their shape with the modelled values being in the range of uncertainty of the observations and showing slightly higher values throughout the boundary layer. Modelled profiles of potential temperature variances were found to be lower than the TRRL observations. For humidity

25    variance, agreement within the uncertainty range was found in the lower and mid-CBL between measurements and LES models. But the modelled variance peaks at the CBL top showed an under-estimation when compared with observations. Significant differences with respect to results from coarser resolved COSMO simulations were not reported. This might in part be due to the so-called semi-idealized set up with periodic boundary conditions and a homogeneous surface forcing. The authors also conclude that the simulated longwave and shortwave surface fluxes simulated with the LES model can be seen as

30    representative in comparison to respective observations at 5 different sites in the HOPE area. The peak shortwave heat flux in the LES and COSMO-DE tends to be overestimated compared to the weighted average, whereas the longwave heat flux tends to be underestimated.

Furthermore, within the Synthesis Module of HD(CP)², high-resolution ICON runs with 625-m, 312-m, and 156-m resolution were extensively evaluated against datasets collected during HOPE-Jülich and from other sources (Heinze et al., 2017). It was

found that the highest-resolved ICON-LES model matches much better the observed variability at small- to meso-scales than the coarser-resolved model runs or the reference model COSMO-DE with its 2.8-km horizontal resolution. It was demonstrated that the simulated turbulence profiles of the vertical velocity approach the observed ones for an increase in the ICON horizontal resolution from 625 m to 156 m. Differences between observed and modelled variance profiles of potential temperature and

5    specific humidity were much larger, which was explained by the absence of surface and soil moisture inhomogeneity in the model setup. The integrated water vapour of all models matched the range of values from the observations, but the temporal variability at short timescales as it was observed with microwave radiometer on a 1-s basis was only reproduced by the 156-m resolution run of ICON. From direct comparisons between modelled and continuous ground-based observations of the cloud field during HOPE-Jülich it was however found that convective boundary layer clouds are under-represented in the model,

10    even though the evaluation of the cloud fields on a larger scale, i.e. in comparison to satellite observations, showed that clouds are well represented in the model.  Heinze et al. (2017) concluded that, despite the given potential for further improvement of the ICON-LES model, it already fits well to the purpose of using its output for parameterisation development.

Regarding the application of HOPE observations for the initialization of NWP models, a first attempt was recently reported by Adam et al. (2016) who concentrated on the 24 April 2013 (IOP 6). In their study the authors assimilated lower-tropospheric

15    temperature profiles from the TRRL, reaching from about 500 to 3000 m above ground, into the Weather Research and Forecasting (WRF; Skamarock et al. (2008)) model using a 3D-variational method (Barker et al., 2004). The WRF model was covering Central Europe with 57 vertical levels and 3-km horizontal resolution. The assimilation of the temperature profiles from the TRRL in addition to the assimilation of conventional data including zenith total delay integrated water vapour field from the Global Navigation Satellite System and operational radiosonde data were found to improve the agreement of measured

20    boundary layer height and temperature gradient to the modelled values. Nine hours after the assimilation of TRRL data was initialized, already an area of 100 km in radius around the HOPE-Jülich area was affected, showing a temperature deviation from the conventional run of up to 2.5 K at 2.5 km height above sea level. Similar impacts can also be expected for the assimilation of profiles of water vapour mixing ratio from continuous lidar observations, as was found in an earlier study of Grzeschik et al. (2008).

25   **5    Summary & conclusions**

[revised manuscript text omitted]

With the completion of the high-resolution ICON LES model a vast number of model evaluation work is currently in progress.
5  First evaluation studies based on HOPE data have shown general agreement between observed and modelled boundary layer height, turbulence characteristics, and cloud coverage, but also point to significant differences that deserve further investigations, both from the observational and from the modelling perspective. Although the meteorological conditions which were prevalent during HOPE-Jülich and HOPE-Melpitz enabled the collection of a broad set of observations, it is obvious that the experimental coverage of the atmospheric boundary layer requires ongoing measurement efforts. In particular the
10  continuous observations from the German supersites will contribute to these efforts. The supersites JOYCE, KIT, and LACROS that have been deployed during HOPE-Jülich continue their long-term measurements at their base institutes and will contribute to further process and model evaluation studies in conjunction with further national and international supersites like Barbados (13.2° N, 59.4° W), Cabauw, the Netherlands, (51.9° N, 4.9 ° E), Lindenberg, Germany, (52.2° N, 14.1° E), Zugspitze mountain, Germany, (47.4° N, 11° E) as well as mobile facilities from the US (ARM) and Germany (mobile deployments of
15  the KIT cube, LACROS) under specific climatological and meteorological conditions.

Future work will take advantage of the synergy of the different active and passive remote sensing measurements. For instance, Doppler lidar and polarimetric radar measurements may link dynamical forcing (up and downdrafts) with microphysical processes (riming, coagulation, ice formation). The cloud radars of JOYCE, KITcube and LACROS were occasionally operating in a synchronized scan mode. Together with vertically pointing and scanning microwave radiometer data, three-
20  dimensional distributions of cloud liquid water may be constructed, and may get even further refined from cloud structure stereoscopy from synchronized sky imager data. Radiation closure studies will be performed based on observed and modelled spatial cloud structures and observed surface radiation budget measurements. High-resolution irradiance data can be used to build stochastic irradiance simulators for specific cloudy sky conditions, which in turn can be used to construct realistic cloud induced solar radiation variability. Combined measurements of temperature, humidity, and vertical wind fluctuations in the
25  PBL under different meteorological conditions will provide important statistical information for improved turbulence parameterizations. HOPE also demonstrated the future potential of the synergy of scanning wind, temperature, and water-vapour lidar systems for 3D studies of land-atmosphere exchange and ABL entrainment in heterogeneous terrain. HOPE data may also reveal to what extent variations in aerosol concentrations and thus in CCN and IN concentrations have an effect on cloud and ice formation compared to dynamical forcing.
30  In future, HOPE data will continue to contribute to the development, evaluation and improvement of high-resolution NWP and LES models because the data will be available via the Standardized Atmospheric Measurement Data (SAMD) data base which fulfils the needs of in particular model experts. Focused on the ICON development and on the collection of observational data for model evaluation, Phase 1 of HD(CP)² set the starting point for an ongoing, synergistic use of HOPE and other observational data by the modelling community. In Phase 2 of HD(CP)², which started in 2016, HD(CP)² participants are

already making use of these observations. For instance, a project on boundary layer clouds will confront ICON with HOPE data for different cloud regimes at different spatiotemporal scales. A project addressing fast cloud adjustment to aerosols will exploit remote-sensing and in-situ observations of aerosol and cloud properties to evaluate the susceptibility of the model performance to different representations of aerosol in the model, e.g., to variations in the concentration of nuclei for cloud droplets or ice crystals. A project on the effects of surface heterogeneity e.g. uses the HOPE observations to challenge the applicability of the Monin-Obukhov Simularity Theory (MOST) and the reproduction of the vertical boundary layer structure and turbulence on small scales. Other projects apply the observations of the 3D water vapour fields and the cloud microphysical properties derived with Cloudnet for the development of convection parameterizations, just to mention a few.

Thanks to the valuable efforts of the community of observers during the HOPE campaigns and given its open-access availability in the SAMD database (See Sect. 2.2.3) the HOPE dataset can serve as excellent tool for the model evaluation and initialization community.

**Acknowledgements**

The work summarized in this review was mainly carried out in the project HD(CP)[2] funded by the German Ministry for Education and Research. We specifically acknowledge the HD(CP)[2] projects 01LK1212A (University of Hohenheim), 01LK1209D (University of Leipzig), 01LK1209B (FZJ), 01LK1209C and 01LK1212C (TROPOS), 01LK1212F and 01LK1204B (KIT), 01LK1219A and 01LK1210A (University of Bonn), 01LK1203B (University of Hannover), 01LK1203A (MPI Hamburg). We also refer to all acknowledgements in the publications cited in section 3.

HOPE is particularly grateful to the Research Center Jülich and RWE Power AG (Hambach) that provided generous logistic support during the Jülich campaign. We thank the Transregional Collaborative Research Centre 32 "Patterns in Soil-Vegetation-Atmosphere Systems - Monitoring, Modelling and Data Assimilation" for contributing their valuable rain observation research infrastructures to the Jülich campaign.

The universities of Cologne and Bonn as well as TROPOS secured intense radiosonde observations from internal budgets.

Raman lidar system BASIL were funded on the basis of a specific cooperation agreement between Scuola di Ingegneria - Università degli Studi della Basilicata, TROPOS and MPI Hamburg.

We appreciated the provision of four Sun photometers for HOPE-Jülich and one device for HOPE-Melpitz by Goddard Space Flight Center, Greenbelt, MD, USA.

[revised manuscript text omitted]

| Date | IOP | Weather conditions |
|---|---|---|
| 20.04.13 | IOP 5 | Clear sky with only some cirrus clouds in the morning and late afternoon |
| 24.04.13 | IOP 6 | Clear-sky day with only few cirrus clouds in the morning and afternoon |
| 25.04.13 | IOP 7 | Cloudy morning (up to 4/8) until 10 UTC, only few clouds during noon, afterwards again increasing cumulus humilis cloudiness |
| 26.04.13 | IOP 8 | Rapidly increasing cloudiness up to complete overcast situation until noon, several rain showers and light to medium rain, decreasing cloudiness in the late afternoon |
| 02.05.13 | IOP 10 | Broken cumulus mediocris cloudiness, decreasing cloud cover during afternoon |
| 05.05.13 | IOP 12 | Clear-sky conditions until 09 UTC, afterwards slightly increasing cumulus humilis cloudiness up to (2/8) |
| 11.05.13 | - | High cloud cover until noon with several rain showers, afterwards broken cloudiness |
| 28.05.13 | IOP 18 | Clear sky conditions until midday (10 UTC) with only very few cirrus clouds, following low cumulus humilis clouds until 17 UTC, afterwards rapidly increasing cloudiness with rain starting in the evening |

[Figure]

**Figure 1: Setup of the HOPE-Jülich campaign showing the location of the three supersites Jülich (JUE), Hambach (HAM), and Krauthausen (KRA) as well as the outpost Wasserwerk (WAS) with their main instrumentation. The cones and arrows illustrate the field-of-view and scanning capabilities of the specific remote-sensing instruments.**

[Figure]

**Figure 2: Illustration of the setup of the HOPE-Melpitz campaign showing the deployed main instrumentation. The cones illustrate the field-of-view of the specific remote-sensing instruments.**

[Figure]

**Figure 3: Map of the spatial distribution of the measurement sites and networks deployed according to Table 1 (left) and a zoomed-in view centred at supersite Jülich (right). Background colours indicate the topography and dashed lines denote circles of constant distance from supersite Jülich (JUE). Shaded areas denote open-pit mines, for which the elevation map is not up to date.**

[Figure]

**Figure 4: Topography around the location of the HOPE-Melpitz campaign. (a) large-scale topography; (b) aerial photograph of the Melpitz field site with the locations of the pyranometers of the PYR network.**

[Figure]

**Figure 5: Spatiotemporal characteristics derived from the pyranometer network under broken-cloud conditions during HOPE-Jülich. This figure illustrates the origin of deviations between a point measurement (labelled as var(TD) in the legend) and a domain-averaged value (representativeness error) for broadband solar atmospheric transmittance and irradiance for different domain sizes. (a) Power spectra of transmittance for a point measurement and domains with different sizes; (b) Explained variance of temporal fluctuations in a point measurement and a domain average as function of period, and (c) total expected deviation between a point measurement and a domain average for transmittance and irradiance as a function of averaging, assuming a value of 680 W m$^{-2}$ for the incoming solar irradiance at the top of atmosphere. The time period of fluctuations (inverse of their frequency) is shown logarithmically on the x-axis. Adapted from (Madhavan et al., 2016a).**

[Figure]

**Figure 6: Integral scales of the temperature fluctuations (black) and humidity fluctuations (red) in the convective boundary layer derived from high-resolved temperature observations obtained between 1130 and 1330 on 20 April 2013 (IOP 5) and 1100 and 1200 UTC on 24 April 2013 (IOP 6) during HOPE-Jülich. Heights are normalized with respect to the height of the convective boundary layer height $z_i$. Adapted from Behrendt et al. (2015), Muppa et al. (2016), and from Di Girolamo et al. [2016].**

[Figure]

**Figure 7: Simultaneous observation of the vertical velocity variations in a stratocumulus layer performed in-cloud with ACTOS (red) and at cloud base with Doppler lidar WiLi of LACROS (blue) on 22 September 2013 during HOPE-Melpitz. The mean vertical velocity of both observations was set to zero to correct for large-scale vertical motions. Adapted from Seifert et al. (2017).**

[Figure]

**Figure 8: Observation of the integrated water vapour (IWV) during HOPE-Jülich for a large suite of different instruments. Right panel shows the frequency distribution of the IWV values recorded with the different techniques. Bottom panel shows the accumulated amount of precipitation. Adapted from Steinke et al. (2015).**

[Figure]

**Figure 9: Calibrated nighttime observations at KRA of the water vapour mixing ratio for April 2013 during HOPE-Jülich obtained from Polly[XT] that were calibrated automatically with the integrated water vapour provided by a co-located microwave radiometer. Adapted from Foth et al. (2015).**

[Figure]

Figure 10: Aerosol target classification for the HOPE-Jülich period from 24 to 26 April 2013 (IOPs 6-8) based on continuous observations of the multi-wavelength polarization lidar Polly[XT]. The methodology is described in Baars et al. (2017).

[Figure]

Figure 11: Correlation between the particle extinction coefficient derived from Mie modelling and hygroscopic-growth correction of in-situ measurements of ACTOS with the respective ones measured with Polly[XT]. The data set is based on 13 data points obtained at different altitudes during two ACTOS flights on 14 and 17 September 2013 during HOPE-Melpitz. Adapted from Düsing et al. (2017).

[Figure]

**Figure 12: Stratocumulus observation at the Melpitz site on 22 September 2013. Time-height cross sections of (a) cloud droplet effective radius and (b) liquid water content as observed from ground-based remote sensing with LACROS. (c-d): Profiles of single data points, mean, and standard deviation (horizontal bars) of (c) liquid water content and (d) effective radius as observed in-situ with ACTOS (red) a and retrieved from LACROS (black) for the time period shown in (a) and (b). Scaled-adiabatic method is based on Merk et al. (2016), Frisch-2002 method is based on Eq. 5 of Frisch et al. (2002).**

[Figure]

**Figure 13: Relationship between mean ice water content (IWC) and ice-to-liquid mass ratio as a function of cloud top temperature of all thin supercooled stratiform clouds detected during HOPE-Jülich. The colours represent the different radar linear depolarization ratios.**

[Figure]

**Figure 14: Time series of rain rates derived from observations of seven disdrometers (including those from the TR32 program) and the three polarimetric radars on 29 May 2013. The shaded grey area indicates the range of rain rates observed by the disdrometers with 1 min temporal resolution in the HOPE area, while the rain rate from the three polarimetric radar observations is calculated at the radar gates that are coincident with disdrometer locations and also averaged over the disdrometer locations. From Xie et al. (2016).**

[Figure]

**Figure 15: 3D reconstruction of a cumulus tower from a stereographic photograph from 24 July 2014, 11:32:00 UTC. Shown are (a) subsection of the image obtained from the reference camera, (b) the reconstruction as an untextured triangulated surface mesh, (c) the color-coded height of the reconstruction with contour lines, and (d) reconstructed distance of the cloud edges from the reference camera obtained along the cross-section (dashed line) shown in (a), (b), and (c) as well as a comparison of the cloud base with the one observed with lidar ceilometer (blue line). Adapted from Beekmans et al. (2016).**

[Figure]

**Figure 16: Temporal evolution of the boundary layer depth $z_i$ for the period from 24 to 30 April 2013. $z_i$ is determined by means of the bulk-Richardson number criterion in all three models (PALM, UCLA-LES and COSMO) and in the radiosonde data. A criterion based on the vertical velocity variance and detected aerosol layers is used for the wind lidar and aerosol lidar PollyXT, respectively. The data point obtained from the temperature rotational Raman lidar (TRRL, rot. Raman lidar) is based on Behrendt et al. (2015). Radiosondes were launched at the KITcube site, the Doppler lidar and Polly$^{XT}$ took measurements at sites JUE and KRA, respectively. Grey and green shading denote twice the standard deviation of $z_i$ in PALM and UCLA-LES, respectively. Adopted from Heinze et al. (2016).**

---

## Author Comment (AC3) · 9 Feb 2017

Dear Referee,

find attached to this post the updated manuscript, including the markup. The final manuscript file without markup was already attached to the reply letter.

Sincerely, Patric Seifert.

Please also note the supplement to this comment:
http://www.atmos-chem-phys-discuss.net/acp-2016-990/acp-2016-990-AC3-supplement.pdf
* * *